# A modular framework for multi-scale tissue imaging and neuronal segmentation

Simone Cauzzo [1,2] ✉, Ester Bruno [1,3], David Boulet [4,5], Paul Nazac [5], Miriam Basile [3], Alejandro Luis Callara[1,3], Federico Tozzi[1,3], Arti Ahluwalia[1,3], Chiara Magliaro[1,3], Lydia Danglot [4,5,6] ✉ & Nicola Vanello [1,3,6] ✉

The development of robust tools for segmenting cellular and sub-cellular neuronal structures lags behind the massive production of high-resolution 3D images of neurons in brain tissue. The challenges are principally related to high neuronal density and low signal-to-noise characteristics in thick samples, as well as the heterogeneity of data acquired with different imaging methods. To address this issue, we design a framework which includes sample preparation for high resolution imaging and image analysis. Specifically, we set up a method for labeling thick samples and develop SENPAI, a scalable algorithm for segmenting neurons at cellular and sub-cellular scales in conventional and super-resolution STimulated Emission Depletion (STED) microscopy images of brain tissues. Further, we propose a validation paradigm for testing segmentation performance when a manual ground-truth may not exhaustively describe neuronal arborization. We show that SENPAI provides accurate multi-scale segmentation, from entire neurons down to spines, outperforming state-of-the-art tools. The framework will empower image processing of complex neuronal circuitries.

Describing the mammalian brain connectome at the cellular and sub-cellular scales is one of the holy grails in neuroscience[1–3]. The brain comprises billions of densely packed neurons[4]. Isolating single cells in their native arrangement within brain tissue represents a crucial step for describing neuronal shape, size and complexity, which enables in turn the characterization of cell types and the identification of any morphological abnormalities characterizing neuropathies[2,5]. Dendritic spines are tiny membranous protrusions on dendritic shafts that can be contacted by glutamatergic axons to form excitatory glutamatergic synapses. Spine morphology was first described by Ramon Y Cajal in 1888 using the Golgi Technique[6]. Their 3D morphology can be studied using the gold standard 3D electron microscopy (as reviewed in Harris et al. and Rasia-Filho et al.[7,8]). Dendritic spines can harbor various morphologies, from stubby spines (short bulges of the membrane) to mushroom shapes (with thin necks and bulbous heads)[9]. More recently, based on ultra-structural analysis of mouse neocortical dendritic spine necks, the existence of subtypes of spines has been questioned, and a continuum of spine morphologies has instead been proposed[10]. Dendritic spines receive electric inputs which are locally regulated by spine neck geometry and resistance[11,12]. Furthermore, spine shape and density are linked to synaptic function, such as learning, memory and motivation[13,14] and changes in synaptic activity are associated with alterations in spine shape, size and number[15,16]. Structural alterations of spiny synapses are found in the pathogenesis of major neurological disorders (ASD, schizophrenia and Alzheimer's disease)[17–19]. Reduction in spine density and synaptic loss are found in patients with Alzheimer's disease and Mild Cognitive Impairment

[1]Research Center "E. Piaggio", University of Pisa, Pisa, Italy. [2]Parkinson's Disease and Movement Disorders Unit, Center for Rare Neurological Diseases (ERN-RND), Department of Neurosciences, University of Padova, Padova, Italy. [3]Dipartimento di Ingegneria dell'Informazione, University of Pisa, Pisa, Italy. [4]Université Paris Cité, Institute of Psychiatry and Neuroscience of Paris (IPNP), INSERM U1266, NeurImag Core Facility, 75014 Paris, France. [5]Université Paris Cité, Institute of Psychiatry and Neuroscience of Paris (IPNP), INSERM U1266, Membrane traffic and diseased brain, 75014 Paris, France. [6]These authors jointly supervised this work: Lydia Danglot, Nicola Vanello. ✉e-mail: simone.cauzzo@unipd.it; lydia.danglot@inserm.fr; nicola.vanello@unipi.it

(MCI)[17,20,21], suggesting their collective and individual role in brain dysfunctions[18,21].

Several imaging methods have been developed to explore neuronal structures at different spatial scales. Among these, single[22] and multi-photon confocal microscopy[23] are the cardinal tools to visualize neurons from ex vivo or in vivo samples within their native environment, since they offer deep imaging solutions, with lateral and axial resolutions from 180 nm in the *XY* plane to 500 nm in the Z direction, respectively[24]. On the other hand, super-resolution microscopy (e.g., STimulated Emission Depletion−STED), enables the characterization of tiny dendritic spines[25–28]. However, brain tissue is prone to scattering which limits the imaging depth (Z) in the tissue. The performance of both confocal and super-resolution imaging can be improved via tissue clearing and refractive index matching[29], which enable the acquisition of larger sample volumes with improved signal-to-noise and contrast-to-noise ratio (SNR and CNR), while maintaining low laser power[30].

Parallel to the advances in brain tissue labeling and imaging at both the micro and nanoscale, automatic algorithms aimed at high-fidelity three-dimensional (3D) reconstruction of single neurons have been extensively developed. Recent reviews on the topic[2,31] provide a categorization point for more than forty algorithms (see Supplementary Table 1), each one exploiting key aspects of image properties (e.g., signal-to-noise ratio, point-spread-function, image contrast) and/or neuron features (e.g., axon tubularity, soma sphericity, tree-like structure) for developing a custom neuron reconstruction strategy. As such, their success is strongly influenced by both the acquisition modality and cell type and by the kind of features they rely on. For example, local methods based on a progressive propagation of the tracing or segmentation from a reference point are particularly sensitive to noise or local inhomogeneity, while global methods based on SNR will fail at handling smooth changes in SNR across the image stack. For this reason, developing general-purpose solutions is often impracticable. Some of these issues are partially solved by meta methods, i.e. modules that can be used on top of base tracing or segmentation algorithms to solve specific problems and improve the result (e.g., G-cut[32] for post-hoc separation of interweaving neurons and UltraTracer[33] for increasing the scalability of tracing algorithms to large image stacks). It is worth highlighting that most of the algorithms for neuron segmentation from microscopy images proposed cannot isolate neurons acquired with different imaging modalities, and only few of them are able to deal with densely packed cells[32,34–36].

Similar considerations apply to methods based on Artificial Intelligence (AI) approaches such as machine learning and deep learning (for a summary categorization see Tables 3 and 5 in Chen et al.[37]), which have been successfully introduced thanks to the availability of high quality and large training datasets[38–40]. For the same reasons, convolutional neural networks were recently introduced[38–40]. AI approaches are in fact rapidly emerging as a solution to the complex problem of neuronal reconstruction in microscopy images. However, their direct application to previously unseen data (e.g., different acquisition modality, cell type, SNR levels) strongly depends on the training set (e.g., type of data, number of training examples), or ad-hoc solutions (e.g., transfer learning[41,42]), limiting their generalizability. Although considerable effort is now being made to achieve explainability in deep learning models[43], their black-box nature makes it difficult for users to exercise control over common issues such as over- or under-segmentation. In addition, existing algorithms based on deep-learning currently struggle in dealing with separating multiple structures within an image block[37] and to retrieve information beyond the limits posed by image resolution, i.e., for spine morphology or spine neck reconstruction. Finally, independent of the method used, manual correction and fine-tuning of the automatic results by experts are still required[44].

Neuroimaging data obtained from clarified tissues[2], a genre of methods which clear high opacity tissue lipids, represent unique features in terms of both image quality (e.g., enhanced signal intensity, SNR, CNR) and the capability of imaging densely packed cells in a 3D arrangement[2,34,45,46]. But, the generalizability of neuron reconstruction algorithms is even lower when dealing with images from densely packed neurons. Moreover, the outcomes of such algorithms and tools are compared with a manually segmented ground truth reconstruction, which is time consuming, difficult to achieve and prone to human bias.

In the light of these challenges, we have designed a framework for extracting faithful morphological information from brain tissues at the neuronal and sub-neuronal level via imaging and image processing tools. Specifically, we developed a method for processing and acquiring brain tissue which returns large volume datasets representing neurons with an improved image quality. Then, we present a data-driven approach−SENPAI (SEgmentation of Neurons using PArtial derivative Information)−for neuron segmentation from 3D optical images. SENPAI benefits from image topological information and K-means clustering to provide faithful descriptions of isolated neuronal morphologies at multiple levels of detail which does not rely on constructing models of image signal and/or noise. Decision-making routines select classes representing neuronal structures in the image. K-means based segmentation is complemented by a parcellation step exploiting topographic distances. We employ morphological reconstruction and the watershed transform to ensure the correct separation of neurons when they are closely packed. The same strategy is used for the grouping and assignment of small spines to the body of a specific dendritic tree in high-resolution datasets without detectable necks.

Here we demonstrate the performance of the proposed approach using images from both confocal and super-resolution STED microscopy. As a proof of concept, we focused on the L7GFP mouse line which expresses cytosolic GFP in cerebellar Purkinje cells. As these cells are particularly densely packed, we cleared the samples to improve imaging depth. Thanks to deep super-resolution 3D STED imaging we were able to resolve dendritic spines over wide dendritic trunks. The algorithm was tested against state-of-the-art software and manual segmentation. Specifically, we compared the complexity of the resulting segmentations and assessed their performance in the detection and correct assignation of dendritic spines to neurons. Furthermore, we provide an original validation paradigm for evaluating the quality of segmentation, in which morphometrics extracted from the segmented neurons are compared with reliable quantitative indices of Purkinje cell morphology available in the literature[47]. To evaluate the generalizability of the segmentation algorithm to other neuron types and imaging modalities, we tested SENPAI's performance using open datasets comprising both cultured and ex-vivo neurons. This study represents a significant advancement in the state-of-the-art for the comparison of segmented neurons, particularly when a reliable manually segmented ground-truth is not available and/or achievable.

## Results

### Labeling and super-resolution STED imaging of thick samples

Purkinje cells are characterized by a dendritic tree studded with numerous dendritic spines, receiving roughly 200,000 synapses[48]. To unravel their morphology, we cleared thick slices of L7GFP mouse brain samples[29,30] and processed them for long-term immunochemistry. The pre-processing steps and the labeling strategy are shown in Fig. 1A−C (see Methods for details).

To access several scale levels of information (from tissue slices with populations of cells, to the cell environment, a single dendritic tree, and local spine density) we set up the acquisition of correlative imaging using both low magnification for tissue scanning (20x) and high-resolution imaging (93x) of dendritic spine with either confocal or 3D STED microscopy. Reconstitution of wide mosaic brain area requires the use of tile imaging acquisitions that are very sensitive to

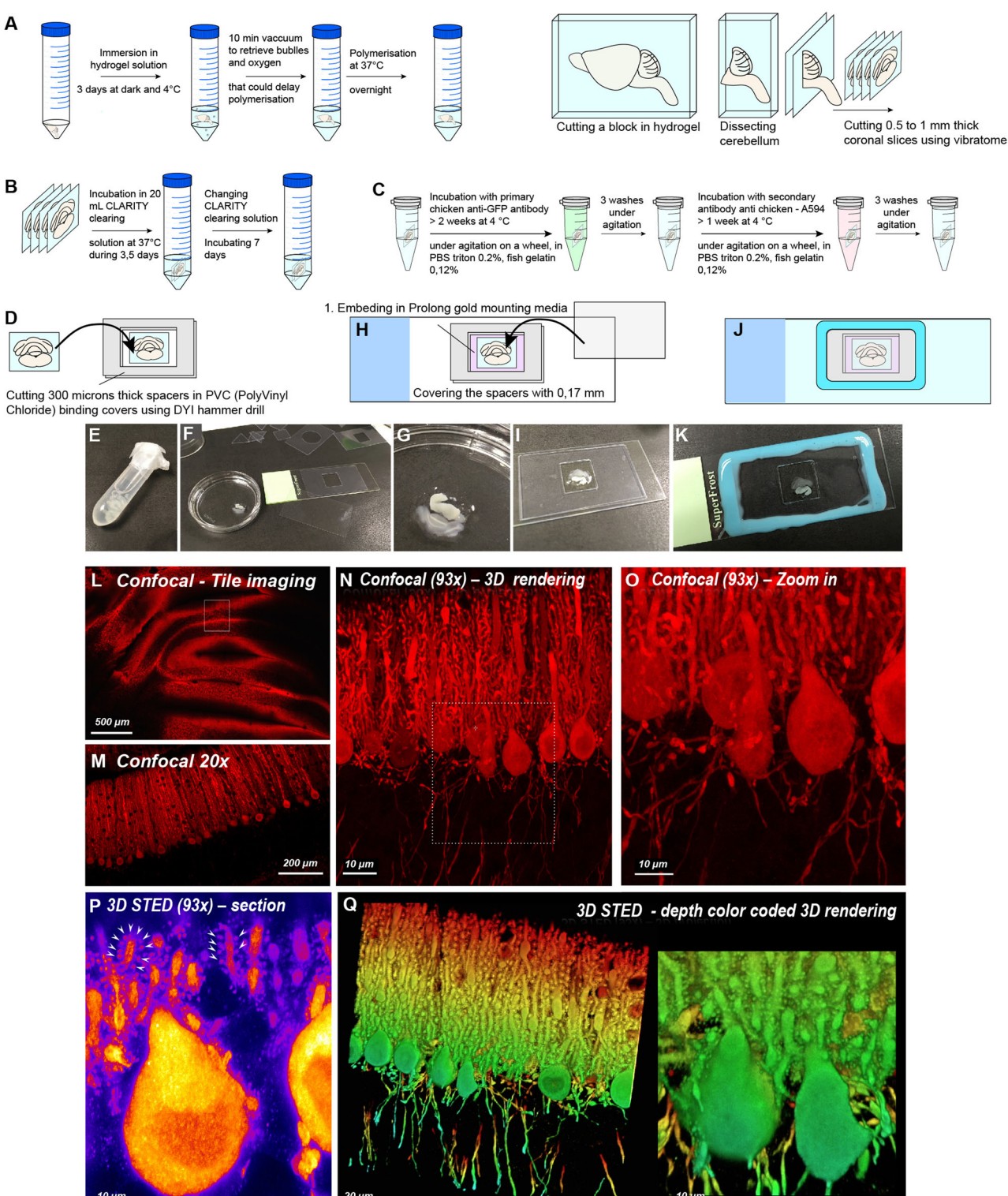

**Fig. 1 | Labeling strategy for thick imaging in 3D super-resolution STED microscopy. A** Embedding the tissue in hydrogel and cutting 6 slices (0.5–1 mm thick) using a vibratome. **B** Passive clearing via incubation in clarity solution. **C** Labeling protocol using primary and secondary fluorescent dedicated STED antibodies. **D–G** Floating slices mounted on a slide in the center of a homemade PVC spacer. **H, I** Spacer hole filled with prolong gold mounting media and covered with super-resolution 0.17 mm glass coverslip. Depending on slice thickness, several spacers can be stacked to keep coverslip parallel to the slide. **J, K** Spacers and coverslip sealed to the slide with silicone. **L** Confocal tile imaging of the cerebellum. **M** Confocal zoom in over Purkinje cells with 20x objective. **N** Conventional 3D confocal imaging over the Purkinje cell layer with dedicated Zoom in (**O**) around the cell body. **P** Single slice of 3D STED microscopy, revealing dendritic spine (white arrows) decorating the entire dendritic tree. **Q** 3D rendering of multiple 3D STED stacked slices showing the high density of dendritic spines within depth. Scale bar: 10 microns.

stage and sample flatness. Thus, to optimize sample flatness and working distance objective, we carefully designed custom-made spacers of calibrated thickness (300 microns thick PVC sheets) allowing light penetration directly into the samples without passing through thick mounting media layers. Using this device, glass coverslips were sealed parallel to the slice surface, allowing deep imaging using 93x 3D STED objective with 300 microns working distance (Fig. 1 D–J). Wide areas could be imaged over the cerebellum using tile 3D imaging with a 20x glycerol objective (Fig. 1L, M) that could be correlated to confocal (Fig. 1N, O) and 3D STED imaging (Fig. 1P, Q) at higher magnification (93x). As shown in Fig. 1, Purkinje cells are exquisitely packed in undulating layers, and dendritic trees can be inferred from conventional confocal microscopy. However, dendritic spines can be resolved more efficiently using 3D STED microscopy (Fig. 1P, Q and Supplementary Movie 1). Because cell and spine densities are high, manual segmentation would be quite time consuming and prone to human bias. Automatic segmentation using conventional thresholding method is not really efficient, since local intensity within Purkinje cells is quite different (Fig. 1P): dendritic trunk intensity is very high (yellow), whereas spine heads have moderate intensity (pink) and spine necks are pretty dark (deep purple). We thus designed the SENPAI pipeline to meet these biological and technical challenges.

### General workflow of SENPAI: step 1–segmentation

SENPAI is developed as a two-step automated workflow for reconstructing neuron morphology starting from 3D optical images (Fig. 2).

As intensity variations are very high in Purkinje cells, we choose to use model-independent topological information. The segmentation strategy in SENPAI consists of a K-means clustering of the image dataset in a 4-dimensional feature space defined by the intensity and the three second derivatives of the image intensity computed along the three main spatial axes (Fig. 2B). Six clusters, or classes of voxels are thus automatically identified and sorted by assigning numeric labels based on the average image intensity within each class, starting from the lowest (class 1) (Fig. 2C). The clusters are then automatically labeled as neuronal structure or background, based on their feature values. The way the image space is subdivided highlights inner and outer borders along the three main directions (Fig. 2D). Neuronal shape correlates with highest average image intensity classes that present negative values for the averages of the three second derivatives (Fig. 2B, C). However, thanks to the use of spatial derivatives, all classes span large intensity ranges: neuronal classes, despite showing high intensity values on average, are thus able to include low-intensity image regions; similarly, non-neuronal classes, despite showing low average intensity, also include high-intensity background regions (see Supplementary Fig. 1). This subdivision was observed to be stable across tests performed on different images (see Supplementary Fig. 1). 3D Gaussian image smoothing with user-defined size can be applied to modulate the values of the second-order derivatives. As the clustering can be focused on different levels of detail, it allows highlighting structures at different scales. This guarantees SENPAI's scalability and versatility since a final multiscale result can be obtained by merging two segmentations after applying different levels of smoothing to the original image (Fig. 2D).

### General workflow of SENPAI: step 2–parcellation

On datasets representing entire neuronal populations, branches belonging to different cells are close to one another (Fig. 1N–Q) and thus not discernible. As a result, segmentation algorithms may fail in isolating single neurons, or may succeed, but at the cost of providing more conservative (i.e., poor) segmentations. Similarly, in higher-resolution datasets, when the goal is to study spine shape and density, the difficulty of segmenting spine necks impairs the assignation of the spine to the corresponding dendrite. To overcome these issues, we included a second parcellation step in SENPAI based on the use of

morphological reconstruction and on the application of the watershed transform.

Morphological reconstruction is an iterative image transformation technique used to smooth out some less relevant local maxima or minima[49], while the watershed transform is employed to subdivide the image space into regions assigned to objects having outstanding intensity with respect to the background[50]. For each neuronal core, i.e., the most discernable portion of an imaged neuron, we define a single well (via morphological reconstruction) and an associated catchment basin employing a watershed transform. A different label is associated with each catchment basin to compose the parcellation process. When applied to isolate single neurons, i.e., to separate merged structures, neuronal cores can be easily defined by marking the soma or the thickest arbor observable for each neuron. On the other hand, when applied to merge unconnected structures, such as spines and their corresponding dendrite, the neuronal core consists of the dendrite branch (Fig. 2E).

### Comparison with other segmentation or tracing algorithms

Confocal images of cleared mouse brain tissue samples (see Methods and Figs. 1–2) were used to compare the segmentation of neurons obtained using SENPAI with those performed with four state-of-the-art algorithms, namely, the automatic HK-Means segmentation plugin[51] included in the software Icy[52] (hereinafter HK-Icy), the semi-automatic tracing software NeuroGPS[35], the semi-automatic segmentation software Ilastik[53] and the semi-automatic tracing software NeuTube[54]. When single neuron identification failed with a specific algorithm, we parcellated its outcome using the step 2 of the SENPAI workflow to provide a reconstruction comparable with the others. The example in Fig. 3 shows how neuron segmentations performed by SENPAI and HK-Icy result in a notably denser and more intricate outcomes than those from NeuTube and NeuroGPS, while Ilastik provides richer segmentations whose complexity lies in between. We also report the 27 single-neuron reconstructions obtained with SENPAI and the state-of-the-art tools and used to test and validate SENPAI in Supplementary Figs. 2, 3 and 4.

We also compared the segmentation performance of SENPAI, HK-Icy, Ilastik, NeuroGPS and NeuTube in terms of the area under the curve (AUC) obtained through Sholl analysis, a method used to quantify dendritic structure[55] (see Fig. 4A). SENPAI, HK-Icy and Ilastik show significantly higher values with respect to Neutube and NeuroGPS (Friedman test, $p < 10^{-16}$, and post-hoc comparisons with Tukey's honest significant difference criterion, $p < 10^{-2}$, Fig. 4B, right) indicating a higher sensitivity to complex dendritic arbor detection. Additionally, we compared the area-to-volume ratio of the segmentations provided by SENPAI, HK-icy and Ilastik (Fig. 4B, left). It should be noted that this index cannot be extracted for NeuroGPS and NeuTube, since they do not provide volumetric information. While HK-Icy and Ilastik show similar values, SENPAI provides significantly higher area-over-volume ratios (Friedman test, $p < 10^{-8}$, and post-hoc comparisons with Tukey's honest significant difference criterion, $p < 10^{-5}$).

### Comparison against documented information on Purkinje cell morphometrics

Purkinje neurons branch extensively and almost completely fill spaces with little overlap[56]. They have been reconstructed using tools such as NeuroMorpho.org[57] from public repositories of single injected neurons[58,59] and several parameters have been used in the literature to describe their morphology. The availability of summary statistics describing the topology and morphology of neuronal populations allows for the validation of new reconstructions when the ground truth for the specific sample is lacking or the manual gold standard cannot be obtained due to high density neuronal packing. Vormberg and co-authors[47] described the topological and morphological properties of six types of neurons (including Purkinje cells) by means of the

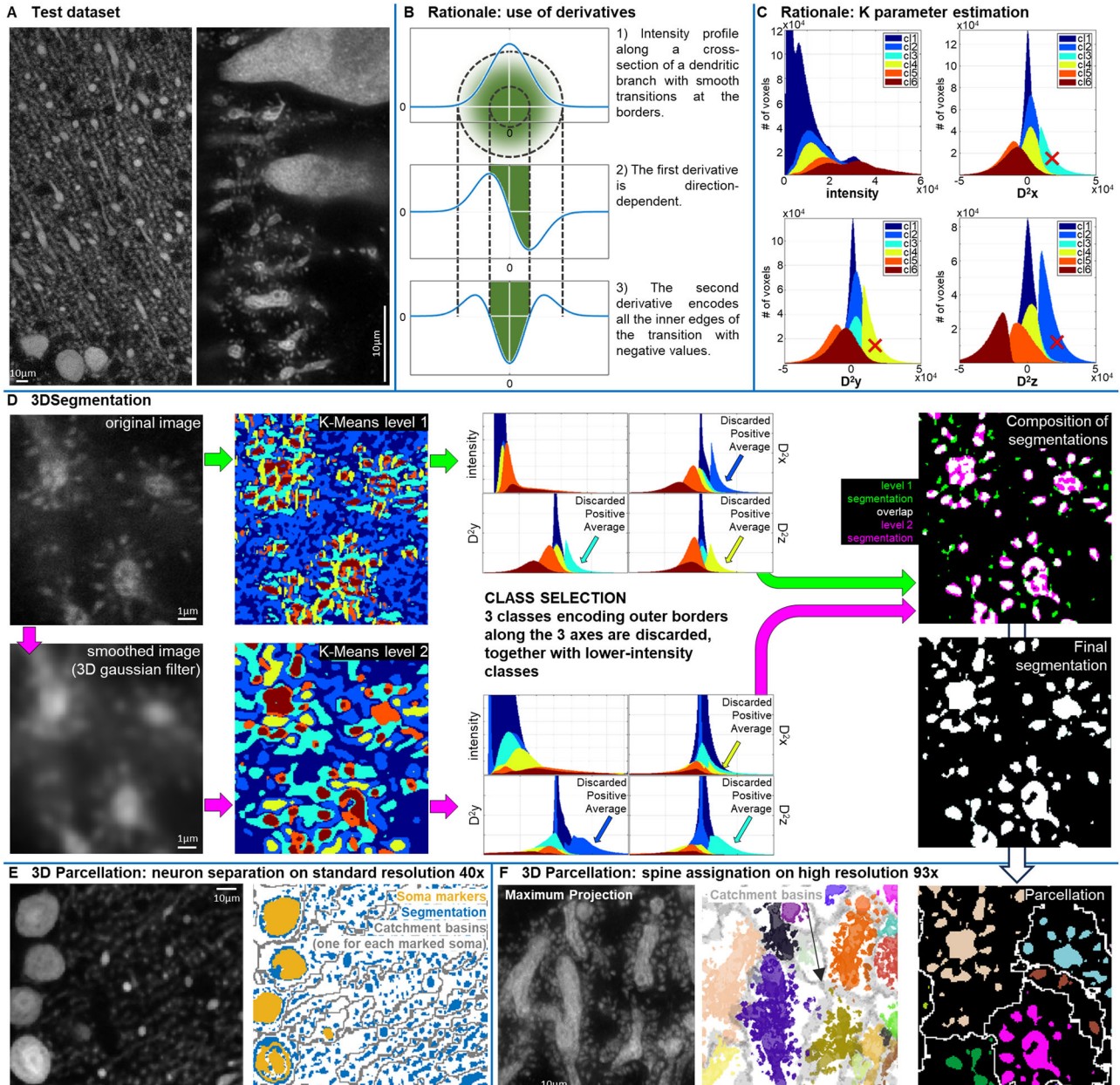

**Fig. 2 | The SENPAI algorithm. A** Test datasets: 40x confocal (left) and 93x STED (right). SENPAI was tested on cleared samples using confocal 40x (27 neurons) and 93x STED (5 neuron branches) datasets. **B** The SENPAI rationale, based on the selection of classes displaying negative values of the second derivative along the three main axes. **C** Rationale for the estimation of the K parameter of the K-means clustering: the selected K is the one for which the histograms of second-order derivatives show three different classes achieving maximal average value: here, from the histogram of $D^2x$ (second derivative along the x axis), class 3 (cyan) encodes outer borders along the $x$ direction, as it is the only class with values clearly above 0; similarly, class 4 encodes outer borders along the $y$ direction ($D^2y$ histogram); class 2 along the z direction ($D^2z$ histogram); class 1 encodes both low-and high-intensity homogeneous image portions, and is labeled as background along with classes 2, 3 and 4. **D** SENPAI workflow−Step 1: K-means clustering, performed on the unsmoothed image (top) and optionally, in parallel, on the image smoothed with a 3D Gaussian filter (below). Class selection is performed independently on K-means classes (color-coded as in (**C**) for each clustering level). Resulting binarized images (green for clustering level 1, pink for clustering level 2, white for the overlap) are merged by logic OR. **E** SENPAI workflow−Step 2 for neuron separation: the segmented image is parcellated using morphological reconstruction and 3D watershed transform computed on the morphologically reconstructed grayscale image and applied to the binary segmentation. Left: raw image; Right: isolation of connected structures belonging to the same neuron; soma markers (yellow, placed by user) define wells for the catchment basins (edges in gray). **F** SENPAI workflow− Step 2 applied to spine assignation. Left: raw image; Middle: 3D rendering; Left: 2D rendering of the parcellation with the connection of a neuronal portion (e.g., a dendrite branch) to smaller clusters (i.e., dendritic spines). Groups of neuronal clusters assigned to a single neuronal entity are displayed in different colors in their relative catchment basin (gray).

centripetal Horton-Strahler ordering (SO, see Fig. 5A). This is a well-known index of branching complexity, which accounts for nine different parameters. These include: (i) Strahler Number (SN), i.e., the maximal SO assigned to a segment in the neuronal tree. A further four topological parameters are defined: (ii) Normalized Number of Segments per SO, (iii) Normalized Number of Branches per SO, (iv) Branch Bifurcation Ratio, (v) Normalized Topological Subtree Size per SO. The final four metric parameters are: (vi) Normalized Branch Diameter per SO, (vii) Total Normalized Dendritic Length per SO, (viii) Normalized Average Segment Length per SO, and (ix) Normalized

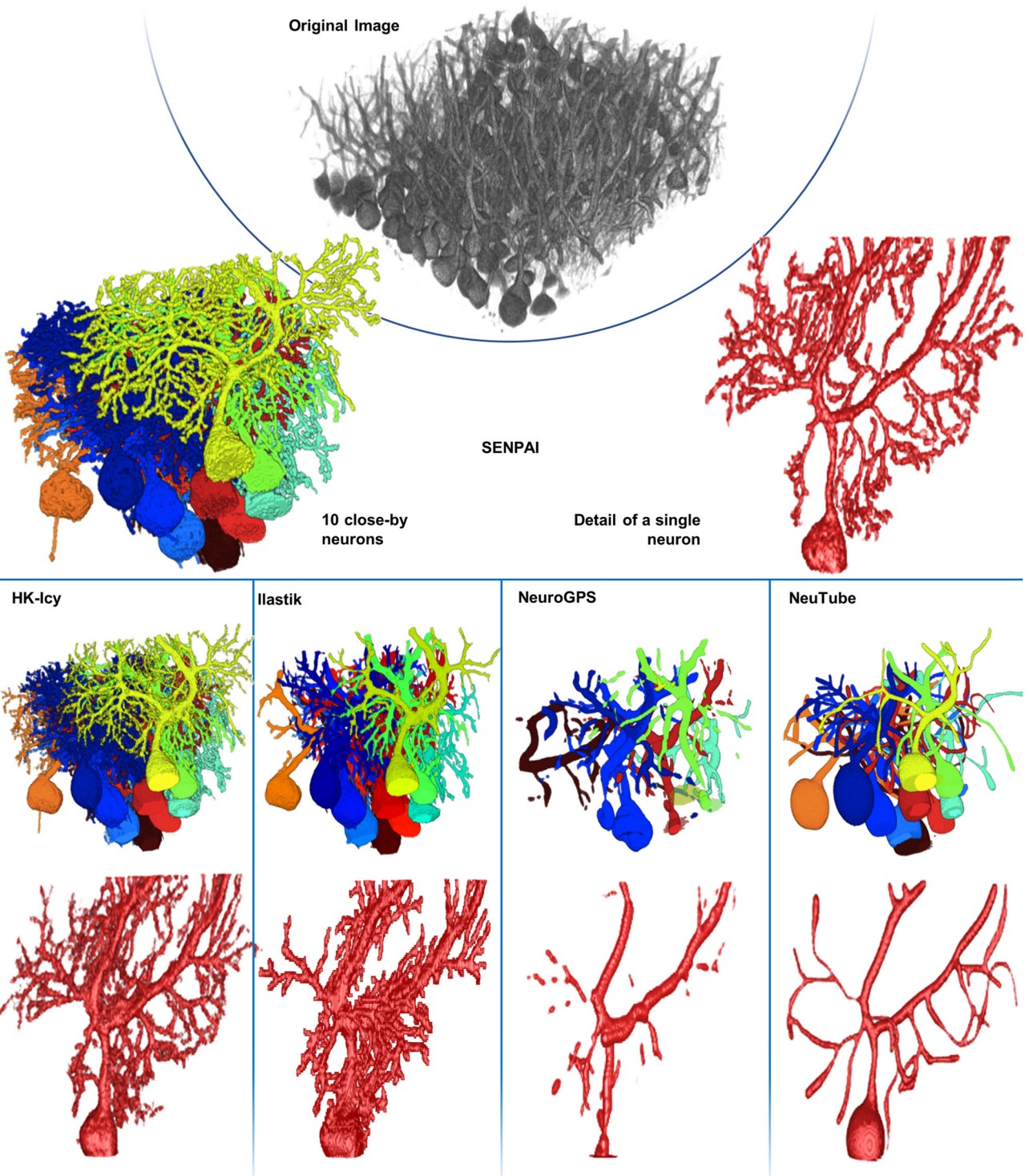

**Fig. 3 | 3D visualization of the original confocal dataset and comparison of the performance of the algorithms tested against SENPAI.** Dataset acquired with a confocal microscope equipped with a 40x objective (details in Methods). All images are produced with the Icy GUI. Tracings are converted to volumetric segmentations with the SWC2IMG ImageJ plugin[87]. The same 10 neighboring neurons and one exemplary single neuron are depicted using the same color code for all tools. A further example is depicted within a 3D rendering of the original image in Supplementary Movie 2.

Average Branch Length per SO. While topological parameters are universal to binary tree-like structures, metric parameters can be used to differentiate cell types. For example, total dendritic length reveals the dimensionality of dendritic trees, with shallow slopes for planar trees, e.g., as in Purkinje cells, and pronounced slopes for 3D trees.

Using an ad hoc script developed in Matlab (see Methods) (Fig. 5B), we extracted the SO for skeletonized neurons obtained from SENPAI, HK-Icy, NeuroGPS, NeuTube and Ilastik. The SOs were compared with those obtained by Vormberg et al. for Purkinje cells. The results are shown in Fig. 5C and detailed in the Supplementary Information (Supplementary Table 2).

SENPAI and HK-Icy achieved segmentations with the same SN mode as that identified by Vormberg et al. for Purkinje cells, with SENPAI scoring more hits. All measures reported in Fig. 5C show that

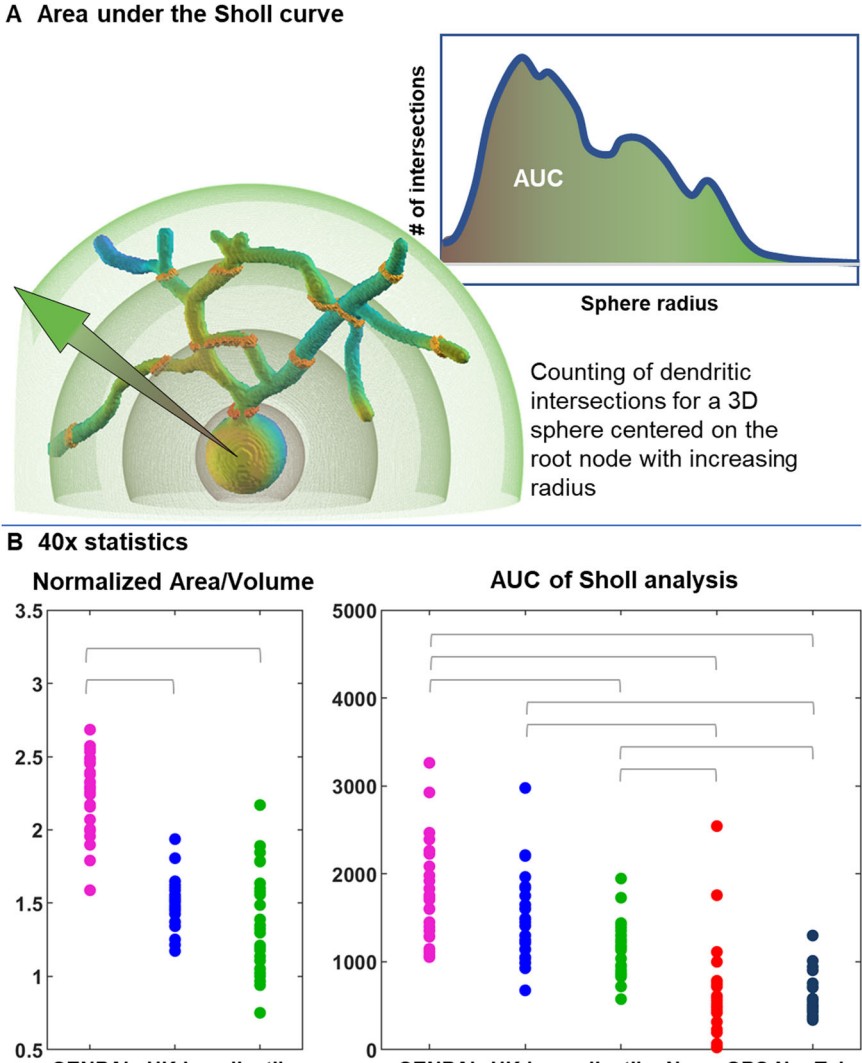

**Fig. 4 | Quantitative comparison of the segmentations on 40x images.**
**A** Schematization of the rationale behind 3D Sholl Analysis: we compute the Area Under the Curve (AUC) for the number of crossings of each neuronal structure for a sphere with increasing radius centered on the root node (i.e., the soma centroid); **B** Quantitative comparison for segmentations from 27 neurons on 40x images; left: Area-Volume ratio for SENPAI (magenta), HK-Icy (blue) and Ilastik (green) (two-sided Friedman test, $p = 1.02 * 10^{-9}$, horizontal bars mark significant differences as determined with post-hoc comparison using Tukey's honest significant criterion, $p < 10^{-5}$); right: AUC of the Sholl analysis across segmentations obtained with SENPAI (magenta), HK-Icy (blue), Ilastik (green), NeuroGPS (red) and NeuTube (gray), (two-sided Friedman test, $p = 6.81 * 10^{-17}$, horizontal bars mark significant differences as determined with post-hoc comparison using Tukey's honest significant criterion, $p < 10^{-2}$). Source data are provided as a Source Data file.

the segmentations obtained with HK-Icy and SENPAI have similar metrics to those described in Vormberg et al., while the results obtained with NeuTube and NeuroGPS differ. Ilastik achieved a lower SN mode (5), although results are close to those of HK-Icy and SENPAI. In addition, SENPAI outperforms the other algorithms matching the steepest decay for the normalized number of segments (slope = −1.23) found for Purkinje cells in the reference work and presenting smaller deviations in the curves of subtree size and total dendritic length.

### Comparison with other algorithms in the segmentation of super-resolution datasets

In addition to the confocal 40x datasets, we also applied SENPAI to L7GFP cleared mouse cerebellum slices imaged with a LEICA SP8 STED 3DX equipped with a 93x objective (see Methods), again comparing its performance with state-of-the-art tools (HK-Icy and the automatic segmentation algorithm Ilastik[53]) and ground-truth manual segmentation (obtained via the ManSegTool[60]). The resulting segmentations of five dendritic branches in the STED dataset are reported in Fig. 6.

Further analyses were performed to compare the branch areas and volumes obtained from the automatic tools with those obtained with the manual segmentation. We observed consistent differences between SENPAI and HK-Icy and Ilastik. Specifically, while Ilastik delivers volumes (Fig. 6B, left) that are significantly larger than those from the manual segmentations (volume normalized with respect to the ground-truth, median ± median absolute deviation (MAD) 1.91 ± 1.08) along with consistently larger area estimates, (area normalized with respect to the ground-truth, median±MAD 1.45 ± 0.42), SENPAI and HK-Icy's outcomes are very close to ManSegTool's (SENPAI area median±MAD 0.88 ± 0.30, SENPAI volume median±MAD 0.86 ± 0.49, HK-Icy area median±MAD 0.83 ± 1.00, HK-Icy volume median±MAD 0.75 ± 1.13). Particularly relevant is the analysis based on the Dice coefficient[61] (Fig. 6C), which quantifies the similarity between shapes (ideal = 1). Considering the whole dendritic branches, SENPAI achieves the highest Dice values for all the five of them, while Ilastik performs worst for all dendritic branches but one (SENPAI Dice median±MAD 0.81 ± 0.02, HK-Icy Dice median±MAD 0.78 ± 0.11,

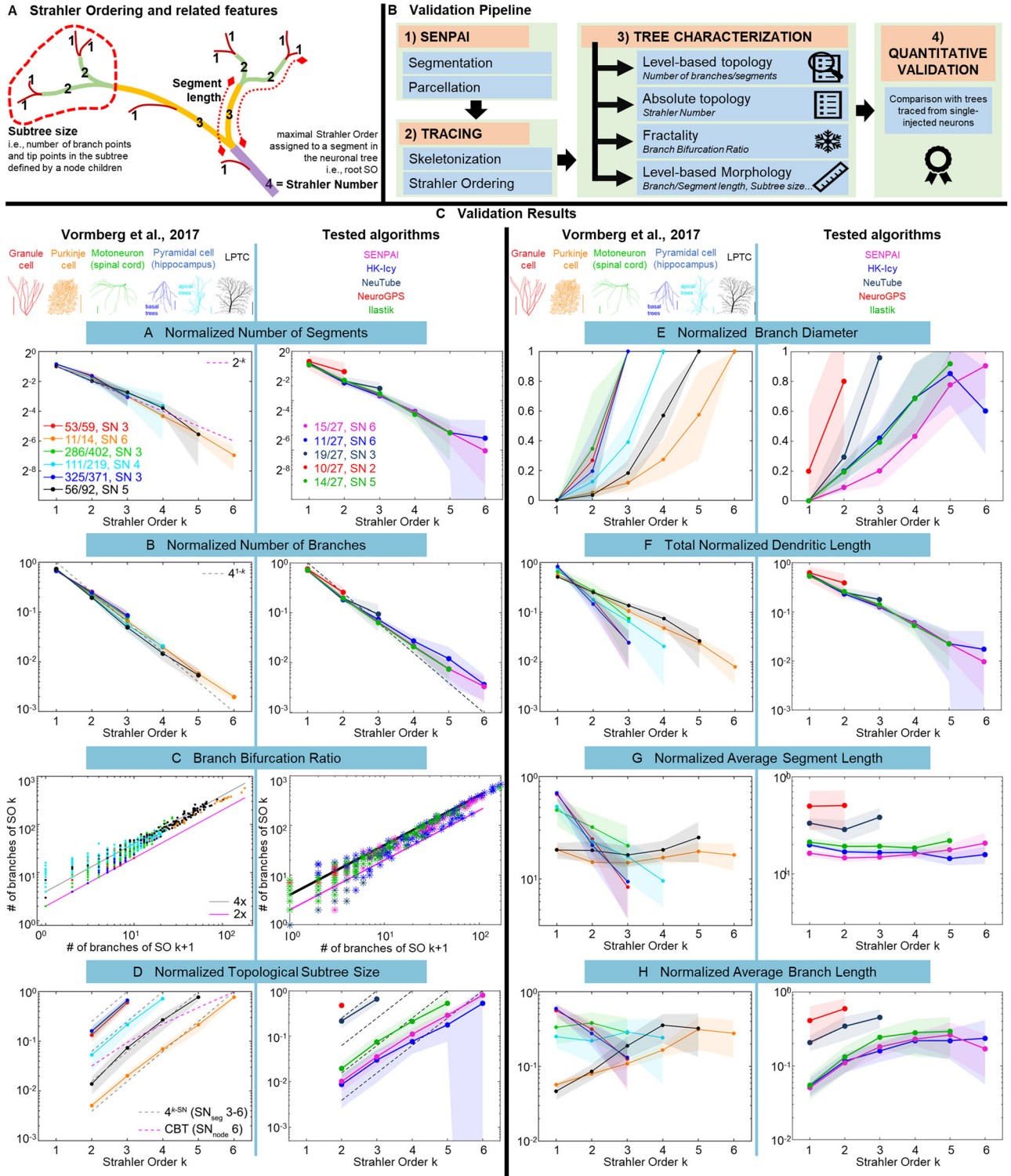

**Fig. 5 | Comparison against documented information on Purkinje cell morphometrics, based on Strahler Ordering. A** Schematization of SO and of the meaning of some of the extracted features; **B** Schematization of the validation pipeline; **C** Graphical comparison of the SO of Purkinje cells found in the literature and measured on the neurons segmented with SENPAI and the state-of-the-art tools. The parameters based on the SO reported by Vormberg et al. on six types of neuronal cells (left, edited from Vormberg et al.) and computed on 27 Purkinje neurons segmented with SENPAI (magenta), HK-Icy (blue), Ilastik (green), NeuroGPS (red) and NeuTube (gray) from 40x images (right). It should be noted that the parameters, with the exception of the Branch Bifurcation Ratio, were computed only for neurons whose SN was equal to their mode, in line with Vormberg et al. Source data are provided as a Source Data file.

Ilastik Dice median±MAD 0.68 ± 0.05). We performed the same analysis with a focus on spines, excluding those voxels that were labeled as belonging to the dendrite by manual segmentation. To avoid bias introduced by oversegmentation of the dendrite, we repeated the analysis recursively dilating the dendrite mask. All points are reported in Fig. 6C (bottom). SENPAI performs best again and is matched only for dendrite Ne by HK-Icy (for the first iteration, SENPAI Dice median±MAD 0.64 ± 0.07, HK-Icy Dice median±MAD

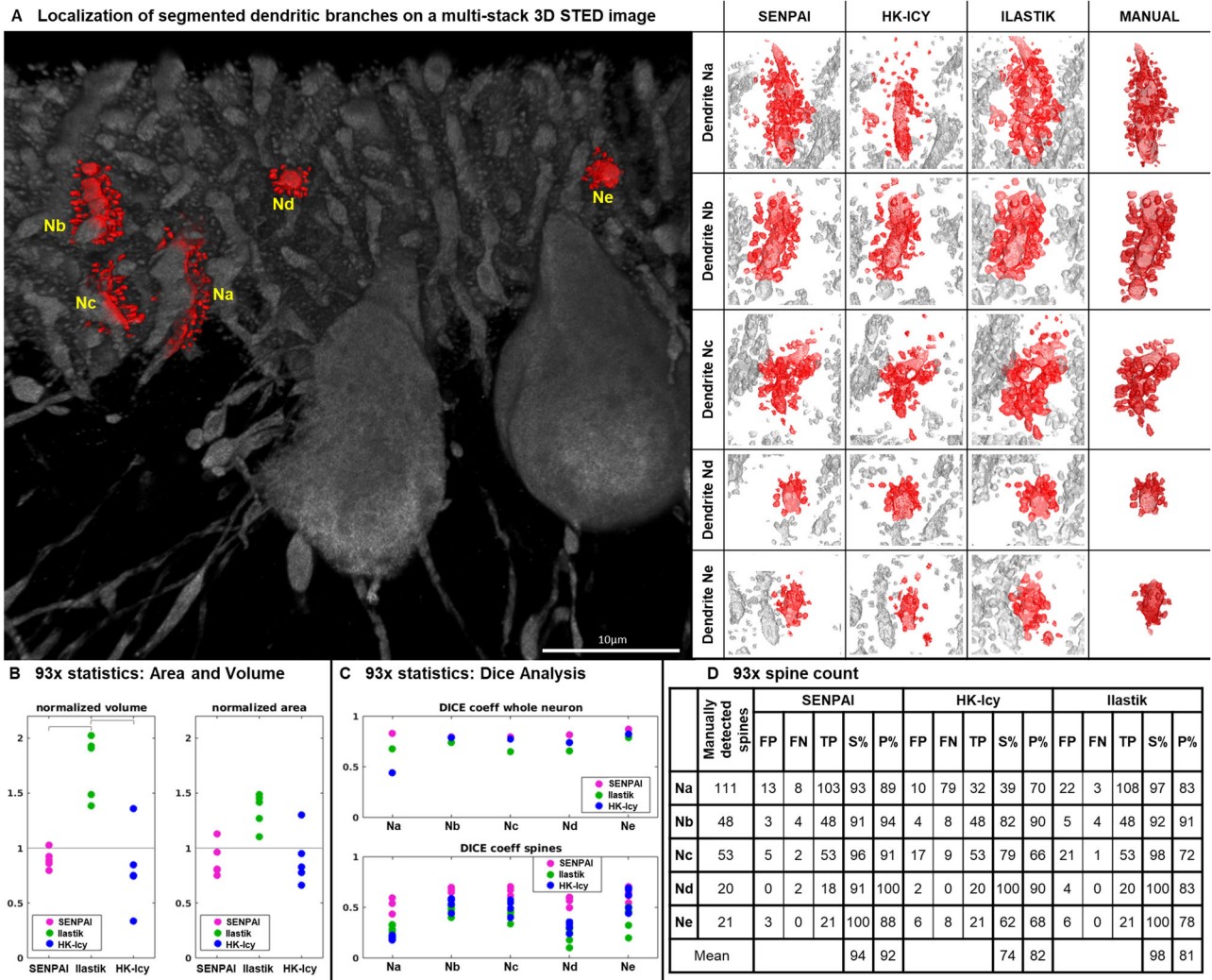

**Fig. 6 | Segmentations of 5 dendritic branches—Na to Ne—and their localization within a 3D multistack 93x STED dataset. A** Left: the dendritic branches Na to Ne contained within a single stack and employed here for the validation of SENPAI are highlighted in red. Right: we compare the segmentations obtained with the state-of-the-art algorithms. The segmentations were performed with SENPAI, HK-Icy, Ilastik and manual segmentation by ManSegTool. Algorithm segmentations of the whole image are underlaid in gray, while the parcellation outcomes are highlighted in red. **B** Quantitative comparison for the segmentations of the 5 dendritic branches; volume and area obtained with SENPAI, Ilastik and HK-Icy. The values are normalized with respect to manual segmentations. HK-Icy and Ilastik were integrated with the SENPAI parcellation step to, respectively, assign spines to the dendrite and separate touching dendrites from each other (volume result showed significant differences; Friedman $p = 0.015$; post-hoc with Conover's test highlighted Ilastik

difference to both SENPAI and HK-Icy, $p < 0.01$, highlighted with gray bars). **C** Quantitative comparison of the segmentations of the dendritic branches; the Dice coefficient was used to compare the segmentations obtained with the algorithms against the manual segmentation, considering the whole neuron (top) and the only spines (bottom). An ideal algorithm would give a Dice coefficient of 1. The analysis on the spines was conducted by masking out the manual segmentation of the dendrite (both whole neuron and average spine Dice coefficients from SENPAI were different to both Ilastik and HK-Icy; Friedman tests $p = 0.015$; post-hoc with Conover's tests, $p < 0.01$). **D** Table summarizing algorithm performance in terms of spine detection. Spines were counted in the manual segmentation, then we defined False Positives (FP) and False Negatives (FN), True Positives (TP), the Sensitivity (S% = % of TP over TP + FN) and the Precision (P% = % of TP over TP + FP) for SENPAI, HK-Icy and Ilastik. Source data are provided as a Source Data file.

$0.50 \pm 0.14$, Ilastik Dice median±MAD $0.43 \pm 0.10$). Merging the information in Fig. 6C, Ilastik clearly oversegments while HK-Icy undersegments.

Spine-by-spine reconstructions obtained with SENPAI, Ilastik and HK-Icy with the manual segmentation were also compared and reported in Fig. 6D. For each of the three algorithms we counted the number of true positives, false positives and false negatives and estimated specificity and precision in spine detection. Due to higher false negatives, SENPAI gives slightly lower specificity with respect to Ilastik, but performs best in terms of precision. On average, SENPAI is better than HK-Icy on all topological and morphological metrics and performs better than Ilastik in terms of morphological reconstruction and number of false positives.

### Generalization to other neuron types and imaging modalities

To support the generalizability of the SENPAI pipeline, we tested the algorithm on multiple neuron types from different species (rat, mouse and human) and in different conditions (tissues and cultured cells, in uncleared samples and acquired with different imaging modalities—see Supplementary Information). The neuron types we segmented include human pyramidal neurons, mouse and rat pyramidal neurons, human cultured excitatory neurons, mouse cultured excitatory neurons, and mouse retinal neurons. The imaging modalities, other than confocal and confocal STED, include two-photon (2P) microscopy. We report both quantitative and qualitative evaluations, that attest to the generalizability of our reconstruction algorithm across different imaging modalities and neuron types.

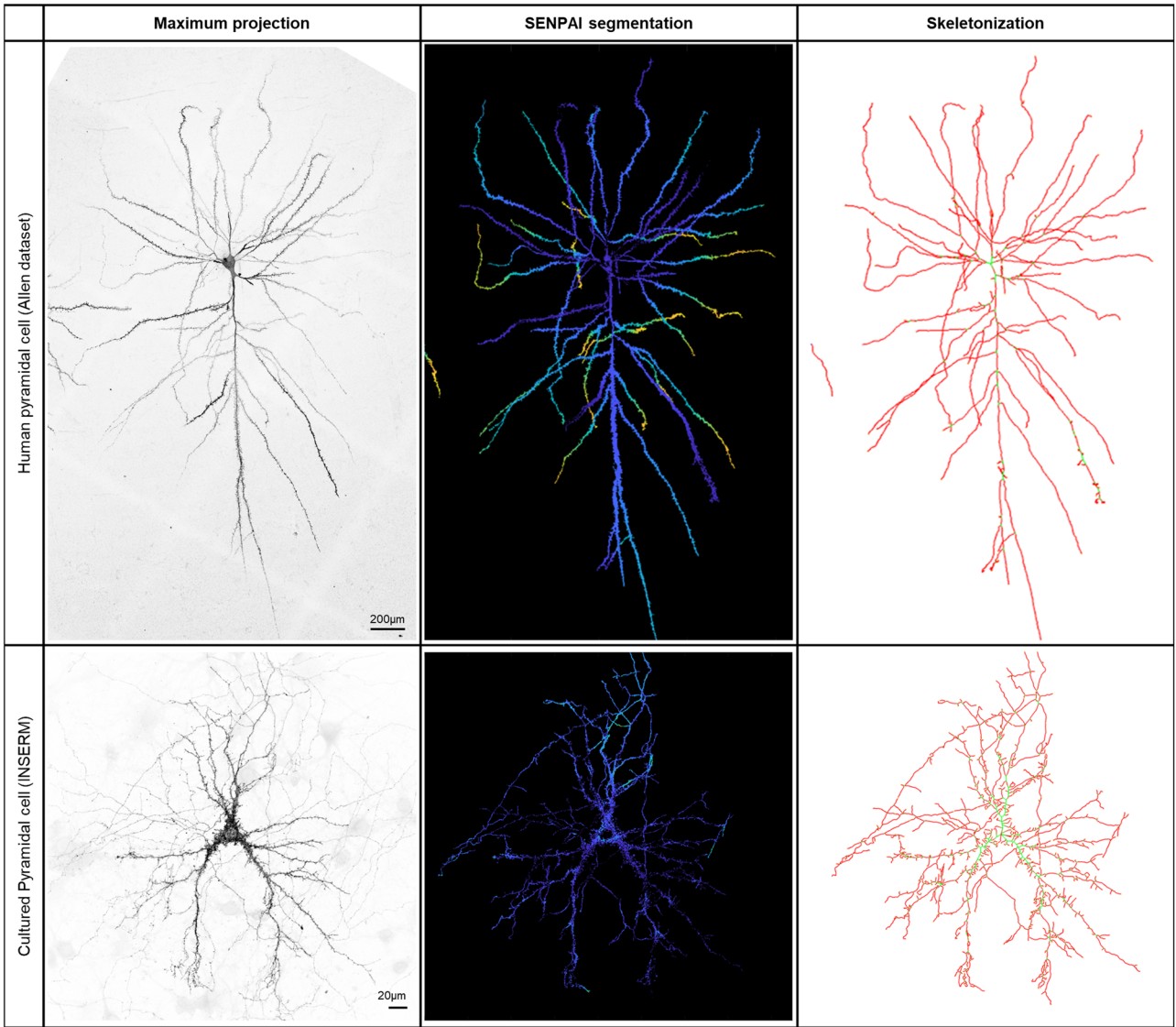

**Fig. 7 | Exemplary results obtained on non-clarified samples.** For both images we report the maximum projection on the left, the segmentation obtained with SENPAI on the middle (depth color-coded, cold colors indicate deeper planes), and the skeletonization of the segmentation on the right, as obtained using the NeuTube software. **Above** Exemplary dataset (m16_cing_1_9_cropped_neurona.v3dpbd, human pyramidal cell labeled with Lucifer Yellow and acquired through confocal microscopy, resolution 0.24 μm x 0.24 μm x 0.42 μm) from Benavides-Piccione et al.[72] available from the BigNeuron gold166 standard[63]. **Below** 3D stack of cultured rat hippocampal pyramidal cells (pixel size 91.41 nm x 91.41 nm x 280 nm). Further tests on non-clarified samples are reported in the Supplementary Information.

Among the tested datasets of native (uncleared) tissue tested with SENPAI, 34 belong to the BigNeuron GOLD166 dataset. On the BigNeuron bench-testing platform[44,62–65], among 35 of the most popular state-of-the-art algorithms, SENPAI scored in the top-three for five metrics over eight, and in the top-eleven for the remaining three metrics (see Supplementary Note 10 in the Supplementary Information). An exemplary result is reported in Fig. 7, upper panels.

A further uncleared image tested is the Neuro-GPS test dataset made available by the authors of Neuro-GPS (see Supplementary Note 11 in the Supplementary Information). Additionally, at the INSERM facilities, four more datasets were produced with rat hippocampal neurons: two datasets including a whole labeled neuron in the field of view, and two higher-resolution datasets from confocal and 3D-STED imaging the same dendritic branch with its spines. The results obtained from these four datasets are presented in Supplementary Notes 12 and 13, and Supplementary Figs. 18 and 19 in the Supplementary Information. Moreover, an exemplary result is reported in Fig. 7, lower panels.

## Discussion

Mapping the structural layout of the brain at different scales is a major challenge in neuroscience[66]. At the cellular level, deciphering neuronal structure and connectivity in situ is particularly difficult because of limitations in imaging depth in vivo. Consequently, imaging thick tissue samples containing densely packed cells requires considerable manipulation both in vivo and ex vivo. Once images are acquired, a further challenge is to analyse and quantify neuronal structure and morphology with high fidelity and reproducibility. To address these challenges, we have developed a framework for clearing and staining thick brain slices which can be imaged using confocal, 2P or super-resolution microscopy.

At the spine level, super-resolution STED combined with FRAP allowed to assess spines morphogenesis both in mice live slices and in vivo[67–70]. Single cell micro-injection and confocal microscopy recently provided a useful tool for spine classification by allowing to analyze a huge database of >7000 individually 3D reconstructed dendritic spines from human pyramidal cells[71,72]. Here, we propose to

complete this panorama with a pipeline that leaves single-cell injection behind and enables the structural analysis of deep tissue images.

The pipeline is based on SENPAI, a two-step segmentation and parcellation tool which returns morphologically faithful segmentations of neurons and their branches at micron and sub-micron scales, preserving surface details as well as multi-scale structure complexity, providing enhanced robustness to signal inhomogeneity even across thick samples. It exploits image intensity and its second order derivatives in space that were previously proposed for edge detection and structure modeling[73,74]. This second order derivatives analysis allows SENPAI to highlight surfaces enclosing higher signal regions, independent of the cross-sectional or intra-slice signal changes. It also optimizes data information usage by merging local segmentation with image parcellation based on global image intensity distribution. As such, the pipeline allows characterizing neuronal arborization, dendritic branch diameters or local spine density more accurately and faithfully than current algorithms and image processing tools (see Supplementary Figs. 2–4). Because it is model model-free and acquisition method-independent, tissular level organization at low magnification and structural nanoscale morphometry of dendritic spines using super-resolution microscopy can be easily correlated without the need for methods such as single cell injection[66,73–75].

SENPAI's performance was compared with state-of-the-art tools, and, when available, manual segmentation. Here we show that it achieves excellent results on different imaging modalities, such as standard confocal and STED. Moreover, SENPAI provides a finer description of isolated neurons while minimizing user effort in tuning the algorithm parameters: only the selection of the 3D Gaussian filter size and the soma identification to run the parcellation step are required.

Comparing performance with current tools and algorithms using 40x confocal microscope datasets, SENPAI outperforms NeuroGPS and NeuTube as it returns isolated neurons with a significantly higher arborization (Figs. 3–5). Indeed, the semi-automatic tracing with NeuTube is strongly limited by user sensitivity to the thinnest low-intensity branches. On the other hand, NeuroGPS could not achieve dense tracings even when lowering the segmentation threshold to the minimum. This is confirmed by the validation based on SO (Horton-Strahler ordering), which highlights that the segmentations provided by SENPAI and HK-Icy are clearly more consistent with quantitative information on Purkinje neuron morphology available in literature (Fig. 5). NeuroGPS provided sparse tracing of dendritic branches, many of which were not connected to any soma. NeuTube captured the basic morphology of a neuron in its native arrangement only for a small subset of neurons and it achieved SNs >4. Although SENPAI and HK-Icy showed similar performance, segmented structure density was observed to have a strong z-direction dependency for HK-Icy, compared to SENPAI. Undoubtedly, such a dependency is intimately linked to non-homogeneous pixel intensity and image contrast across sections (see Supplementary Fig. 7) and it could be related to the fact that segmentation in HK-Icy is based on intensity values and thus does not exploit any topological information.

For the 93x datasets obtained with STED, the results—in terms of assigned spines to the correct dendritic branch—were compared with manual segmentations performed with ManSegTool. All the approaches tested accurately reconstruct the dendritic branch of interest (Fig. 6A). Ilastik clearly results in over-segmentation with respect to the manual ground truth, with false positive assignments (Fig. 6A and D) and over-estimated areas and volumes both at the whole branch and at the spine level (Fig. 6B), while SENPAI correctly segments and assigns the spines in 5 out of 6 of the cases investigated (Fig. 6A, C) and provides estimates for area and volume in line with the manual ground-truth (Fig. 6B).

As far as neuronal tree reconstruction is concerned, most of the state-of-the-art tools for revealing neuronal arborization or dendritic

spines require a further step for reliably comparing their outcomes with the ones obtained with SENPAI. For instance, HK-Icy deals with a considerable number of parameters to be tuned for reaching a reasonable segmentation, but even tuning the parameters for maximizing the segmentation density without losing structural complexity (details on the parametric optimization reported in Supplementary Fig. 7), the algorithm returns a unique object including all the neurons that should be separated. This is also true for spine detection on 93x datasets, indeed, neither HK-Icy nor Ilastik provide any assignation criteria for the spines disconnected to the neuron body. Therefore, we observed that a parcellation step is needed to overcome the limitations of these state-of-the-art tools. This proves the need, particularly in the specific context of dense packed scenarios, typical of mammalian brain tissues, for a built-in codified criterion for the assignment and separation of objects of interest at different scales, be they branches or spines. We provide this versatile feature in SENPAI.

Although SENPAI was shown to be robust in the segmentation of single-cell scenarios, with different cell morphologies and imaging modalities, its unique contribution to the scientific community is the ability to deal with dense scenarios from cleared samples, a context that is particularly challenging for existing algorithms. Moreover, the SENPAI pipeline allows for reconstructing large groups of neurons down to the single spine level with a single, unified approach across different scenarios. This result is achieved thanks to the upgrade of the K-means clustering approach using the topological information at different scales, along with proper class selection criteria for neuronal structure identification. Another relevant feature is the development and integration with a parcellation step, based on watershed transform of the intensity image. This allows successfully isolating single neurons in different scenarios as well as performing an automatic assignment of spines to the relevant dendrite. Indeed, increasing magnification is often unfeasible, and it comes at the cost of shrinking the field of view. In these cases, spine neck segmentation is impaired by resolution limits. Using the watershed transform on the intensity image, we provide an efficient parcellation step for the automatic assignment of spines. Spine assignment is a crucial step preliminary to neck reconstruction. The latter can be achieved either using model-based[75–77] or data-driven[78,79] approaches.

Our results are also promising in the context of the segmentation of confocal imaging from brain samples. We report here the complex arborizations of Purkinje cells in their native arrangement such as those in Fig. 1. This scenario was chosen as a proof of concept since Purkinje cells are known to be highly densely arranged within the murine cerebellum. These reconstructions are unique since they provide fine segmentation of several neighboring cells with a very dense arborization. SENPAI provides a faithful panorama of the whole forest, where other solutions usually only provide segmentation of single cells isolated by fluorescent reporter injection. Thanks to its capability of imaging and segmenting complex dendritic arbors, the pipeline here proposed is a powerful tool that can be directly applied to transgenic reporter mouse lines. Indeed, in contrast to sparse injections for single cell imaging which are applied to thin live slices, SENPAI can be directly used on live or fixed thick blocks of tissue to quickly segment genetically-labeled neighboring cells.

Furthermore, we demonstrate the ability of the SENPAI pipeline to reconcile low magnification topographical maps from various brain regions (using either confocal or lightsheet) with super-resolution data giving access to the nanoscale level of the synapse. We also emphasize the relevance of the quality assessment of segmentations from ex vivo samples by using accurate morphological descriptors presented in literature. This validation paradigm might overcome the limitation of manual segmentations on dense images as gold standard for neuron segmentation. In fact, our results suggest that manual approaches cannot always be considered the ground truth since they may result in poor segmentations especially in large volume acquisitions whose

resolution is comparable to the size of crucial details of the image (e.g., thin neuronal branches). To this end, we propose an alternative method for evaluating automated segmentation algorithms through comparison with available knowledge on cell morphometrics (e.g., ramification density of Purkinje cells).

SENPAI was successfully tested on samples prepared using different methods (non-cleared tissue, cells in culture) acquired with multiple acquisition modalities, such as confocal, 3D STED and 2P, and different neuron types. It should be noted that several other imaging and labeling modalities that are specific to some parts of the neuron, e.g., membright for the membranes[25] or phalloidin for spine heads[80] are used by researchers. In all the cases where local inhomogeneities characterize neuronal structures, e.g., for non-uniform distribution of staining across the cell, SENPAI might show a suboptimal behavior. Specifically, the use of spatial derivatives, that are sensitive to image intensity changes, might lead to hollow 3D segmentations or poorly connected reconstruction of the dendrites. In some cases, such as those presented in the Supplementary Information, the 3D spatial smoothing preprocessing step resulted in increased neuronal structure homogeneity and consequently accentuated the behavior of second order derivatives. Further preprocessing steps might thus be needed to improve the generalizability of SENPAI to specific staining and imaging modalities.

It is worth noting that deep learning is now emerging as a potential solution to the complex problem of neuronal reconstruction from microscopy images. Yet, the time needed for training deep-learning models, as well as the fact that they heavily depend on the heterogeneity of training data, still limits their application to a variety of problems, such as single neuron segmentation. In addition, algorithms available in the state-of-the-art still struggle when dealing with separating multiple neurons within an image block[37]. Moreover, they do not easily allow performing semi-supervised tasks to provide control over under- and over-segmentation. As far as we know, no deep learning solution is yet able to segment microscopy images acquired using different modalities. Nonetheless, several groups are working on this (e.g., ebrains.eu) and likely future efforts will combine automatic algorithms like SENPAI and deep learning analysis to improve 3D neuronal segmentation.

Other future developments will include the application of SENPAI to cleared samples from other brain regions, to fully evaluate the potential of the proposed framework. We will also explore the possibility of estimating the path of spines necks even when the algorithm is not able to connect the spine with the main neuronal body and to provide the user with the possibility of characterizing spine population in terms of density and morphology.

In conclusion, we report a two-step segmentation and parcellation method –SENPAI- which can be combined with brain tissue preparation and labeling to isolate neurons from densely packed neighborhoods and return morphologically faithful segmentations of the cells and their branches at micron and sub-micron scales. The pipeline preserves surface details as well as multi-scale structural complexity, providing enhanced robustness to signal inhomogeneity even across thick samples. We show that SENPAI is superior in performance and robustness with respect to other segmentation algorithms and can be used across scales for different imaging modalities and different types of neuron or brain tissue samples. SENPAI can be exploited as a standalone, open-source pipeline to support the development of connectome maps and improve our understanding of the brain's structure and consequently its functional behavior.

## Methods

### Labeling and imaging of brain slices

We used the L7GFP mouse line, expressing cytosolic EGFP within dendrites, axons and soma of Purkinje cells under the L7 promoter. Mice were housed in rooms maintained at $22 \pm 2$ C, 65% humidity, and artificial light between 08:00 a.m. and 08:00 p.m. Male 3 month-old

mice were anesthetized (7% chloral hydrate) and intracardiacally perfused with 20 mL of ice-cold Phosphate Buffered Saline (PBS 1X, Sigma-Aldrich, Milan, Italy) followed by 20 mL of ice-cold hydrogel solution (4% acrylamide, 0.05% bis-acrylamide (Biorad Lab Inc., California, USA)), 4% formaldehyde (PFA, Sigma-Aldrich) and 0.25% VA-044 thermally triggered initiator (Wako Chemicals, Neuss, Germany). Brains were extracted and immersed in hydrogel solution for 3 days in the dark at 4 °C. Samples were placed under a vacuum for 10 min for bubble removal and to facilitate acrylamide polymerization and incubated at 37 °C overnight. The cerebella were then cut in slices of 0.5–1 mm using a Leica VT1200S vibratome. Slices were then incubated in two sequential baths of clarity solution (200 mM Boric Acid (Farmitalia Carlo Erba spa, Italy) and 4% Sodium Dodecyl Sulfate (SDS, Sigma-Aldrich) pH 8.5) at 37 °C for 3 and 5 days respectively. After clarification, the slices were washed in PBS, incubated in blocking solution in an Eppendorf tube on a wheel at room temperature (PBS 0.2% triton 0.25% cold water fish gelatin 3% donkey serum) for 1 h and then incubated in primary antibody solution for at least 2 weeks on a wheel at 4 degrees. Antibody solution was composed of washing buffer (PBS supplemented with 0.2 % triton−0.12% cold water fish gelatin (SIGMA G7765)−0.5% donkey serum (Jackson ImmunoResearch) and chicken anti GFP from Aves labs diluted to 1/1000. After 3 washes in washing buffer (1 h each) at RT, slices were incubated for at least 1 week at 4 °C under gentle agitation in washing solution supplemented with Donkey anti Chicken antibody coupled to Alexa 594 diluted at 1/500 (Thermofisher). Slices were then washed 3 times and mounted on a glass slide. To build horizontal calibrated spacers, PVC cover binders (300 microns thick from Fellowes) were punched with a hammer drill so that they remained very flat and horizontal and could surround the cleared slices. Several spacers could be stacked depending on slice thickness to adjust the height of the mounting. The spacer well was then filled with prolong gold and covered with a 0.17 mm (super-resolution #1.5) Menzel Gläser glass coverslip. Spacers and coverslip were sealed to the slide using Twinsil Picodent bi-component silicone.

Confocal and STED images were acquired with a confocal laser scanning microscope LEICA SP8 STED 3DX equipped with a 20 x /0,75NA and 93 × /1.3 NA glycerol immersion objective or a 40x NA1.3 oil objective and with 3 hybrid detectors (HyDs). The specimens were excited with a pulsed white-light laser (598 nm) and depleted with a pulsed 775 nm depletion laser to acquire nanoscale imaging (fluorescence beyond 600 nm was detected using SMD HyD detector, dedicated bandwidths are indicated in the metadata on Zenodo; refer to the Data Availability section). Depletion laser power balance (between 2D and 3D) was adapted depending on transparency sample laser to optimize resolution and preserve signal in depth. Confocal images of rat hippocampal pyramidal cells in culture were taken with 40x NA 1.3 Leica oil objective with a pixel size of 91.41 nm x 91.41 nm x 280 nm, bandwidth 603–644 nm.

3D-STED was used with a 93x NA1.3 glycerol objective. Images were averaged 16 times in line and acquired with a magnification zoom >3. Excitation laser was adjusted to avoid any saturated pixels. Rat hippocampal pyramidal cells in culture were acquired with voxel size of 40 x 40 x 130 nm within a Z stack of 21 slices (bandwidth 598–631 nm). Cerebellum Purkinje cells were acquired with voxel size of 62 x 62 x 60 nm (bandwidth 605–777 nm) within a Z stack of 319 (Fig. 1) or 35 slices (Fig. 2).

### SENPAI

SENPAI is an automated tool implemented in Matlab (The MathWorks-Inc., United States) and released as a freely available toolbox at: https://github.com/cauzzo-s5/SENPAI. It can be used to reconstruct neuronal morphology, from spine detection and morphometry up to whole neuron arborization, starting from 3D optical images from any type of microscope including confocal, multiphoton, STED, etc. It implements two fundamental steps: a segmentation step and a parcellation step.

**Segmentation step.** The first step to obtain faithful three-dimensional neuron reconstructions is to isolate neurons from the background. SENPAI tackles this step through its core segmentation algorithm, a topology-informed K-means clustering. Briefly, the K-means portions n observations ($X_1$, ..., $X_n$) defined on a d-dimensional space into K clusters, by minimizing the within-cluster variances[81]. In SENPAI, the space is made of d = 4 features of interest, comprising the pixel intensity of the image and second order spatial derivatives computed along the three main cartesian axes, while the n observations are represented by the total number of pixels entering the K-means. Since K-means convergence may be affected by initial conditions, additional hyperparameters, such as the number of replicates and the maximum number of iterations to reach convergence are included (in SENPAI, replicates are set to 10, and for each replicate, a maximum of 1000 iterations is allowed). To reduce the computational load for the K-means, a subset of background pixels is excluded from the clustering. This is tackled by simple thresholding, although more sophisticated methods could be used[82]. Preprocessing steps might include pointwise transformations (e.g., logarithmic), to adjust the image dynamics. Moreover, since the K-means may be computationally expensive, it can also be run on smaller user-defined cropped datasets to limit memory usage. Each 3D structure defined by a high-intensity cluster on a low-intensity background is characterized by intensity gradients at the boundaries, coded in the second derivative by a positive peak at the external boundary and a negative peak at the internal one. A key feature of SENPAI's K-means clustering is the choice of the number of classes K, and the means by which such classes are interpreted and used for segmentation. The algorithm iteratively searches for the smallest K that allows to have (i) a class encoding positive values of the second order derivatives for each direction in the 3D space (representing the outer borders of the neuron structures), (ii) each direction (x, y, z) is encoded in a different class. Then, for a given number of classes K, SENPAI identifies neurons by sorting and selecting the clusters obtained. Sorting is based on average pixel intensity values of each cluster, from the lowest to the highest. Then, the clusters associated with the neuronal structures are given by all the clusters with the highest average pixel intensity values and with all the second order derivates whose values are negative (these clusters represent signal from the neuron body see Fig. 2B, C). The union of the selected classes gives the final segmentation.

An optional parameter is the size of the pre-filtering kernel, which represents one of the salient features of SENPAI and allows focusing on multiple levels of detail of the image. First, the dataset is filtered using a 3D Gaussian filter and derivatives are computed. After the pre-filter step, the K-means clustering is performed on the pixel intensity of the unsmoothed dataset, and second order spatial derivatives computed on the smoothed image. This pre-filtering step can also be performed twice in a row by using two smoothing levels. This option allows capturing structures at different scales, since it eliminates spurious and unwanted boundaries, e.g., those within thick branches or big somata. When multiple levels of smoothing are used, the resulting segmentations are merged to compose the final segmentation (Fig. 2D).

However, since inhomogeneities in pixel intensity within big neuronal structures (e.g., the soma) cannot all be smoothed and may be identified as boundaries—leading to false negatives-, a function for filling such holes is implemented within the segmentation. Similarly, spurious small clusters (<7 voxels, the smallest 3D symmetrical structure) detected from inhomogeneities in the background are eliminated.

**Parcellation step.** The second step of the algorithm aims at isolating single neurons from the foreground identified via the K-means clustering. To this end, SENPAI performs a parcellation procedure based on the watershed transform. This defines a catchment basin for each structure, which allows marking the regions/pixels that belong to a particular structure and those that do not pertain to it. The parcellation serves two purposes: when dealing with single neuron reconstruction: first it splits neurons that may have been merged in a single structure, while when dealing with spine segmentation, it groups together disconnected structures. Thus, the parcellation is performed through two different functions.

The catchment basins are produced for each marked core. The cores are defined automatically in the super-resolution case, by imposing a threshold on the size of the clusters composing the segmented image. While big clusters are labeled as cores, smaller clusters are labeled as spines to be assigned to such cores. On the other hand, for standard microscopy, cores correspond to somata. These can be marked by the user thanks to a GUI implemented in SENPAI.

Once the cores are defined, the original grayscale dataset is filtered with a 3-by-3-by-3 median filter, and its complement is obtained. A morphological reconstruction is applied on this latter, so that the regional intensity minima outside the cores are eliminated. A watershed transform produces one catchment basin for each remaining regional minimum.

The parcellation can be modified a posteriori, thanks to a routine to visually assess the quality of the segmentation neuron-by-neuron, and to set additional markers on spuriously assigned branches. Incorrect assignments are usually caused by the presence of branches of neurons whose soma lies outside the acquired dataset. Operationally, this is done thanks to a function within the SENPAI toolkit, implemented to correct the parcellation by adding to the mask of the somata some additional clusters. These allow the definition of separate catchment basins for wrongly assigned branches.

### Testing datasets

The algorithm was tested on datasets acquired with a Nikon A1 confocal microscope (40x objective, excitation length 457 nm, bandwidth 500–550 nm), and a LEICA SP8 STED 3DX (93x objective, pulsed white-light laser 598 nm, pulsed 775 nm depletion laser, bandwidth 605–777 nm). All image stacks were from Purkinje cells within murine cerebella, cleared as in Magliaro et al.[30]. We evaluated SENPAI's performance in distinguishing both cellular (i.e., neurons) and subcellular (i.e., spines) structures.

The first 40x confocal image is a 512-by-512-by-143 image containing 114 somas (Fig. 2A), while the second one is a 512-by-512-by-139 image containing 103 somas. SENPAI was used on these datasets with a single clustering instance, without Gaussian smoothing. Then, postprocessing routines including parcellation and pruning were exploited for isolating single cells.

The 93x STED dataset is a 1024-by-1024-by-35 image in which several sections of neurons can be observed (Fig. 2A). The segmentation with SENPAI was obtained merging two clustering instances with different levels of smoothness (i.e., setting the standard deviations of the 3D Gaussian filter to 0 and 3). Finally, we assigned the dendritic spines to their parent branch.

### Performance assessment

In the following, we report the methodology used to compare the performance of SENPAI with that of HK-Icy, NeuTube, NeuroGPS, Ilastik and manual segmentation. This section of the methodology is not part of the SENPAI pipeline as it was only intended to be used for its validation. First, we describe how the competing toolboxes were employed, then we provide details on how we extracted morphometrics and statistics to be used for benchmarking.

**40x confocal.** The comparisons on the single neuron segmentations obtained with SENPAI, HK-Icy and Ilastik, as well as the tracings obtained with NeuroGPS and NeuTube, were performed on 27 cells.

We provide here the key parameters that had to be set for segmenting those neurons on HK-Icy, and NeuroGPS. For the HK-Means

plugin, the optimal segmentation was achieved by setting the intensity classes to 6 for the confocal datasets The minimum object size was set to 9000 pixels while the maximum object size was set to 7800000 and 4600000 pixels respectively for the two confocal datasets. The minimum object intensity was set to 0. Further details on the methods and rationale are reported in the Supplementary Information (see Supplementary Fig. 7). The segmentation with Ilastik was obtained using two classes, corresponding to the background and the neurons, and employing the maximum number of features for classification. For NeuroGPS, we set the intensity threshold to the minimum to achieve a dense segmentation. Since NeuTube is a semi-automatic tracing tool, there are no parameters to be set.

**40x confocal: morphometric evaluations and statistical analyses**. The volumes and areas of the segmented neurons were computed with an ad hoc script in Matlab. Sholl Analysis was performed via Neuroanatomy[83], a Fiji[84] plugin, whereas the corresponding area under the curve (AUC) was obtained in Matlab.

To evaluate any significant differences of area-to-volume ratio among SENPAI, HK-Icy and Ilastik segmentations, we employed the two-tailed Friedman non-parametric test for paired data. Data obtained from NeuroGPS and NeuTube are not included in the analysis, since they produce tracings with insufficient details on neuron morphology. A two-tailed Friedman non-parametric test for paired data followed by a post-hoc analysis (alpha = 0.05) was used to compare AUCs.

**40x confocal: Skeletonization and Strahler analysis**. SENPAI is a segmentation algorithm which provides volumetric reconstructions of neurons in the form of binary 3D volumes. Since the validation metrics for neuron reconstruction, such as Strahler analysis, are usually designed for assessing the goodness of neuron tracings, i.e., reconstructions in the form of tree-like graphs, we provide a routine for the skeletonization of our segmentations.

Starting from the raw 3D image stack, the binary segmentation of one neuron and the markers of the somata are exploited to produce the skeletonization, in the shape of a Matlab tree structure and of a SWC-like matrix, which is the format usually used to describe a graph or tree as a list of nodes, with their coordinates, diameter, and parent node. After the reduction of the neuron volumetric segmentation to 1-pixel wide curved lines, an ad hoc function converts it into a graph object. For each node the function automatically attributes an identification index. Node diameters are defined based on the intensity of the original image at the node coordinates. The graph is converted to a tree object: cycles are identified and then removed by cutting them where the image intensity is lowest. Finally, the SWC matrix is built by storing each node's identification index, coordinates in space, diameter and the identification index of its parent node.

Starting from the binary segmentation of one neuron and the relative SWC matrix, a function was also implemented to produce the Strahler ordering. Specifically, the functions' output is a matrix storing the statistics computed on: numSegSO (number of segments per SO), numSegSOnorm (number of segments per SO, normalized by the total number of segments), segLAve (average segment length per SO), PSnum (coefficients of the linear fit for the normalized number of segments per SO), segDAve (average segment diameter per SO), TOTL (total length of segments), TopoSubLAve (topological subtree size per SO), numBrSO (number of branches per SO), numBrSOnorm (number of branches per SO, normalized by the total number of branches), brDAve (average branch diameter per SO), brLAve (average branch length per SO), PBnum (coefficients of the linear fit for the normalized number of branches per SO), normTotL (total dendritic length per SO). The definition of the Strahler order of the segments constituting the whole arborized structure is based on the idea of assigning auxiliary weights to the tree edges, in a way that the minimum path length from each node to the root node, rounded to the floor, will correspond to

the node's SO. The SO of a segment is computed as the mode of the SO of its nodes. The code used to compute the Strahler metrics is also available in the same public repository (https://github.com/cauzzo-s5/SENPAI).

**93x STED**. With HK-Icy, the 93x image was not pre-processed and the number of classes parameter was set to 7. The minimum and the maximum object size, in terms of pixels, were set to 50 and 50000 respectively, while the minimum object intensity was set to 0. The segmentation with Ilastik was obtained using two classes, corresponding to background and neuron, and employing the maximum number of features for classification.

Dendritic spines were manually mapped using the AFNI[85] GUI, which allows marking the centers of each spine detected in the original image with a sphere. For each tool (SENPAI, HK-Icy and Ilastik), the map is used to assess (1) the percentage of spines correctly identified; (2) the percentage of spines correctly identified within the catchment basin defined by the SENPAI parcellation step; (3) the percentage of spines correctly identified and univocally assigned by the algorithm to its parent dendritic branch. For HK-Icy and Ilastik, we consider a spine as assigned by the algorithm if (1) it belongs to the same 3D connected cluster of the dendritic branch, or (2) it is marked with the same label of the dendritic branch after the 3D parcellation.

We compared the area-to-volume ratios (normalized with respect to the ratio computed on manual segmentations) of the objects segmented with three different algorithms (i.e., SENPAI, HK-Icy and Ilastik), by means of a two-tailed Friedman non-parametric test for paired data, followed by a post-hoc multiple comparison analysis.

### Statistics and reproducibility
No statistical method was used to predetermine sample size. The number of neurons from clarified samples for which we reported results and statistics is in line with other similar works in literature (e.g., 13 Purkinje neurons in Vormberg et al.[35,47], a maximum of 36 neurons segmented by any algorithm for mouse/human neurons in the Gold166 dataset). The experiments were not randomized, and the investigators were not blinded to allocation during experiments because only wildtype genotypes were analyzed.

### Reporting summary
Further information on research design is available in the Nature Portfolio Reporting Summary linked to this article.

## Data availability
Raw microscopy images are made available on zenodo website https://zenodo.org/uploads/10805555, the DataSet[86] Digital Object Identifier being 10.5281/zenodo.10805555. The human/mouse subset from BigNeuron GOLD166 dataset[63] can be downloaded at http://web.bii.a-star.edu.sg/bigneuron/gold166.zip. NeuroGPS dataset was downloaded at https://sourceforge.net/projects/neurogps-tree/files/, as reported in the Code Availability Statement in Quan et al.[35]. Source data are provided with this paper. The data generated in this study and shown in Figs. 4–6 are provided in the Source Data file. The data generated in this study and shown in Supplementary Information Figs. 9–16 are provided in the Source Data file. Source data are provided with this paper.

## Code availability
Matlab code for the SENPAI algorithm is freely available at the following repository: https://github.com/cauzzo-s5/SENPAI.

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

## Acknowledgements

We would like to thank the NeurImag Imaging core Facility team (part of IPNP, Inserm U. 1266 and Université Paris Cité) and member of the national infrastructure France-BioImaging supported by the French National Research Agency (ANR-10-INBS-04) for their technical and scientific support. We thank the Leducq foundation for supporting the acquisition of the Leica SP8 Confocal/STED 3DX microscope. E.B., D.B. and P.N. were supported by the FLAG-ERA (grant Sensei by ANR-19-HBPR-0003 and MUR provvedimento n. 1637, 19/10/2020, CUP: I54I19002660006). A.L.C. was supported by the European Union—Next Generation EU, in the context of The National Recovery and Resilience Plan, Investment 1.5 Ecosystems of Innovation, Project Tuscany Health Ecosystem (THE), Spoke 3 "Advanced technologies, methods, materials and heath analytics " CUP: I53C22000780001. This work was funded by the French National Research Agency (ANR) ANR-19-CE16-0012 (to LD), and by FLAG-ERA (grant SENSEÏ by ANR-19-HBPR-0003 and MUR- CUP: I54I19002660006) to L.D. and N.V.

## Author contributions

S.C., C.M., L.D. and N.V. conceived and designed the study. S.C., E.B., D.B., P.N. A.L.C., A.A., C.M. and L.D. handled data curation. S.C., E.B., P.N., M.B., A.L.C., F.T. and L.D. carried out formal analysis. S.C., E.B., A.L.C. and N.V. designed the methodology for segmentation algorithm and statistical tools development. D.B., P.N., C.M. and L.D. designed the methodology for sample processing. S.C., E.B., D.B., P.N., M.B., A.L.C., C.M. and L.D. carried out the investigation. S.C., E.B. and A.L.C designed the software. S.C., E.B., P.N, M.B., A.L.C., F.T., L.D. and N.V. worked on validation. S.C., E.B., D.B., P.N. and M.B. worked on data and results visualization. S.C. and E.B. prepared the original draft. S.C., E.B., A.L.C., A.A., C.M., L.D. and N.V. edited and revised the manuscript. A.A., C.M., L.D. and N.V. managed resources and funding acquisition. C.M., L.D. and N.V. curated project administration. L.D. and N.V. supervised the study.

## Competing interests

The authors declare no competing interests.

## Ethics

This research complies with all relevant national and international ethical regulations. L7GFP were obtained for a UNIPI project (PRA_2016_56) from the Department of Translational Research, New Technologies in Medicine and Surgery of the University of Pisa (Italy). The study protocol was approved by the Animal Welfare Committee from University of Pisa, specifically Organismo Preposto al Benessere Animale, Unità Etica e tutela animale nella Ricerca, Lungarno Pacinotti 43/44, 56126, Pisa.
