## [Peer Review File · Nature Communications]

Reviewers' Comments:

Reviewer #1:

Remarks to the Author:

Cauzzo et al. present an image segmentation software based on Matlab that combines different existing image analysis and segmentation tools/methods with a k-means clustering method to improve image segmentation. The authors apply the segmentation tool to purkinje cells that have been imaged with confocal microscopy and STED microscopy. The segmentation of entire purkinje cells is quite impressive compared to the other tested and existing methods. The spine segmentation based on SENPAI compared with existing tools is not as convincing. The cited literature in the introduction and discussion could be more elaborated and complete. The Matlab code was not available under the provided link. A segmentation with SENPAI of the most frequent neuron in the brain, a pyramidal neuron, is not provided. Although the manuscript has some potential, it falls short at several points. E.g. it is not discussed how machine learning and AI-based methods might perform in comparison to SENPAI. To me the manuscript appears to be premature and at the current stage not suitable for Nature Communications.

Some points that need to be addressed are listed below:

Literature coverage about spines could be more elaborated and precise. Spires 2005 is about spine loss in a mouse model not human AD.

Same is true for literature on super-resolution in vivo and in vitro. Close to ground truth data is still considered to be electron microscopy. It is still outperforming any visible light based super-resolution technique in fixed tissue. This should be clearly mentioned.

The diffraction barrier at 200 nm should not be specified. It is depending on the wavelength and the NA of the used objective and can vary.

Please provide a reference for purkinje cells bearing 200.000 synapses.

I usually do not care about wording. However, the authors should check whether „clearing“ brain samples should be rather used than „clarified“. This sounds a bit awkward.

Methodologies like how brains were cleared should be moved to the Mat&Met Part of the manuscript. (L103-104)

Figure 1k-l. Has no scale bar. It is difficult to judge about resolution.

For correct spine detection Ilastik is obviously outperforming SENPAI.

Unclear, whether SENPAI works similarly on other neurons (e.g. pyramidal neurons; GABAergic neurons).

Other imaging modalities like 2P and 2P-STED image analysis with SENPAI would have been a great add.

L446: White laser excitation? What was the exact excitation wavelength? What was the emission wavelength of the used dye. What were the bp settings to detect the emission?

Under the given address the Matlab code is not deposited. It cannot be evaluated.

Reviewer #2:

Remarks to the Author:

The manuscript by Cauzzo and coworkers describes a novel algorithm (termed SENPAI) for

segmenting microscopic images into individual neurons, dendrites and dendritic spines. The authors show using an example dataset of Purkinje cells that their algorithm performs better than existing segmentation algorithms even with extremely densely packed neuronal assemblies, which represents an important advance compared to other (semi-)automated neuron segmentation tools. Furthermore, the segmentation of dendritic spines from STED microscopic images of Purkinje cells was shown to be nearly as successful as with a machine learning based image analysis tool (Ilastik). Therefore, the new algorithm could be very useful for reconstructing large groups of neurons down to the single spine level. While the results of the segmentation using SENPAI are promising and the manuscript is well written, it is important to show the applicability of the algorithm to different neuron types and there are some important methodological details missing.

Major Comments:

1. The major limitation of the performance assessment is that only one cell type, namely Purkinje cells, is used. The dendritic arbor of Purkinje cells is rather two-dimensional compared to for instance cortical pyramidal cells, and it would be important to show that the algorithm works on different types of neuronal morphologies. Furthermore, in order to fairly evaluate the performance of the SENPAI algorithm compared to the previously published neuron segmentation tools, it would be best to use an independent dataset.
2. If the Ilastik image segmentation tool could also be applied to the 40x confocal dataset, the results would be interesting to compare to those of the SENPAI algorithm. If Ilastik is not applicable to this dataset, this should be stated in the manuscript as it could represent a clear advantage of the SENPAI algorithm.
3. The description and illustration of the SENPAI workflow could be improved. The preprocessing steps (deconvolution, marking the somata) that were performed should be clearly stated, including the software used if applicable. The individual steps of the workflow shown in Figure 2 could be better illustrated by using the same source image throughout, and by including the final segmentation in panel D and the grayscale image used for the parcellation step in panel E.
4. The rationale for using Strahler order based branching statistics to validate the segmentation of the Purkinje cells should be stated more explicitly. As Vormberg et al. (2017) showed, there are two categories of Strahler order based branching parameters which differ in their universality: the topological parameters are the same for all binary trees, while the metric parameters differ based on neuron type and can therefore be meaningfully used to classify different neuron types. Therefore, the statement in lines 244-245, "each neuron type is described by a specific combination of these 9 parameters" is misleading. Also, the Methods contain no details of how the analyses shown in Figure 5 were done.
5. Judging from the images in Figure 6A, it appears that fewer spine necks were reconstructed with the SENPAI algorithm compared to the manual segmentation. Given the importance of the spine neck width in determining the electrical compartmentalization of the spine head (e.g., Tonnesen et al., Nat. Neurosci. 2014), it would be interesting to know whether the assignment of the spine necks could be improved.

Minor Comments:

1. Running the test script only produced 8 cells in the final visualization instead of the 10 cells shown in Figure 3. The indices of the two remaining neurons appear to be missing in the test_script file.
2. The manuscript doesn't contain any information about the hardware used to run the code. It would also be helpful to know how the algorithm scales with the image resolution/the number of stacks.
3. What is the rationale for using 6 classes in the k-means clustering? Were these determined empirically, and would the ideal number of classes potentially differ for different cell types or microscopic parameters?
4. The microscopic parameters used to generate the test datasets should be described in more detail. What were the acquisition parameters for the 40x confocal dataset? What was the step size used for the confocal and STED datasets? Was Nyquist sampling used?
5. Figure 5 could be improved by only displaying the Purkinje cell data from Vormberg et al., perhaps even in the same plot as the data obtained by the different segmentation algorithms.

Reviewer #3:

Remarks to the Author:

The authors discuss the interesting problem of neuron segmentation at cellular and sub-cellular scales in microscopy images of brain tissues. The topic is of possible interest for the journal reader. The paper is well structured. The used reference list is proper. The state of art in the field must be realized in a more critical way. The novelty of the proposed algorithm must be highlighted in comparison with similar algorithms. The general workflow of the proposed method must be described in more detail. In present form a reader can not reproduce the algorithms. The obtained advantages must be expressed using numerical performance measures. The Conclusions must be completed with these measures of advantages. The algorithm is used for the particular case of Purkinje cells. How about other type of cells? Which are the limitations or disadvantage of the method in other cases?

Color Legend:

Reviewer's comments

Answers to the reviewer's comments

Citations from the manuscript/supplementary material

Edited portions in the citations from the manuscript/ supplementary material

To all the reviewers: brief summary of major updates

We deeply thank all reviewers for their critical reading of the manuscript and their objective comments. We addressed all of them carefully, and here we summarize the main revisions to the manuscript:

- The introduction was extended with a detailed state of the art of automatic segmentation and tracing algorithms, as well as of spine segmentation.
- We tested the SENPAI algorithm on different cell types (human pyramidal neurons, human cultured excitatory neurons, mouse cultured excitatory neurons, and mouse retinal neurons, Figures S8-S11), also acquired exploiting another imaging method, i.e., two-photon imaging. We extended this list with rat hippocampal neurons in culture (glutamatergic excitatory) that were grown in the lab and acquired with the already-tested confocal and STED imaging. A summary of the datasets that were processed can be found in the table below. All the new included tests are now described and reported in the supplementary materials. To maintain coherence with the overall pipeline, in the manuscript we only provide details of the reconstructions performed on cleared tissue samples. All the other tests are described and reported in the supplementary materials. We modified the introduction to underline the fact that we tested the algorithm on images from freely available databases.
- As requested, we added the results obtained with Ilastik on the 40x images.
- On the 93x datasets, we provided a new analysis based on Dice's coefficient to compare the algorithms in terms of false positives and false negatives, and we provided further details on the performance obtained in spine reconstruction.
- We enriched SENPAI with new functions, performing the estimation of the optimal number of classes for clustering, skeletonization and extraction of the Strahler ordering of the reconstructions. We extended the discussion with considerations on the potential limitations of SENPAI, and on the state of the art of artificial intelligence in neuron reconstruction.
- We propose to add Eng. Federico Tozzi as an author for his valuable work in answering the reviewers' comments. Specifically, he analysed and compared the Bigneuron dataset as well as the new data from INSERM and he improved the characterization of SENPAI behaviour and supported the overall improvement of the paper.

Dataset	Acquisition modality	Independent (YES/NO)	Results shown in
Purkinje Cells, Mice – L7GFP, Cleared (Clarity)	Confocal (40x), STED (93x) Pixel size, etc	NO	Main Paper
Hippocampal Pyramidal Cells, Mice, Cultured	Confocal (40x,93x) STED (93x) culture in vitro	NO	Supplementary

BigNeuron Dataset (GOLD 166) Selected datasets:	Confocal/ 2 photon	YES	Supplementary
NeuroGPS Pyramidal Cells	Confocal	YES	Supplementary

Reviewer #1

Cauzzo et al. present an image segmentation software based on Matlab that combines different existing image analysis and segmentation tools/methods with a k-means clustering method to improve image segmentation. The authors apply the segmentation tool to purkinje cells that have been imaged with confocal microscopy and STED microscopy. The segmentation of entire purkinje cells is quite impressive compared to the other tested and existing methods. The spine segmentation based on SENPAI compared with existing tools is not as convincing. The cited literature in the introduction and discussion could be more elaborated and complete. The Matlab code was not available under the provided link. A segmentation with SENPAI of the most frequent neuron in the brain, a pyramidal neuron, is not provided. Although the manuscript has some potential, it falls short at several points. E.g. it is not discussed how machine learning and AI-based methods might perform in comparison to SENPAI. To me the manuscript appears to be premature and at the current stage not suitable for Nature Communications.

We thank the Reviewer for the comment. As we stated in the general introduction of this document, several changes were applied to the original manuscript. Here we sequentially answer the Reviewer's specific comments and suggestions.

The reviewer #1 states "The segmentation of entire purkinje cells is quite impressive compared to the other tested and existing methods. The spine segmentation based on SENPAI compared with existing tools is not as convincing"

We modified the description of the performance of spine segmentation task in the main text, highlighting pros and cons of SENPAI against existing state of the art tools. We discussed the difficulties in performing spine segmentation, and the quality of the results achieved by the other tested algorithms. Regarding spine segmentation, according to the referee comments, we realized that our statistics were not enough detailed to account for segmentation difference between SENPAI and Ilastik or Icy. We thus improved the description of spine segmentation results, including statistics about spine detection performance and specific morphometric measures of dendrite trunk and spines. These results are reported in Figure 6. In brief, we show better performance for SENPAI with respect to HK-Icy and Ilastik in describing the spine morphology while controlling False Positives. For details, we redirect the reviewer to our answer to his further comment below, "For correct spine detection Ilastik is obviously outperforming SENPAI.". We also added in the supplementary material the results

obtained on other datasets, namely pyramidal cells, acquired with both confocal and 3D-STED microscopy showing the performance of SENPAI in spine segmentation task.

Reviewer #1 states **“The cited literature in the introduction and discussion could be more elaborated and complete.”**

We added a new paragraph in the Introduction, detailing the state of the art of tracing and segmentation algorithms for neuronal reconstruction. We report below the edited part.

[...] while maintaining low laser power [Magliaro et al., 2016].

Parallel to the advances in neural tissue labeling and imaging at both the micro and nanoscale, automatic algorithms aiming at obtaining faithful three-dimensional (3D) reconstruction of single neurons have been extensively developed. Recent reviews on the topic [Liu et al., 2022; Magliaro et al., 2019] provide a categorization point for more than forty algorithms (see Table 1 in the supplementary materials), each one exploiting key aspects of image properties (e.g., signal-to-noise ratio, point-spread-function, image contrast) and/or neuron features (e.g., axon tubularity, soma sphericity, tree-like structure) for developing their own neuron reconstruction strategy. As such, their success is strongly influenced by both the acquisition modality and cell type and by the kind of features they rely on (e.g., local methods based on a progressive propagation of the tracing or segmentation from a reference point are particularly sensitive to noise or local inhomogeneity, while global methods based on SNR will fail at handling smooth changes in SNR across the image stack) to the point that developing general-purpose solutions is often impracticable. Some of these issues are partially solved by meta methods, i.e. modules that can be used on top of base tracing or segmentation algorithms to solve specific problems and improve the result (e.g., G-cut [Li et al., 2019] for post-hoc separation of interweaving neurons and UltraTracer [Peng et al., 2017] for increasing the scalability of tracing algorithms to large image stacks). It is worth highlighting that most of the algorithms for neuron segmentation from microscopy images proposed cannot isolate neurons acquired with different imaging modalities, and only few of them deal with densely packed cells [Callara et al., 2020; Li et al., 2019; Milligan et al., 2019; Quan et al., 2016].

Similar considerations apply to methods based on Artificial Intelligence (AI) approaches such as machine learning and deep learning (for a summary categorization see Tables 3 and 5 in [Chen et al., 2023]), successfully introduced also thanks to the availability of high quality and large training datasets [Ghahremani et al., 2021; Mazzamuto et al., 2018; Nourbakhsh et al., 2018]. For the same motivation, convolutional neural networks were recently introduced [Ghahremani et al., 2021; Mazzamuto et al., 2018; Nourbakhsh et al., 2018]. Indeed, AI approaches are rapidly emerging as a solution to the complex problem of neuronal reconstruction in microscopy images. Their direct application to previously unseen data (e.g., different acquisition modality, cell type, SNR levels) strongly depends on the training set (e.g., type of data, number of training examples), or ad-hoc solutions (e.g., transfer learning [von Chamier et al., 2021; Stuckner et al., 2022]), limiting their generalizability. Although considerable effort is now being made to achieve explainability in deep learning models [Liu and Xu, 2023], the black-box nature of deep-learning based algorithms makes it difficult for users to exercise control over common segmentation issues such as over- or under-segmentation. In addition, existing algorithms based on deep-learning currently struggle in dealing with separating multiple structures within an image block [Chen et al., 2023] and to retrieve information beyond the limits posed by the resolution of the image, i.e., for spine morphology or spine neck reconstruction. Finally, independent of the method used, manual correction and fine-tuning of the automatic results by experts are still required [Manubens-Gil et al., 2023].

The generalizability of neuron reconstruction algorithms' is even lower when dealing with images from densely packed neurons, in particular those obtained from cleared tissues [Magliaro et al., 2019], which represent a novel class of previously unseen neuroimaging data with unique features in terms of both image quality (e.g., enhanced signal intensity, SNR, CNR) and the cell density in a 3D arrangement [Callara et al., 2020; Magliaro et al., 2019; Pesce et al., 2022; Ueda et al., 2020]. Moreover, the outcomes of such algorithms and tools are compared with a manually segmented ground truth reconstruction, which is time consuming, and, particularly when dealing with dense-packed neurons, difficult to achieve and prone to human bias.

This paragraph was integrated in the Supplementary Materials with a Table (Table S1) summarizing the automatic segmentation and tracing algorithms cited by [Magliaro et al., 2019] and [Liu et al., 2022]. We report table S1 below:

Algorithm	Authors	Doi
ViterBrain	Athey et al. (2022)	10.1038/s42003-022-03320-0
RPCT	Bas and Erdogmus (2011)	10.1007/s12021-011-9105-2
SmartTracing	Chen et al. (2015)	10.1007/s40708-015-0018-y
Neural Circuit Tracer	Chothani et al. (2011)	10.1007/s12021-011-9121-2
Active learning of neuron morphology for accurate automated tracing of neurites	Gala et al. (2014)	10.3389/fnana.2014.00037
CAAT	Huang et al. (2021)	10.3389/fnana.2021.712842
ORION2	Jiménez et al. (2015)	10.1007/s12021-014-9256-z
ShuTu	Jin et al. (2019)	10.3389/fninf.2019.00068
SparseTracer	Li et al. (2017)	10.1371/journal.pone.0182184
Rivulet2	Liu et al. (2018c)	10.1109/TMI.2018.2833420
ORION	Losavio et al. (2008)	10.1152/jn.90627.2008
flNeuronTool	Ming et al. (2013)	10.1371/journal.pone.0084557
APP1	Peng et al. (2011)	10.1093/bioinformatics/btr237
UltraTracer	Peng et al. (2017)	10.1038/nmeth.4233
NeuroGPS-Tree	Quan et al. (2016)	10.1038/nmeth.3662
PHD	Radojević and Meijering (2017a)	10.1093/bioinformatics/btw751
PNR	Radojević and Meijering (2019)	10.1007/s12021-018-9407-8
NeuronStudio	Wearne et al. (2005)	10.1016/j.neuroscience.2005.05.053
Open-Curve Snake	Wang et al. (2011)	10.1007/s12021-011-9110-5
ENT	Wang et al. (2017)	10.1007/s12021-017-9325-1
DiMorSC	Wang et al. (2018)	10.1101/321489
MOST	Wu et al. (2014)	10.1016/j.neuroimage.2013.10.036
APP2	Xiao and Peng (2013)	10.1093/bioinformatics/btt170
SimpleTracing	Yang et al. (2013)	10.1186/1471-2105-14-93
FMST	Yang et al. (2019)	10.1007/s12021-018-9392-y
MDL constrained 3-D grayscale skeletonization	Yuan et al. (2009)	10.1007/s12021-009-9057-y
neuTube	Zhao et al. (2011)	10.1007/s12021-011-9120-3
Neuron Crawler	Zhou et al. (2015b)	10.1109/ISBI.2015.7164009
TReMAP	Zhou et al. (2016)	10.1007/s12021-015-9278-1
neurolucida	Glaser et al. (1990)	10.1016/0895-6111(90)90105-k
mansegtool	Magliaro et al. (2017)	10.3389/fninf.2017.00036
tree2tree	Basu et al. (2013)	10.1109/TITB.2012.2209670
trees	Cuntz et al. (2010)	10.1371/journal.pcbi.1000877
g-cut	Li et al. (2019)	10.1038/s41467-019-09515-0

"It is not discussed how machine learning and AI-based methods might perform in comparison to SENPAI"

We added a brief section in the discussions to review the machine learning and AI-based solutions for neuron segmentation.

[...] through comparison with available knowledge on cell morphometrics (e.g., ramification density of Purkinje cells).

It is worth noting that deep learning is now emerging as a potential solution to the complex problem of neuronal reconstruction from microscopy images. Yet, the time needed for training deep-learning models, as well as the fact that they heavily depend on the heterogeneity of training data, still limits their application to a variety of problems, such as single neuron segmentation. In addition, algorithms available in the state of art still struggle in dealing with separating multiple neurons within an image block [Chen et al., 2023]. Moreover, they do not easily allow to perform semi-supervised tasks such as having control over under- and over-segmentation. Up to now, we are not aware of any deep learning solution able to segment several microscopy modality images. However, we can hope that in the following years the various consortium and infrastructure like eBrains (<https://www.ebrains.eu/>) will greatly facilitate those grouping of data, thus enabling card reshuffling concerning deep learning solution. We believe that automatic segmentation algorithms like SENPAI could help to build the huge database of manually curated annotated images needed to improve deep learning versatility. Furthermore, we suggest that the combination of both manual segmentation, automatic segmentation using algorithms like SENPAI and deep learning analysis will probably be the winning combo to improve 3D neuronal segmentation.

Reviewer # 1 says ***"The Matlab code was not available under the provided link."***

We are sorry to hear that it was not possible for the reviewer to run the Matlab code. As another reviewer was able to run it, we wonder whether this could be an issue related to Mathworks account or maybe to a firewall. We asked the editor to provide, along with the revised manuscript and this letter, a zipped folder containing all the code. We hope that this will solve the problem.

A segmentation with SENPAI of the most frequent neuron in the brain, a pyramidal neuron, is not provided.

We provide now new segmentations from independent datasets including images of pyramidal neurons. For the details, we redirect the reviewer to our answer to his comment below, "Unclear, whether SENPAI works similarly on other neurons (e.g. pyramidal neurons; GABAergic neurons). Other imaging modalities like 2P and 2P-STED image analysis with SENPAI would have been a great add."

Some points that need to be addressed are listed below:

Literature coverage about spines could be more elaborated and precise. Spires 2005 is about spine loss in a mouse model not human AD.

Text has been rephrased to be more precise, key concept on dendritic spines and recent reviews and book have been added for clarity.

Dendritic spines are tiny membranous protrusions on dendritic shaft that can be contacted by glutamatergic axon to form an excitatory glutamatergic synapse. Spine morphology have been early described by Ramon Y Cajal in 1888 using Golgi Technique [García-López et al., 2007]. Their 3D morphology has been described over the years using the gold standard 3D electron microscopy (see for review [Harris, 2020; Rasia-Filho et al., 2023]). Dendritic spines can harbor various morphology, from stubby spines (short bulged of the membrane) to mushroom shape (with a restriction neck and a bulbous head) [Harris et al., 1992]. More recently, based on ultrastructural analysis of mouse neocortical dendritic spine necks, the existence of subtypes of spines has been questioned, propending for a continuum of spine morphologies [Ofer et al., 2021]. Dendritic spines, that receive electric inputs, are both cellularly and electrically locally regulated due to their neck geometry and resistance [Citri and Malenka, 2008; Noguchi et al., 2005]. Spine shape and density are linked to synaptic function, such as learning, memory and motivation [Knott and Holtmaat, 2008; Sala and Segal, 2014] and changes in synaptic activity are associated with alterations in spine shape, size and number [Bourne and Harris, 2007; Kasai et al., 2010]. Structural alterations of spiny synapses are also found in the pathogenesis of major neurological disorders (ASD, schizophrenia and Alzheimer's disease) [Glausier and Lewis, 2013; Penzes et al., 2011; Spires, 2005]. Reduction in spine density and synaptic loss is found in patients with Alzheimer's disease and MCI [Scheff et al., 2006; Serrano-Pozo et al., 2011; Spires, 2005], suggesting its role in brain dysfunctions [Penzes et al., 2011; Scheff et al., 2006].

Same is true for literature on super-resolution in vivo and in vitro. Close to ground truth data is still considered to be electron microscopy. It is still outperforming any visible light based super-resolution technique in fixed tissue. This should be clearly mentioned.

The reviewer is right: we previously focused our discussion on fluorescence microscopy. As reported in the previous answer, we acknowledge these aspects in the updated version of the manuscript. Particularly, we have now added a sentence clearly stating that 3D electron microscopy is still considered the gold standard for characterizing spines' 3D morphology (we define 3D electron microscopy "gold standard 3D electron microscopy"):

[...] Their 3D morphology has been described over the years using the gold standard 3D electron microscopy (see for review [Harris, 2020; Rasia-Filho et al., 2023]).

The diffraction barrier at 200 nm should not be specified. It is depending on the wavelength and the NA of the used objective and can vary.

*The reviewer is right. 200 nm is a sort of 'averaged value', and it is regulated by the Abbe equation. Abbe Diffraction limit can be calculated as follows: $d = \text{wavelength} / (2 * \text{objective Numerical aperture})$. Since numerical aperture can vary from 1,3 to 1,57 for high magnification, and wavelength can vary from 488 to 750 nm in the most used confocal microscopes, this means that d can vary from 155 nm to 288 nm depending on the considered combination. The sentence "which breaks the diffraction barrier of 200nm" has been deleted, as in the edited section reported below.*

[...] XY to 500 nm in Z, respectively [Schermelleh et al., 2010]. On the other hand, super-resolution microscopy (e.g., STimulated Emission Depletion – STED), which breaks the diffraction barrier, enables the characterization of tiny dendritic spines [Collot et al., 2019; Godin et al., 2014; Hell and Wichmann, 1994; Wegel et al., 2016]. However, [...]

Please provide a reference for purkinje cells bearing 200.000 synapses.

A new reference was added, specifically:

Joseph Feher, "Balance and Control of Movement", in "Quantitative Human Physiology", Academic Press, 2012; doi: 10.1016/B978-0-12-382163-8.00037-2

After the following line:

They are characterized by a dendritic tree studded with numerous dendritic spines, receiving roughly 200000 synapses [Feher, 2012].

I usually do not care about wording. However, the authors should check whether „clearing“ brain samples should be rather used than „clarified“. This sounds a bit awkward.

Thank you for pointing out this. We used the term "clarified" as it is used in the literature of the field (e.g., [Tomer et al., 2014]). Nonetheless, we realize that "cleared" is used as well. We replaced "clarified" with "cleared".

Methodologies like how brains were cleared should be moved to the Mat&Met Part of the manuscript. (L103-104)

The first paragraph of the Results section was shortened:

To unravel the neuronal morphology on complex dendritic arbors, we focus on cerebellar Purkinje cells. They are characterized by a dendritic tree studded with numerous dendritic spines, receiving roughly 200000 synapses [Feher, 2012]. For reaching this objective, we used the L7GFP mouse line, that expresses cytosolic GFP (under the L7 promoter) within Purkinje cells. To optimize brain imaging on thick samples, we cleared them [Chung and Deisseroth, 2013; Magliaro et al., 2016] and processed them for long-term immunocytochemistry. The pre-processing steps and the labeling strategy are shown in Figures 1A to 1C (see Materials and Methods for details).

The removed information on tissue clearing was already present in the "Labeling and Imaging of brain slices" paragraph of the Materials and Methods section.

Figure 1k-l. Has no scale bar. It is difficult to judge about resolution.

We updated Figure 1 with a scale bar on panel K, and we report it below:

For correct spine detection Ilastik is obviously outperforming SENPAI.

We are aware that Ilastik is the best algorithm if its performance is measured as the number of spines whose centroids are included in the segmentation. Nonetheless, segmentation performance can be evaluated also taking into account other metrics, e.g., the spine volume and the spine surface area. In the first version of the manuscript, we hypothesized that Ilastik was over-segmenting the image, since, based on the measured volume and area data, it

seemed to avoid false negatives but also express more false positives with respect to HK-Icy and SENPAI.

To integrate the results on area and volume of segmented dendritic branches, we added in the manuscript an analysis based on Dice's coefficient, i.e., a measure of similarity between shapes, using the manual gold standard as reference. In addition, we carefully compared the segmentation provided by each algorithm with the manual one, and assessed performance for spine detection in terms of false negatives (manually detected spines left undetected by the evaluated algorithm) and false positives (spines detected by the evaluated algorithm in absence of manually detected spines). Based on this count, we compared SENPAI, HK-Icy and Ilastik in terms of sensitivity and precision.

Considering these measures, the comparison of segmentation algorithm performances is richer, as well as the information for potential users.

According to this more complete performance evaluation, SENPAI results can be considered very good with respect to HK-Icy and Ilastik, when precision and the aspects related to spine shape are considered.

We report below the updated Figure 6 in which we added the results obtained with the Dice coefficient analysis.

The caption was modified accordingly.

Figure 6 – Segmentations of 5 dendritic branches (from A to E) and their localization within a 93x STED dataset. A) On the left, we highlight in red on a 3D multi-stack STED image the dendritic branches A to E contained within a single stack and employed here for the validation of SENPAI. On the right, we compare the segmentations obtained for these branches with the state-of-the-art algorithms. The segmentations were performed with SENPAI (1st column), HK-Icy (2nd column), Ilastik (3rd column) and ManSegTool (4th column). For SENPAI, HK-Icy and Ilastik, the segmentations of the whole image are overlaid in gray, while the parcellation outcomes are highlighted in red. Note that ManSegTool is used to manually segment neurons, so the neighboring neurons are not displayed in grey. **B)** Quantitative comparison for the segmentations of the 5 dendritic branches; area (left) and volume (right) for the segmentations obtained with SENPAI (magenta), Ilastik (green) and HK-Icy (blue). The values are normalized with respect to manual segmentations. HK-Icy and Ilastik were integrated with SENPAI parcellation step to, respectively, assign spines to the dendrite and separate touching dendrites from each other. **C)** Quantitative comparison for the segmentations of the 5 dendritic branches; DICE coefficient was used to compare the segmentations obtained with the three algorithms against the manual segmentation, considering the whole neuron (top) and the only spines (bottom). Ideal algorithm would give a DICE coefficient of 1. The analysis on the spines was conducted by masking out the manual segmentation of the dendrite, both as it is and with 4 levels of dilation, to account for effects from oversegmentation of the dendritic branch. **D)** Table summarizing performances in terms of spine detection. Spines were counted in the manual segmentation, then, with respect to the manual segmentation, we defined for SENPAI, HK-Icy and Ilastik the number of False Positives and False Negatives, True Positives, the Sensitivity and the Precision. FP=false positives [count]; FN=false negatives [count]; TP=true positives [count]; S%=sensitivity [%of TP over TP+FN]; P%=precision [% of TP over TP+FP].

Information reported in Figure 6 was integrated by Figure S12 in the Supplementary materials, to better compare the performance of the three algorithms:

Figure S12: 2D and 3D renderings of manual, SENPAI, HK-Icy and Ilastik reconstructions for the dendritic branches displayed in Figure 6 of the manuscript. top) 2D renderings: for the manual segmentation, spines (purple) and the dendritic trunk (green) are visualized with different colors; for the three algorithms, the 2D rendering of Ilastik (blue), HK-Icy (red) and SENPAI (yellow) are overlaid on the manual segmentation. **middle)** 3D renderings using the same colors as in the top row. 3D renderings of the reconstructions obtained with the three algorithms are displayed with no overlay. **bottom)** 3D renderings using the same colors as in the top row. All segmentations are displayed

together on the left panel. On the three other panels, the segmentation obtained with each algorithm is rendered together with the manual segmentation.

We also added some sentences in the results section, paragraph "Segmentation of super-resolution datasets", describing these results:

Further analyses were performed to compare the branch areas and volumes obtained from the automatic tools with those obtained with the manual segmentation. We observed consistent differences between SENPAI and HK-Icy and Ilastik. While Ilastik delivers areas and volumes (Figure 6B) that are significantly larger than those from the manual segmentations (area normalized with respect to the ground-truth, median ± median absolute deviation (MAD) = 1.45 ± 0.42, volume normalized with respect to the ground-truth median ± MAD = 1.91 ± 1.08), SENPAI and HK-Icy's outcomes are very close to ManSegTool's (SENPAI area median ± MAD 0.88 ± 0.30, SENPAI volume median ± MAD 0.86 ± 0.49, HK-Icy area median ± MAD 0.83 ± 1.00, HK-Icy volume median ± MAD 0.75 ± 1.13). Particularly relevant is the analysis based on the Dice's coefficient [Dice, 1945] (Figure 6C), which quantifies the similarity between shapes. Considering the whole neuron branches, SENPAI achieves the highest values for all the five of them, while Ilastik performs worst for all neural branches but one (SENPAI Dice median ± MAD 0.81 ± 0.02, HK-Icy Dice median ± MAD 0.78 ± 0.11, Ilastik Dice median ± MAD 0.68 ± 0.05). We performed the same analysis with a focus on spines, excluding those voxels that were labelled as belonging to the dendrite by manual segmentation. To avoid bias introduced by eventual oversegmentation of the dendrite, we repeated the analysis with the dendrite mask being recursively dilated, and reported all the points in Figure 6C, bottom. SENPAI performs best again and is matched only for neuron E by HK-Icy (for the first iteration, SENPAI Dice median ± MAD 0.64 ± 0.07, HK-Icy Dice median ± MAD 0.50 ± 0.14, Ilastik Dice median ± MAD 0.43 ± 0.10). Merging information in Figure 6C, Ilastik clearly over-segments while HK-Icy under-segments.

Further comparisons in terms of spine detection are reported in Figure 6D. We compared spine-by-spine the reconstructions obtained with SENPAI, Ilastik and HK-Icy with the manual segmentation. We counted for each of the three algorithms the number of true positives, false positives and false negatives and estimated specificity and precision. While producing slightly lower specificity with respect to Ilastik, due to higher false negatives, SENPAI performs best in terms of precision. On average, SENPAI is better than HK-Icy on all topological and morphological metrics and performs better than Ilastik in terms of morphological reconstruction and number of false positives.

And similarly, in the discussion section:

For the 93x datasets obtained with STED, the results - in terms of assigned spines to the correct dendritic branch - were compared with manual segmentations performed with ManSegTool. All the approaches tested accurately reconstruct the neuron branch of interest (Figure 6A), and in particular for SENPAI, in 5 out of 6 cases, it provides faithful information about the morphology of the dendrites and their own corresponding spines. Ilastik provides a clear over-segmentation with respect to the manual ground truth, thus resulting in false positive assignments (Figure 6A, 6D) and over-estimated areas and volumes both at the whole branch and at the spines level (Figure 6B), while SENPAI correctly segments and assigns the spines in most of the cases investigated (Figure 6A, C) and provides estimates for area and volume in line with the manual ground-truth (Figure 6B).

We also present new results obtained with Ilastik on 40x images. Even in that case, SENPAI is outperforming Ilastik, and, in our opinion, this further confirms an advantage for SENPAI over Ilastik in the task proposed in this manuscript (segmentation of neuronal structures in dense scenarios). Nonetheless, we cannot exclude that in future works Ilastik might be proved better for specific tasks, such as spines' necks identification.

We also segmented new confocal and 3D-STED datasets, whose cells were labelled with membrane GFP. With respect to cytosolic GFP of the PCs presented in the manuscript, the

membrane GFP enable a better visualization of the spine neck – made of membrane more than of cytosol-. This image includes a dendritic branch of a pyramidal neuron with visible spines' heads and necks. On this image, SENPAI was able to correctly segment all the heads and necks along with the dendrite. The result is reported in section 13 and Figure S11 in the Supplementary Materials, and we report it below:

13. Reconstructions on non-cleared samples: focus on spines of hippocampal pyramidal cells

For one of the samples presented on section 12 (see Figure **S10**), two higher-resolution images were acquired, including in their field of view one same region (Figure **S11**). Labeling was performed with membrane GFP instead of cytosolic GFP, gaining specificity for spine necks. The two images were acquired with confocal (Figure **S11.C**) and 3D STED (Figure **S11.D**) techniques. We report in panels **E** and **F** of Figure S11 the segmentations obtained with SENPAI, color-coded for depth.

Figure S11: SENPAI's outcome on high resolution images of dendritic spines. **A)** Original data, rat pyramidal hippocampal neuron transfected with a CMV-membrane GFP plasmid. GFP was then amplified with a GFP immunochemistry and revealed with Alexa594. **B)** Global SENPAI reconstruction of A). **C)** Region of A acquired with confocal at 93x **D)** Region of A acquired with STED at 93x. **E)** SENPAI reconstruction (depth color-coded) on confocal data in C) **F)** SENPAI reconstruction (depth color-coded) on STED data in D). **G)** In-depth detailed zoom of SENPAI reconstruction on confocal data 93x slice. **H)** In-depth detailed zoom of SENPAI reconstruction on STED data 93x (slice). Fine details are shown by orange arrows.

We added a sentence in the Discussion to state the relevance of working on spines' necks identification when the resolution limit challenges their segmentation.

Another relevant novelty is the development and integration with a parcellation step, based on watershed transform of the intensity image. This allowed to successfully isolate single neurons in different scenarios as well as perform an automatic assignment of spines to the relevant dendrite. Indeed, increasing magnification is often unfeasible, and it comes at the cost of shrinking the field of view. In these cases, spine neck segmentation is impaired by resolution limits. Using the watershed transform on the intensity image, we provide now an efficient parcellation step for the automatic assignment of spines. Spine assignment is a crucial step preliminary to neck reconstruction. The latter can be achieved either using model-based [Erdil et al., 2019; Jain et al., 2021; Rodriguez et al., 2008] or data-driven [Cheng et al., 2007; Su et al., 2014] approaches.

Unclear, whether SENPAI works similarly on other neurons (e.g. pyramidal neurons; GABAergic neurons). Other imaging modalities like 2P and 2P-STED image analysis with SENPAI would have been a great add.

We tested the SENPAI algorithm on new cell types, i.e., **human pyramidal neurons, mouse pyramidal neurons, human cultured excitatory neurons, mouse cultured excitatory neurons, and mouse retinal neurons.**

A summary of the datasets we analyzed can be found in the table at the beginning of this document.

To answer your specific comment, here we stress that, some of the newly-added images are part of the BigNeuron GOLD166 dataset. This dataset included, among others, **2P images** of mouse excitatory neurons. The new datasets included in our work do not represent the dense cleared scenarios for which SENPAI was designed. Therefore, we report these new results in the supplementary materials, since in the main paper we discuss the overall framework merging sample preparation, imaging and processing.

On this independent dataset, we could provide validation of our results, using the BigNeuron bench-testing platform to quantitatively compare SENPAI's reconstructions with those obtained by 35 of the most popular state of the art algorithms, showing the excellent performances on these datasets. See Supplementary Material Fig10 in the section "SENPAI's benchmarking against state-of-the-art algorithms on independent datasets using the BigNeuron resource". We believe that the results reported for these new images speak for the generalizability of SENPAI to a large number of cell types. Details about the dataset and the results can also be found in the response to Reviewer #2 - Comment #1.

Other images used to test SENPAI include the test dataset of the NeuroGPS software, made available by the authors in [Quan et al., 2016], and rat hippocampal neurons in culture (glutamatergic excitatory) that were grown in the INSERM lab and acquired with the already-tested confocal imaging. Noticeably, a dataset is a correlative one that allowed to

confirm the multiscale performance of our algorithm (pyramidal neurons imaged with confocal 40x, 93x and STED 93x).

We added a paragraph in the Results section of the manuscript to list the neuron types and imaging modalities on which the SENPAI algorithm was tested.

Generalization to other neuron types and imaging modalities

To support the generalizability of our framework, we tested the algorithm on multiple neuron types (from rat, mice and human), including tissues and cultured cells, in uncleared samples and acquired with different imaging modalities – see supplementary materials. The neuron types we segmented include human pyramidal neurons, mouse pyramidal neurons, human cultured excitatory neurons, mouse cultured excitatory neurons, and mouse retinal neurons. The imaging modalities, other than confocal and confocal STED, include 2P microscopy. We report both quantitative and qualitative evaluations, that suggest the generalizability of our reconstruction algorithm across different imaging modalities and neuron types.

Among the tested datasets of native (uncleared) tissue tested with SENPAI, 34 belong to the BigNeuron GOLD166 dataset. On the BigNeuron bench-testing platform [Manubens-Gil et al., 2023; Peng et al., 2010; Peng et al., 2014a; Peng et al., 2014b; Peng et al., 2015], among 35 of the most popular state of the art algorithms, SENPAI scored in the top-three for five metrics over eight, and in the top-eleven for the remaining three metrics (see section 10 in the supplementary materials).

Another uncleared tested image is the Neuro-GPS test dataset made available by the authors of Neuro-GPS (see section 11 in the Supplementary Materials). Furthermore, at the INSERM facilities, four more datasets were produced: two datasets including in their field of view a whole labelled neuron, and two higher-resolution datasets from confocal and 3D-STED imaging the same dendritic branch with its spines. The results obtained from these four datasets are presented in sections 12 and 13, and Figures 10 and 11 in the Supplementary Materials.

We also tested SENPAI on inhibitory GABAergic interneurons. Samples are from rat hippocampal neurons from brain slice. Using transgenic mice expressing red fluorescent tomato within GABAergic interneurons, we could label hippocampal GABAergic interneurons. The segmentation result is promising and speaks in favour of a potential application of SENPAI to interneuron segmentation. Nonetheless, the lack of ground truth of such a segmentation, and of a reference statistic, as well as the complexity of the segmented structure, does not allow us to be 100 percent sure about result quality so that we took the decision not to include the results in the supplementary material although it seems impressive to us.

We redirect the reviewer to the Supplementary Materials, sections 10, 11, 12 and 13, for a detailed exposition of the new results. We report here two images as an example of the results achieved by SENPAI on a confocal image of a human pyramidal neuron (Figure S8.0 top), on a 2P image of a mouse excitatory neuron (Figure S8.0 bottom) and on two confocal images of hippocampal cells (Figure S10):

Figure S8.0 Benchmark of SENPAI using the BigNeuron resource. **A)** Exemplary dataset (m16_cing_1_9_cropped_neurona.v3dpbd, human pyramidal cell labeled with Lucifer Yellow and acquired through confocal microscopy, resolution 0.24 μm x 0.24 μm x 0.42 μm [Peng et al., 2017]) from the human Allen confocal dataset along with **B)** gold-standard (GS) segmentation and **C)** SENPAI segmentation. **D)** Exemplary dataset (neuron2.v3dpbd, excitatory neuron from auditory cortex, layer 2/3, acquired through 2-photon microscopy, resolution 0.27 μm x 0.27 μm x 1 μm [Peter et al., 2013]) from the mouse cultured cell Cambridge dataset along with **E)** gold-standard (GS) segmentation and **F)** SENPAI segmentation.

Figure S10: SENPAI's outcome on two 3D stacks of cultured hippocampal pyramidal cells (pixel size 91,41 nm x 91,41 nm x 0.28nm). A-B) Original Image Maximum projection for the GFP channel. **C-D)** segmentation obtained with SENPAI with depth color-coded (cold colors indicate deeper planes). **E-F)** Skeletonizations of SENPAI segmentations produced with NeuTube.

In addition, we modified a paragraph in the Discussion to elaborate on possible limitations of our algorithm:

The performance of SENPAI was assessed on cleared datasets of Purkinje cells imaged with confocal and 3D STED modalities. This scenario was chosen as a proof of concept since Purkinje cells are known to be highly densely arranged within the murine cerebellum. Moreover, SENPAI has been successfully tested on non-cleared samples acquired with multiple widespread acquisition modalities, such as confocal, 3D STED and 2P, and including different neuron types. Nonetheless, several other imaging modalities are in use within the scientific community, as well as labelling modalities that are specific to some parts of the neuron, e.g., membright for the membranes [Collot et al., 2019] or phalloidin for spine heads [Hotulainen and Lappalainen, 2006]. In all the cases where local inhomogeneities characterize neuronal structures, e.g., for non-uniform distribution of staining across the cell, SENPAI might show a suboptimal behaviour. Specifically, the use of spatial derivatives, that are sensitive to image intensity changes, might lead to hollow 3D segmentations or poorly connected reconstruction of the dendrites. In some cases, such as those presented in the supplementary material, the 3D spatial smoothing preprocessing step allowed to increase neural structure homogeneity and consequently the behaviour of second order derivatives. Further work, as for instance the use of other preprocessing steps, might thus be needed to improve the generalizability of SENPAI to specific staining and imaging modalities.

Other future developments will include the test of SENPAI to cleared samples from other brain regions, to fully evaluate the potential of the proposed framework. We will also explore the possibility of estimating the path of spines necks even when SENPAI is not able to connect the spine with the main neural body and to provide the user with the possibility of characterizing spine population in terms of density and morphology.

In addition, we added a sentence in the introduction to inform the reader about the tests performed on freely available databases:

Furthermore, we provide an original validation paradigm for evaluating the quality of segmentation, in which morphometrics extracted from the segmented neurons are compared with reliable quantitative indexes of Purkinje cell morphology available in literature [Vormberg et al., 2017]. To evaluate the generalizability of the segmentation algorithm to other neuron types and imaging modalities, we tested SENPAI performances using datasets comprising both cultured and ex-vivo neurons, most of them freely available. This study represents a significant advance in the state of the art for the comparison of segmented neurons, particularly when a reliable manually segmented ground-truth is not available and/or achievable.

L446: White laser excitation? What was the exact excitation wavelength? What was the emission wavelength of the used dye. What were the bp settings to detect the emission?

Excitation laser wavelength was 594 nm laser line of a white pulsed laser with Alexa 594. Depletion laser was a pulse 775 nm. Emission was collected using an SMD HyD detector between 600 nm and 640 nm. We added these details in the methods section as follows:

Confocal and STED images were acquired with a confocal laser scanning microscope LEICA SP8 STED 3DX equipped with a 20x and 93x/1.3 NA glycerol immersion objective or a 40x NA1.3 oil objective and with 3 hybrid detectors (HyDs). The specimens were imaged with a pulsed white-light laser and a pulsed 775 nm depletion

laser to acquire nanoscale imaging. Depletion laser power balance (between 2D and 3D) is adapted depending on transparency sample laser to optimize resolution and preserve signal in depth.

Confocal Image of rat hippocampal pyramidal cell in culture was taken with 40X NA 1.3 Leica oil objective with a pixel size of (91,41 nm x 91,41 nm x 280 nm).

3D-STED was done with a 93X NA1.3 glycerol objective. Images were averaged 16 times in line and acquired with a magnification zoom >3. Excitation laser was adjusted to avoid any saturated pixel. Rat hippocampal pyramidal cells in culture were acquired with voxel size of 40x40x130 nm within a Z stack of 21 slices. Cerebellum Purkinje cells were acquired with voxel size of 62x62x60 nm within a Z stack of 35 slices.

Under the given address the Matlab code is not deposited. It cannot be evaluated.

We regret that our test code was not easy to run. As suggested in the README file in the main folder available at: <https://drive.matlab.com/sharing/b815586c-cd0d-413d-8ce4-14712bca33ba/> it is necessary to download the whole folder on a laptop equipped with Matlab and the 'Statistics and Machine Learning Toolbox' and the 'Image Processing Toolbox' installed.

Then, after unzipping the folder, it is sufficient to run the script test_script.m from its folder. Once these steps are completed, a 3D rendering of a subregion of the Figure reported in Fig. 3 of the manuscript should appear in about 10 minutes.

To make the revision process easier, we are attaching a .zip folder containing all the necessary files (scripts, functions and data) to run the senpai code.

Reviewer #2

The manuscript by Cauzzo and coworkers describes a novel algorithm (termed SENPAI) for segmenting microscopic images into individual neurons, dendrites and dendritic spines. The authors show using an example dataset of Purkinje cells that their algorithm performs better than existing segmentation algorithms even with extremely densely packed neuronal assemblies, which represents an important advance compared to other (semi-)automated neuron segmentation tools. Furthermore, the segmentation of dendritic spines from STED microscopic images of Purkinje cells was shown to be nearly as successful as with a machine learning based image analysis tool (Ilastik). Therefore, the new algorithm could be very useful for reconstructing large groups of neurons down to the single spine level. While the results of the segmentation using SENPAI are promising and the manuscript is well written, it is important to show the applicability of the algorithm to different neuron types and there are some important methodological details missing.

Major Comments:

1. The major limitation of the performance assessment is that only one cell type, namely Purkinje cells, is used. The dendritic arbor of Purkinje cells is rather two-dimensional compared to for instance cortical pyramidal cells, and it would be important to show that the algorithm works on different types of neuronal morphologies. Furthermore, in order to fairly evaluate the performance of the SENPAI algorithm compared to the previously published neuron segmentation tools, it would be best to use an independent dataset.

We thank the reviewer for the comment. We are aware that assessing the performance of SENPAI framework on one single cell type, namely Purkinje cells (PCs), may represent a limitation. For this reason, we tested the proposed algorithm on new independent datasets, including both different types of neurons and different acquisition modalities (see data table at the beginning of this document); we both quantitatively and qualitatively evaluate the segmentation performance – see manuscript and supplementary material.

We kept the description of the results on PCs from ex-vivo samples since, even if they show a rather two-dimensional structure, they still offer some interesting features. Specifically, the sample is a thick and cleared tissue slice, showing cells in their native arrangement; moreover, the PCs are densely packed and present a highly branched dendritic tree such that even the manual segmentation gold standard can be a difficult or impracticable avenue.

To answer the reviewers' comments and show potential readers the generalizability of the proposed approach, we analysed the datasets listed in the table at the beginning of this document. Most of the images in these new datasets depict cortical pyramidal cells. Since segmenting uncleared images is beyond the aim of the present work, we added the new results only in the supplementary materials.

All the results obtained on uncleared samples have been described in detail in the supplementary materials, and are listed below for convenience.

- 1. We exploited the BigNeuron resource [Manubens-Gil et al., 2023] to test the efficacy of SENPAI on **independent datasets** and against previously published neuron segmentation tools. The BigNeuron is a resource to benchmark and predict the performance of algorithms for automated tracing of neurons in light microscopy datasets whose details are available in [Manubens-Gil et al., 2023]. Among the many features available in the BigNeuron resource, authors provide (i) a database for which gold-standard manual segmentations are available, (ii) a set of algorithms for which automatically traced neurons are available, (iii) a set of metrics for comparing different algorithms, and (iv) a shiny app to compare new segmentations (in our case, the segmentations provided by SENPAI) of the benchmark datasets with the gold standard manual segmentation and against other algorithms.*

Here, we focused on a subset of the Gold166 dataset, containing images from mammal samples. Particularly, we segmented neurons from the (i) human Allen (confocal datasets of pyramidal neurons), (ii) human cultured (confocal datasets of excitatory neurons stem-cell derived), (iii) mouse cultured (two-photon imaging datasets of excitatory neurons) and (iv) mouse RGC (confocal datasets of retinal ganglion cells) subsets. Many of these image volumes were generated within large-scale neuroinformatics projects, such as the Allen Mouse and Human Cell types projects (<http://celltypes.brain-map.org/>). For each subset, we used the quality metrics available on BigNeuron: (i) the entire-structure-average-from-goldstandard-to-neuron (ii) the entire-structure-average-from-neuron-to-goldstandard, (iii) the average of bidirectional entire structure averages with respect to the gold standards, (iv) the different structure average, (v) the percent of different structure from goldstandard to neuron, (vi) the percent of different structure from neuron to goldstandard, (vii) the percent of different structure and (viii) the aggregated distance [Manubens-Gil et al., 2023; Peng et al., 2014a]. Finally, we exploited the BigNeuron shiny app to compare the SENPAI reconstructions with those obtained from commonly used state of the art algorithms. The result of the benchmark, along with an exemplary of the SENPAI's segmentation, is reported in section "10. SENPAI's benchmarking against state-of-the-art algorithms on independent datasets using the BigNeuron resource" and in Fig. From S8.0 to S8.8 in the updated version of the supplementary materials. We observed that SENPAI performed extraordinarily well, with comparable or better performance with respect to most popular state of the art algorithms. In this light, we emphasize that although SENPAI was developed for completely different purposes, i.e. the segmentation of cleared samples, it still performs satisfactorily well on independent benchmark available in the state of art. Briefly, for each metric we report between brackets the ranking of SENPAI among all the considered algorithms:

- a. the entire-structure-average-from-goldstandard-to-neuron (2/35)
- b. the entire-structure-average-from-neuron-to-goldstandard (3/35)
- c. the average of bidirectional entire structure averages with respect to the gold standards (1/35)
- d. the different structure average (1/35)
- e. the percent of different structure from goldstandard to neuron (11/35)
- f. the percent of different structure from neuron to goldstandard (11/35)
- g. the percent of different structure (8/35)
- h. aggregated distance (3/35)

Where, metrics (a), (b), (c) and (h) measure how different two reconstructions are, and metrics (d), (e), (f), (g) measure the extent of differences between two reconstructions considering only points above a tolerance threshold S (here set to the default value of 2 voxels, as implemented in Vaa3d and in the BigNeuron resource) [Peng et al., 2011]. Of note, the BigNeuron shiny app for comparing algorithms is available both as standalone application, as well as a web app. Here, we exploited the standalone version of the app in a system running Ubuntu for comparing the segmentations. Indeed, among the many features available in the shiny app, the comparison of different segmentations can be performed only locally, in a machine running Ubuntu and with Vaa3d already installed (see details here: <https://github.com/lmanubens/BigNeuron#os-requirements> "To allow upload of novel reconstruction algorithm bench testing results, the Shiny app relies on a Linux binary version of Vaa3D. Thus, this functionality is only supported when running on Linux.").

2. We downloaded the test dataset linked to the NeuroGPS-Tree paper [Quan et al., 2016], an **independent confocal dataset** of a neuronal population from neocortex. We reconstructed with SENPAI the four neurons reconstructed in Figure 2, panel C, in [Quan et al., 2016]. In section 11 and Figure S9 in the supplementary materials we present our reconstruction next to the original image, a manual reconstruction, and reconstructions obtained with NeuroGPS, Open-Snake and NeuroStudio, as reported in [Quan et al., 2016].
3. We applied the SENPAI algorithm to four new datasets depicting mice cultured hippocampal **pyramidal cells**, acquired at the INSERM facilities: two datasets including in their field of view a whole labelled neuron, and two higher-resolution datasets imaging with confocal and 3D-STED techniques the same dendritic branch with its spines. The results obtained from these four datasets are presented in sections 12 and 13, and Figures 10 and 11 in the Supplementary Materials.

We report below a paragraph that we added to the Results section, describing the generalizability of SENPAI to different imaging modalities and imaged neuron types, in light of the new reconstructions.

Generalization to other neuron types and imaging modalities

To support the generalizability of our framework, we tested the algorithm on multiple neuron types (from rat, mice and human), including tissues and cultured cells, in uncleared samples and acquired with different imaging modalities – see supplementary materials. The neuron types we segmented include human pyramidal neurons, mouse pyramidal neurons, human cultured excitatory neurons, mouse cultured excitatory neurons, and mouse retinal neurons. The imaging modalities, other than confocal and confocal STED, include 2P microscopy. We report both quantitative and qualitative evaluations, that suggest the generalizability of our reconstruction algorithm across different imaging modalities and neuron types.

Among the tested datasets of native (uncleared) tissue tested with SENPAI, 34 belong to the BigNeuron GOLD166 dataset. On the BigNeuron bench-testing platform [Manubens-Gil et al., 2023; Peng et al., 2010; Peng et al.,

2014a; Peng et al., 2014b; Peng et al., 2015], among 35 of the most popular state-of-the-art algorithms, SENPAI scored in the top-three for five metrics over eight, and in the top-eleven for the remaining three metrics (see section 10 in the supplementary materials).

Another unclassified test image is the Neuro-GPS test dataset made available by the authors of Neuro-GPS (see section 11 in the Supplementary Materials). Furthermore, at the INSERM facilities, four more datasets were produced: two datasets including in their field of view a whole labelled neuron, and two higher-resolution datasets from confocal and 3D-STED imaging the same dendritic branch with its spines. The results obtained from these four datasets are presented in sections 12 and 13, and Figures 10 and 11 in the Supplementary Materials.

2. If the Ilastik image segmentation tool could also be applied to the 40x confocal dataset, the results would be interesting to compare to those of the SENPAI algorithm. If Ilastik is not applicable to this dataset, this should be stated in the manuscript as it could represent a clear advantage of the SENPAI algorithm.

We tested Ilastik on 40x. Visual inspection and Strahler analysis show that it performs less than Icy and SENPAI, better than Neutube and NeuroGPS (see Figure 4 of the manuscript). Nonetheless, it is worthwhile noting that Ilastik cannot separate neurons. In this light, we had to apply SENPAI parcellation step to the segmentation performed by Ilastik to properly compare the results.

We modified Figures 3 and 4 of the manuscript. In the Results section of the manuscript, we modified as follows:

Segmentation step: Comparison with other segmentation or tracing algorithms

Confocal images acquired from cleared mouse brain samples (see Materials and Methods and Figures 1-2) were used to compare the segmentation of neurons obtained with SENPAI with those performed with four state-of-art algorithms, namely, the automatic HK-Means segmentation plugin [Dufour et al., 2008] included in the software Icy [de Chaumont et al., 2012] (hereinafter HK-Icy), the semi-automatic tracing software NeuroGPS [Quan et al., 2016], the semi-automatic segmentation software Ilastik [Berg et al., 2019] and the semi-automatic tracing software NeuTube [Feng et al., 2015]. When single neuron identification failed with a specific algorithm, we parcellated its outcome using the step 2 of the SENPAI workflow to provide a reconstruction comparable with the others. The example in Figure 3 shows how neuron segmentations performed by SENPAI and HK-Icy result in a notably denser and more intricate outcomes than those provided by NeuTube and NeuroGPS, while Ilastik provides richer segmentations whose complexity lays in between. We also report in supplementary Figures S2, S3 and S4 the 27 single-neuron reconstructions obtained with SENPAI and the state-of-art tools and used to test and validate SENPAI.

Figure 3 – 3D visualization of the original confocal dataset (acquired with a confocal microscope equipped with a 40x objective – details in Methods) and comparison of the performance of the five – including SENPAI - algorithms tested. All images are produced with the Icy GUI. Tracings are converted to volumetric segmentations with the SWC2IMG ImageJ plugin [Radojević and Meijering, 2019]. The same 10 neighboring neurons and one exemplary single neuron are depicted using the same color code for all tools. Another example segmented neuron is depicted within a 3D rendering of the original image in supplementary movie S2.

We also compared the segmentation performance of SENPAI, HK-Icy, Ilastik, NeuroGPS and NeuTube in terms of the area under the curve (AUC) obtained through Sholl analysis, a method used to quantify dendritic structure [Sholl, 1955] (see Figure 4A). SENPAI, HK-Icy and Ilastik show significantly different higher values with respect to NeuTube and NeuroGPS (Friedman test, $p < 10^{-16}$, and post-hoc comparisons with Tukey's honest significant difference criterion, $p < 10^{-2}$, Figure 4B, right) indicating a higher sensitivity to detect complex dendritic arbors. Additionally, we compared the area-to-volume ratio of the segmentations provided by SENPAI, HK-icy and Ilastik (Figure 4B, left). It should be noted that this index cannot be extracted for NeuroGPS and NeuTube, since they

do not provide volumetric information. While HK-Icy and Ilastik show similar values, SENPAI provides significantly higher area-over-volume ratios (Friedman test, $p < 10^{-8}$, and post-hoc comparisons with Tukey's honest significant difference criterion, $p < 10^{-5}$).

A) Area under the Sholl curve

B) 40x statistics

Figure 4 – Quantitative comparison of the segmentations on 40x images. **A)** Schematization of the rationale behind 3D Sholl Analysis: we compute the Area Under the Curve for the number of crossings of each neuronal structure for a sphere centered on the root node (i.e., the soma centroid) and having its radius increased; **B)** Quantitative comparison for segmentations from 27 neurons on 40x images; Left: Area-Volume ratio for SENPAI (magenta), HK-Icy (blue) and Ilastik (green) (Friedman test, $p < 10^{-8}$, horizontal bars mark significant differences as determined with post-hoc comparison using Tukey's honest significant criterion, $p < 10^{-5}$); Right: AUC of the Sholl analysis across segmentations obtained with SENPAI (magenta), HK-Icy (blue), Ilastik (green), NeuroGPS (red) and NeuTube (gray). (Friedman test, $p < 10^{-16}$, horizontal bars mark significant differences as determined with post-hoc comparison using Tukey's honest significant criterion, $p < 10^{-2}$).

[...]

Using an ad hoc script developed in Matlab (see Methods) (Figure 5B), we extracted the SO for skeletonized neurons obtained from SENPAI, HK-Icy, NeuroGPS, NeuTube and Ilastik. The SOs were compared with those obtained by Vormberg et al for Purkinje cells. The results are shown in Figure 5C and detailed in the supplementary materials (Table S2).

SENPAI and HK-Icy achieved segmentations with the same SN mode as the one identified by Vormberg et al. for Purkinje cells, with SENPAI scoring more hits. All measures reported in Figure 5C shows that the segmentations obtained with HK-Icy and SENPAI have similar metrics to those described in Vormberg et al, while the results obtained with NeuTube and NeuroGPS differ. Ilastik achieved a lower SN mode (5), although results are close to those of HK-Icy and SENPAI. In addition, SENPAI outperforms the other algorithms matching the steepest decay

for the normalized number of segments (slope= -1.23) found for Purkinje cells in the reference work and presenting smaller deviations in the curves of subtree size and total dendritic length.

Figure 5 –A) Schematization of SO and of the meaning of some of the extracted features; **B)** Schematization of the validation pipeline; **C)** Graphical comparison of the SO of Purkinje cells found in literature and measured on the neurons segmented with SENPAI and the state-of-art tools. The parameters based on the SO reported by Vormberg et al. on six types of neural cells (left, edited from Vormberg et al.) and computed on 27 Purkinje neuronal cell segmented with SENPAI (magenta), HK-Icy (blue), Ilastik (green), NeuroGPS (red) and NeuTube (gray) from 40x images (right). It should be noted that the parameters, with the exception of the Branch Bifurcation Ratio, were computed only for neurons whose SN was as equal to its mode across all example neurons, in line with Vormberg et al.

3. The description and illustration of the SENPAI workflow in could be improved. The preprocessing steps (deconvolution, marking the somata) that were performed should be clearly stated, including the software used if applicable. The individual steps of the workflow shown in Figure 2 could be better illustrated by using the same source image throughout, and by including the final segmentation in panel D and the grayscale image used for the parcellation step in panel E.

Figure 2 was modified to better illustrate each step of the workflow, by using the same source image throughout, by including the final segmentation in panel D and the grayscale image used for the parcellation step in panel E.

The caption was modified accordingly:

Figure 2 – the SENPAI algorithm. A) Test dataset: 40x confocal (left) and 93x STED (right) datasets. B) The SENPAI rationale, based on the selection of classes displaying negative values of the second derivative along the three main axes. C) Rationale for the estimation of the K parameter of the K-means clustering: the selected K is the one for which the histograms of second-order derivatives show three different

classes achieving maximal average value: here, from the histogram of D^2x (second derivative along the x axis), class 3 (cyan) encodes outer borders along the x direction, as it is the only class with values clearly above 0; similarly, class 4 encodes outer borders along the y direction (D^2y histogram); class 2 along the z direction (D^2z histogram); class 1 encodes both low- and high-intensity homogeneous image portions, and is labelled as background along with classes 2, 3 and 4. D) SENPAI workflow - Step 1: K-means clustering, performed on the unsmoothed image (top) and optionally, in parallel, on the image smoothed with a 3D gaussian filter (below). Class selection is performed independently on K-means classes (color-coded as in panel C) for each clustering level. Resulting binarized images (green for clustering level 1, pink for clustering level 2, white for the overlap) are merged by logic OR. E) SENPAI workflow - Step 2: the segmented image is parcellated using morphological reconstruction and 3D watershed transform computed on the morphologically reconstructed grayscale image and applied to the binary segmentation. Left: isolation of connected structures belonging to the same neuron: soma markers (yellow, placed by user) define wells for the catchment basins (edges in gray). Right: connection of a neuronal portion (e.g., a neuronal branch) to smaller clusters (i.e., dendritic spines). Groups of neuronal clusters assigned to a single neuronal entity are displayed in different colors in their relative catchment basin (grey).

In addition, we provided further details in the Materials and Method section, clarifying in particular how we marked the somata and listing the functions that we used and that can be found in the toolbox shared folder. We report here the modified section:

SENPAI is an automated tool implemented in Matlab (The MathWorks-Inc., United States) and released as a freely available toolbox at: <https://drive.matlab.com/sharing/b815586c-cd0d-413d-8ce4-14712bca33ba>. It can be used to reconstruct neuronal morphology, from the spines up to the whole neuron arborization, starting from 3D optical images from any type of microscope including confocal, multiphoton, STED, etc. It implements two fundamental steps: a segmentation step, based on topology-Informed K-means, and a parcellation step.

Segmentation step

The first step to obtain faithful three-dimensional neuron reconstructions is to isolate neurons from background. SENPAI tackles this step through its main core segmentation algorithm, a topology-informed K-means clustering. Briefly, the K-means portions n observations (X_1, \dots, X_n) defined on a d -dimensional space into K clusters, by minimizing the within-cluster variances [Lloyd, 1982]. In SENPAI, such space is made of $d=4$ features of interest, comprising the pixels' intensity and second order spatial derivatives computed along the three main cartesian axes, while the n observations are represented by the total number of pixels entering the K-means. Since K-means convergence may be affected by initial conditions, additional hyperparameters, such as the number of replicates and the maximum number of iterations to reach convergence are included (in SENPAI, replicates are set to 10, and for each replicate, a maximum of 1000 iterations is allowed). To reduce the computational load for the K-means, a subset of background pixels is excluded from the clustering. In SENPAI this is tackled by simple thresholding, although more sophisticated methods could be used [Sternberg, 1983]. Preprocessing steps might include pointwise transformations as the logarithmic one, to adjust the image dynamics. Moreover, since the K-means may be computationally expensive, it can be also run on smaller dataset crops, whose size is user-defined, loaded, and processed sequentially to limit the usage of memory.

Each 3D structure defined by a high-intensity signal on a low-intensity background is characterized by intensity gradients at the boundaries, coded in the second derivative by a positive peak at the external boundary and a negative peak at the internal one. A key feature of SENPAI's K-means clustering is given by the choice of the number of classes K , and by how such classes are interpreted and used for the segmentation. The algorithm iteratively searches for the smallest K that allows to have (i) a class encoding positive values of the second order derivatives for each direction in the 3D space (representing the outer borders of the neuron structures), (ii) each direction (x, y, z) is encoded in a different class. Afterwards, for a given number of classes K , SENPAI identifies neurons by sorting and choosing the obtained clusters. Sorting is based on average pixels' intensity values of each cluster, from the lowest to the highest. Then, the clusters associated with the neuronal structures are given by all the clusters with the highest average pixel intensity values and with all the second order derivatives whose values are negative (these clusters represent signal from the neuron body see Figure 2B-C).

An optional parameter is the size of the pre-filtering kernel, which represents one of the salient features of SENPAI and allows to focus on multiple levels of detail of the image. First, the dataset is filtered using a 3D

Gaussian filter and derivatives are computed. After the pre-filter step, the K-means clustering is performed on the pixels' intensity of the unsmoothed dataset, and second order spatial derivatives computed on the smoothed image. This pre-filter step can be also progressively performed twice in a row by using two smoothing levels. This option allows to capture structures at different scales, since it eliminates spurious and unwanted boundaries, e.g., those within thick branches or big somata. When multiple levels of smoothing are used, the resulting segmentations are merged by union to compose the final segmentation (Figure 2D).

However, since inhomogeneities in pixel intensity within big neuron structures (e.g., the soma) cannot be all smoothed, and may be identified as boundaries - leading to false negatives -, we implement a function for filling such holes within the segmentation. Similarly, we eliminate spurious small clusters (smaller than 7 voxels, the smallest 3D symmetrical structure) detected from inhomogeneities in the background.

Parcellation step

The second step of the algorithm aims at isolating single neurons from the foreground identified via the K-means clustering. To this end, SENPAI performs a parcellation procedure based on the watershed transform. This defines a catchment basin for each structure, that allows to mark what belongs to a particular structure and what does not. The parcellation has a two-fold purpose: when dealing with single neuron reconstruction, it splits neurons that may have been merged in a single structure, while when dealing with spine segmentation, it groups together disconnected structures. Thus, the parcellation is performed through two different functions.

The catchment basins are produced for each marked core. The cores are defined automatically in the super-resolution case, by imposing a threshold on the size of the clusters composing the segmented image. While big clusters are labeled as cores, smaller clusters are labeled as spines to be assigned to such cores. On the other hand, for the standard resolution microscopy, cores correspond to somata. These can be marked by the user thanks to a GUI implemented in SENPAI.

Once the cores are defined, the original grayscale dataset is filtered with a 3-by-3-by-3 median filter, and its complement obtained. A morphological reconstruction is applied on this latter, so that the regional intensity minima outside the cores are eliminated. A watershed transform produces one catchment basin for each remaining regional minimum.

The parcellation can be modified *a posteriori*, thanks to a routine to visually assess the quality of the segmentation neuron-by-neuron, and to set additional markers on spuriously assigned branches. Incorrect assignments are usually caused by the presence of branches of neurons whose soma lies outside the acquired dataset. Operationally, this is done thanks to a function within the SENPAI toolkit, implemented to correct the parcellation by adding to the mask of the somata some additional clusters. These will allow the definition of separate catchment basins for the branches wrongly assigned.

4. The rationale for using Strahler order based branching statistics to validate the segmentation of the Purkinje cells should be stated more explicitly. As Vormberg et al. (2017) showed, there are two categories of Strahler order based branching parameters which differ in their universality: the topological parameters are the same for all binary trees, while the metric parameters differ based on neuron type and can therefore be meaningfully used to classify different neuron types. Therefore, the statement in lines 244-245, "each neuron type is described by a specific combination of these 9 parameters" is misleading. Also, the Methods contain no details of how the analyses shown in Figure 5 were done.

We acknowledge that further details can be reported on the role of the different measures addressed in Vormberg et al. (2017). In Figure 5, we placed topological measures on the left side and metric measures on the right side. Although only metric measures (normalized branch diameter, total normalised dendritic length, normalised average segment length,

normalised average branch length, right side of Figure 5) can be used to discriminate cell types, we believe that topological measures as well have to be considered for the evaluation of algorithms, in order to confirm at least the binary tree-like nature of proposed structures. We modified the addressed sentence to be clearer to the reader.

Comparison against documented information on Purkinje cell morphometrics

Purkinje neurons branch extensively and almost completely fill spaces with little overlap [Hirano, 2018]. They have been reconstructed using tools such as NeuroMorpho.org [Ascoli et al., 2007] from public repositories of single injected neurons [Ledderose et al., 2014; Nanda et al., 2020] and several parameters have been used in literature to describe their morphology. The availability of ground-truth summary statistics describing the topology and morphology of neuronal populations allows for validation of new reconstructions when for the specific sample under exam a ground truth is not available and the time-consuming manual gold standard can not be obtained due to the density of the scenario. Vormberg and co-authors [Vormberg et al., 2017] described the topological and morphological properties of six types of neurons (including Purkinje cells) by means of the centripetal Horton-Strahler ordering (SO, see Figure 5A). This is a well-known index of branching complexity, which accounts for nine different parameters. These include the Strahler Number (SN), i.e., the maximal SO assigned to a segment in the neuronal tree. Then, four topological parameters: Normalized Number of Segments per SO, Normalized Number of Branches per SO, Branch Bifurcation Ratio, Normalized Topological Subtree Size per SO. And finally, four metric parameters: Normalized Branch Diameter per SO, Total Normalized Dendritic Length per SO, Normalized Average Segment Length per SO, Normalized Average Branch Length per SO. While topological parameters are universal to binary tree – like structures, metric parameters can be used to differentiate cell types. For example, total dendritic length reveals the dimensionality of dendritic trees, with shallow slopes for planar trees, e.g., Purkinje, and pronounced slopes for 3D trees.

Using an *ad hoc* script developed in Matlab (see Methods) (Figure 5B), we extracted the SO for skeletonized neurons obtained from SENPAI, HK-Icy, NeuroGPS, NeuTube and Ilastik. The SOs were compared with those obtained by Vormberg et al for Purkinje cells. The results are shown in Figure 5C and detailed in the supplementary materials (Table S2).

SENPAI and HK-Icy achieved segmentations with the same SN mode as the one identified by Vormberg et al. for Purkinje cells, with SENPAI scoring more hits. All measures reported in Figure 5C shows that the segmentations obtained with HK-Icy and SENPAI have similar metrics to those described in Vormberg et al, while the results obtained with NeuTube and NeuroGPS differ. Ilastik achieved a lower SN mode (5), although results are close to those of HK-Icy and SENPAI. In addition, SENPAI outperforms the other algorithms matching the steepest decay for the normalized number of segments (slope=-1.23) found for Purkinje cells in the reference work and presenting smaller deviations in the curves of subtree size and total dendritic length.

In the Methods, as part of the “Performance Assessment” subchapter, we expanded the “Neuron morphometric evaluations and statistical analyses” paragraph by introducing the custom code used to compute Strahler-based metrics. In addition, we shared this code together with the SENPAI toolbox, under the same link.

40x Confocal

The comparisons on the single neuron segmentations obtained with SENPAI, HK-Icy and Ilastik, as well as the tracings obtained with NeuroGPS and NeuTube, were performed on 27 cells.

We provide here the key parameters that had to be set for segmenting those neurons on HK-Icy, and NeuroGPS. For the HK-Means plugin, the optimal segmentation was achieved by setting the intensity classes to 6 for the confocal datasets. The minimum object size was set to 9000 pixels while the maximum object size was set to 7800000 and 4600000 pixels respectively for the two confocal datasets. The minimum object intensity was set to 0. Further details on the methods and rationale are reported in the Supplementary Materials (see Figure S7). The segmentation with Ilastik was obtained using two classes, corresponding to the background and the neurons,

and employing the maximum number of features for classification. For NeuroGPS, we set the intensity threshold to the minimum to achieve a dense segmentation. Since NeuTube is a semi-automatic tracing tool, there are no parameters to be set.

Neuron morphometric evaluations and statistical analyses

The volumes and areas of the segmented neurons were computed with an *ad hoc* script in Matlab. Sholl Analysis was performed via Neuroanatomy [Ferreira et al., 2014], a Fiji [Schindelin et al., 2012] plugin, whereas the corresponding area under the curve (AUC) was obtained in Matlab.

To evaluate any significant differences of area-to-volume ratio among SENPAI, HK-Icy and Ilastik segmentations, we employed the two-tailed Friedman non-parametric test for paired data. Data obtained from NeuroGPS and NeuTube are not included in the analysis, since they produce tracings with insufficient details on neuron morphology. A two-tailed Friedman non-parametric test for paired data followed by a post-hoc analysis ($\alpha = 0.05$) was used to compare AUCs.

Skeletonization and Strahler Analysis

SENPAI is a segmentation algorithm meant to provide the volumetric reconstructions of neurons in the form of binary 3D volumes. Since the validation metrics for neuron reconstructions, such as Strahler analysis, are usually designed for assessing the goodness of neuron tracings, i.e., reconstructions in the form of tree-like graphs, we provide a routine for the skeletonization of our segmentations.

Starting from the raw 3D image stack, the binary segmentation of one neuron and the markers of the somata will be exploited to produce the skeletonization, in the shape of a Matlab tree structure and of a SWC-like matrix, which is the format usually used to describe a graph or tree as a list of nodes, with their coordinates, diameter, and parent node. After the reduction of the neuron volumetric segmentation to 1-pixel wide curved lines, the function converts it into a graph object. For each node the function automatically attributes an identification index. Node diameters are defined on the basis of the intensity of the original image at the nodes' coordinates. The graph is converted to a tree object: cycles are identified and then removed by cutting them when the image intensity is the lowest one. Finally, the SWC matrix is built by storing in a matrix, for each node, its identification index, its coordinates in the space, its diameter and the identification index of the parent node.

In addition, the code used to compute the Strahler metrics is available. Starting from the binary segmentation of one neuron and the relative SWC matrix, a function is implemented to produce the Strahler ordering. Specifically, the output is a matrix storing the statistics based on this ordering, namely: numSegSO (number of segments per SO), numSegSONorm (number of segments per SO, normalized by the total number of segments), segLAve (average segment length per SO), PSnum (coefficients of the linear fit for the normalized number of segments per SO), segDAve (average segment diameter per SO), TOTL (total length of segments), TopoSubLAve (topological subtree size per SO), numBrSO (number of branches per SO), numBrSONorm (number of branches per SO, normalized by the total number of branches), brDAve (average branch diameter per SO), brLAve (average branch length per SO), PBnum (coefficients of the linear fit for the normalized number of branches per SO), normTotL (total dendritic length per SO). The definition of the Strahler order of the segments constituting the whole arborized structure is based on the idea of assigning auxiliary weights to the tree edges, in a way that the minimum path length from each node to the root node, rounded to the floor, will correspond to the node's SO. The SO of a segment is computed as the mode of the SO of its nodes.

5. Judging from the images in Figure 6A, it appears that fewer spine necks were reconstructed with the SENPAI algorithm compared to the manual segmentation. Given the importance of the spine neck width in determining the electrical compartmentalization of the spine head (e.g., Tonnesen et al., Nat. Neurosci. 2014), it would be interesting to know whether the assignment of the spine necks could be

improved.

We agree with the reviewer in remarking the importance of spine neck reconstruction. Nonetheless, resolution limits often impair the optimal reconstruction of spine necks, impacting on spine detection and morphological characterization as well. With the new analyses added in the current version of the manuscript, we show that spine reconstruction is a challenging task for all of the tested algorithms: SENPAI can't segment spine necks below a certain resolution, but neither can the other algorithms, unless by over-segmenting.

One possible solution is of course to increase the magnification: as we report in the supplementary materials, section 13, we tested SENPAI on higher-magnification images, showing SENPAI ability to correctly assign spines and reconstruct necks. We report here Figure S11 depicting the achieved results, along with its caption.

Figure S11: SENPAI's outcome on high resolution images of dendritic spines. **A)** Original data, rat pyramidal hippocampal neuron transfected with a CMV-membrane GFP plasmid. GFP was then amplified with a GFP immunochemistry and revealed with Alexa594. **B)** Global SENPAI reconstruction of A). **C)** Region of A acquired with confocal at 93x **D)** Region of A acquired with STED at 93x. **E)** SENPAI reconstruction (depth color-coded) on confocal data in C) **F)** SENPAI reconstruction (depth color-coded) on STED data in D). **G)** In-depth detailed zoom of SENPAI reconstruction on confocal data 93x slice. **H)** In-depth detailed zoom of SENPAI reconstruction on STED data 93x (slice). Fine details are shown by orange arrows.

However, increasing magnification is not always possible, and often results in a smaller field of view. In this light, we underline that SENPAI is developed to face the challenge of spine assignment even when image resolution does not allow to reconstruct necks. This is implemented in the parcellation step thanks to the combination of image morphological reconstruction and the watershed transform.

We added in the discussions a sentence on how the problem of neck tracing at the diffraction limit could be addressed in future works.

[...] be they branches or spines. We provide this versatile feature in SENPAI.

SENPAI best contribution to the scientific community is the ability to deal with dense scenarios from cleared samples, a context that is particularly challenging for existing algorithms, even though it was proven robust in the segmentation of single-cell scenarios, with different cell morphologies and imaging modalities. Moreover, our algorithm allows for reconstructing large groups of neurons down to the single spine level with a single, unitary approach across different scenarios. This result is achieved thanks to the upgrade of the k-means clustering approach using the topological information at different scales, along with proper class selection criteria for neural structure identification. Another relevant novelty is the development and integration with a parcellation step, based on watershed transform of the intensity image. This allowed to successfully isolate single neurons in different scenarios as well as perform an automatic assignment of spines to the relevant dendrite. Indeed, increasing magnification is often unfeasible, and it comes at the cost of shrinking the field of view. In these cases, spine neck segmentation is impaired by resolution limits. Using the watershed transform on the intensity image, we provide now an efficient parcellation step for the automatic assignment of spines. Spine assignment is a crucial step preliminary to neck reconstruction. The latter can be achieved either using model-based [Erdil et al., 2019; Jain et al., 2021; Rodriguez et al., 2008] or data-driven [Cheng et al., 2007; Su et al., 2014] approaches.

Minor Comments:

1. Running the test script only produced 8 cells in the final visualization instead of the 10 cells shown in Figure 3. The indices of the two remaining neurons appear to be missing in the test_script file.

We apologized with the reviewer for the inconvenience. We updated the test_script.m file in order to produce the 10 cells of Figure 3.

2. The manuscript doesn't contain any information about the hardware used to run the code. It would also be helpful to know how the algorithm scales with the image resolution/the number of stacks.

We report now in section 14 of the Supplementary Material the following:

14. Time to run the algorithm

On our test_data.tif dataset, i.e., a 512-by-512-by-143 voxels 3D stack:

- RAM 16 GB, processor Intel® Core™ i7-10750H CPU @ 2.60GHz: segmentation+parcellation in 804 seconds.

- RAM 16 GB, processor Intel® Core™ i7-1165G7 CPU @ 2.80GHz: segmentation+parcellation in 1000 seconds.
- RAM 32 GB, processor Intel® Core™ i7-12700K CPU @ 3.60GHz: segmentation+parcellation in 360 seconds.

3. What is the rationale for using 6 classes in the k-means clustering? Were these determined empirically, and would the ideal number of classes potentially differ for different cell types or microscopic parameters?

We thank the reviewer for this comment. The rationale behind such a choice is based on the following considerations:

- *Each class in the k-means outcome encodes a set of image intensity values and second order derivatives.*
- *Neuron external boundaries are encoded by positive second order derivatives (as detailed in Figure 2 of the Manuscript).*

In the light of this, we assumed that a good value for k would be the smallest one that allows to have a class encoding positive values of the second order derivatives for each direction in the 3D space (i.e., x, y, z). Empirically, we observed that having a separate class for each of the three axes led to better results. In this way, we ensured that for each direction there is a class encoding outer borders of structures, possibly maximizing the possibility of separating close-by neurons.

By adopting such a criterion, we obtain $K = 6$ for one 40x and the 93x test datasets and $K=7$ for the other 40x test dataset.

However, since this value may vary based on the image type (as the reviewer suggests), we updated the SENPAI routines to automatically estimate the K parameter for each new dataset.

We added details about the definition of the desired K by adding to the Materials and Methods section:

Each 3D structure defined by a high-intensity signal on a low-intensity background is characterized by intensity gradients at the boundaries, coded in the second derivative by a positive peak at the external boundary and a negative peak at the internal one. A key feature of SENPAI's K-means clustering is given by the choice of the number of classes K, and by how such classes are interpreted and used for the segmentation. The algorithm iteratively searches for the smallest K that allows to have (i) a class encoding positive values of the second order derivatives for each direction in the 3D space (representing the outer borders of the neuron structures), (ii) each direction (x, y, z) is encoded in a different class. Afterwards, for a given number of classes K, SENPAI identifies neurons by sorting and choosing the obtained clusters. Sorting is based on average pixels' intensity values of each cluster, from the lowest to the highest. Then, the clusters associated with the neuronal structures are given by all the clusters with the highest average pixel intensity values and with all the second order derivatives whose values are negative (these clusters represent signal from the neuron body see Figure 2B-C).

4. The microscopic parameters used to generate the test datasets should be described

in more detail. What were the acquisition parameters for the 40x confocal dataset? What was the step size used for the confocal and STED datasets? Was Nyquist sampling used?

We provided more detail on the microscopic parameters used to generate the test datasets by adding in the Methods section the following:

Confocal and STED images were acquired with a confocal laser scanning microscope LEICA SP8 STED 3DX equipped with a 20x and 93x/1.3 NA glycerol immersion objective or a 40x NA1.3 oil objective and with 3 hybrid detectors (HyDs). The specimens were imaged with a pulsed white-light laser and a pulsed 775 nm depletion laser to acquire nanoscale imaging. Depletion laser power balance (between 2D and 3D) is adapted depending on transparency sample laser to optimize resolution and preserve signal in depth.

Confocal Image of rat hippocampal pyramidal cell in culture was taken with 40X NA 1.3 Leica oil objective with a pixel size of (91,41 nm x 91,41 nm x 280 nm).

3D-STED was done with a 93X NA1.3 glycerol objective. Images were averaged 16 times in line and acquired with a magnification zoom >3. Excitation laser was adjusted to avoid any saturated pixel. Rat hippocampal pyramidal cells in culture were acquired with voxel size of 40x40x130 nm within a Z stack of 21 slices. Cerebellum Purkinje cells were acquired with voxel size of 62x62x60 nm within a Z stack of 35 slices.

5. Figure 5 could be improved by only displaying the Purkinje cell data from Vormberg et al., perhaps even in the same plot as the data obtained by the different segmentation algorithms.

Unfortunately, we did not manage to obtain from Vormberg et al. the numerical results, therefore the only complete, unmodifiable figure is at our disposal. In this scenario, being the figure in Vormberg et al. already dense, we prefer to maintain a division between the visualization of their and our results. Please note that we updated Figure 5 with the results obtained using Ilastik, therefore we believe that adding the Purkinje cell data from Vormberg in the same plot as the data obtained by the different segmentation algorithms would make the latter much too dense. We attach below the updated figure.

We updated accordingly the caption:

Figure 5 –A) Schematization of SO and of the meaning of some of the extracted features; **B)** Schematization of the validation pipeline; **C)** Graphical comparison of the SO of Purkinje cells found in literature and measured on the neurons segmented with SENPAI and the state-of-art tools. The parameters based on the SO reported by Vormberg et al. on six types of neural cells (left, edited from Vormberg et al.) and computed on 27 Purkinje neuronal cell segmented with SENPAI (magenta), HK-Icy (blue), Ilastik (green), NeuroGPS (red) and NeuTube (gray) from 40x images (right). It should be noted that the parameters, with the exception of the Branch Bifurcation Ratio, were computed only for neurons whose SN was as equal to its mode across all example neurons, in line with Vormberg et al.

Reviewer #3

The authors discuss the interesting problem of neuron segmentation at cellular and sub-cellular scales in microscopy images of brain tissues. The topic is for possible interest for the journal reader. The paper is well structured. The used reference list is proper.

The state of art in the field must be realized in a more critical way

We added a new paragraph in the introduction section, detailing the state of the art for what regards tracing and segmentation algorithms for neuronal reconstruction.

[...] while maintaining low laser power [Magliaro et al., 2016].

Parallel to the advances in neural tissue labeling and imaging at both the micro and nanoscale, automatic algorithms aiming at obtaining faithful three-dimensional (3D) reconstruction of single neurons have been extensively developed. Recent reviews on the topic [Liu et al., 2022; Magliaro et al., 2019] provide a categorization point for more than forty algorithms (see Table 1 in the supplementary materials), each one exploiting key aspects of image properties (e.g., signal-to-noise ratio, point-spread-function, image contrast) and/or neuron features (e.g., axon tubularity, soma sphericity, tree-like structure) for developing their own neuron reconstruction strategy. As such, their success is strongly influenced by both the acquisition modality and cell type and by the kind of features they rely on (e.g., **local methods** based on a progressive propagation of the tracing or segmentation from a reference point are particularly sensitive to noise or local inhomogeneity, while **global methods** based on SNR will fail at handling smooth changes in SNR across the image stack) to the point that developing general-purpose solutions is often impracticable. Some of these issues are partially solved by **meta methods**, i.e. modules that can be used on top of base tracing or segmentation algorithms to solve specific problems and improve the result (e.g., G-cut [Li et al., 2019] for post-hoc separation of interweaving neurons and UltraTracer [Peng et al., 2017] for increasing the scalability of tracing algorithms to large image stacks).

Similar considerations apply to methods based on Artificial Intelligence (AI) approaches such as **machine learning** and **deep learning** (for a summary categorization see Tables 3 and 5 in [Chen et al., 2023]). For the same motivation, convolutional neural networks were recently introduced [Ghahremani et al., 2021; Mazzamuto et al., 2018; Nourbakhsh et al., 2018]. Indeed, AI approaches are rapidly emerging as a solution to the complex problem of neuronal reconstruction in microscopy images. Their direct application to previously unseen data (e.g., different acquisition modality, cell type, SNR levels) strongly depends on the training set (e.g., type of data, number of training examples), or ad-hoc solutions (e.g., **transfer learning** [von Chamier et al., 2021; Stuckner et al., 2022]), limiting their generalizability. Although considerable effort is now being made to achieve explainability in deep learning models [Liu and Xu, 2023], the black-box nature of deep-learning based algorithms makes it difficult for users to exercise control over common segmentation issues such as over- or under-segmentation. In addition, existing algorithms based on deep-learning currently struggle in dealing with separating multiple structures within an image block [Chen et al., 2023] and to retrieve information beyond the limits posed by the resolution of the image, i.e., for spine morphology or spine neck reconstruction. Finally, independent of the method used, manual correction and fine-tuning of the automatic results by experts are still required [Manubens-Gil et al., 2023].

The poor generalizability of neuron reconstruction algorithms' is even lower when dealing with images from densely packed neurons, in particular those obtained from cleared tissues [Magliaro et al., 2019], which represent a novel class of previously unseen neuroimaging data with unique features in terms of both image quality (e.g., enhanced signal intensity, SNR, CNR) and the cell density in a 3D arrangement [Callara et al., 2020; Magliaro et al., 2019; Pesce et al., 2022; Ueda et al., 2020]. Indeed, algorithms purposely developed to deal with cleared tissues and/or dense scenarios are more scarce [Callara et al., 2020; Li et al., 2019; Milligan et al., 2019;

Quan et al., 2016]. Moreover, the outcomes of such algorithms and tools are compared with a manually-segmented ground truth reconstruction, which is time consuming, and, particularly when dealing with dense-packed neurons, difficult to achieve and prone to human bias.

This paragraph was integrated in the Supplementary Materials with a Table (Table S1) summarizing the automatic segmentation and tracing algorithms cited by [Magliaro et al., 2019] and [Liu et al., 2022]:

Algorithm	Authors	Doi
ViterBrain	Athey et al. (2022)	10.1038/s42003-022-03320-0
RPCT	Bas and Erdogmus (2011)	10.1007/s12021-011-9105-2
SmartTracing	Chen et al. (2015)	10.1007/s40708-015-0018-y
Neural Circuit Tracer	Chothani et al. (2011)	10.1007/s12021-011-9121-2
Active learning of neuron morphology for accurate automated tracing of neurites	Gala et al. (2014)	10.3389/fnana.2014.00037
CAAT	Huang et al. (2021)	10.3389/fnana.2021.712842
ORION2	Jiménez et al. (2015)	10.1007/s12021-014-9256-z
ShuTu	Jin et al. (2019)	10.3389/fninf.2019.00068
SparseTracer	Li et al. (2017)	10.1371/journal.pone.0182184
Rivulet2	Liu et al. (2018c)	10.1109/TMI.2018.2833420
ORION	Losavio et al. (2008)	10.1152/jn.90627.2008
fNeuronTool	Ming et al. (2013)	10.1371/journal.pone.0084557
APP1	Peng et al. (2011)	10.1093/bioinformatics/btr237
UltraTracer	Peng et al. (2017)	10.1038/nmeth.4233
NeuroGPS-Tree	Quan et al. (2016)	10.1038/nmeth.3662
PHD	Radojević and Meijering (2017a)	10.1093/bioinformatics/btw751
PNR	Radojević and Meijering (2019)	10.1007/s12021-018-9407-8
NeuronStudio	Wearne et al. (2005)	10.1016/j.neuroscience.2005.05.053
Open-Curve Snake	Wang et al. (2011)	10.1007/s12021-011-9110-5
ENT	Wang et al. (2017)	10.1007/s12021-017-9325-1
DiMorSC	Wang et al. (2018)	10.1101/321489
MOST	Wu et al. (2014)	10.1016/j.neuroimage.2013.10.036
APP2	Xiao and Peng (2013)	10.1093/bioinformatics/btt170
SimpleTracing	Yang et al. (2013)	10.1186/1471-2105-14-93
FMST	Yang et al. (2019)	10.1007/s12021-018-9392-y
MDL constrained 3-D grayscale skeletonization	Yuan et al. (2009)	10.1007/s12021-009-9057-y
neuTube	Zhao et al. (2011)	10.1007/s12021-011-9120-3
Neuron Crawler	Zhou et al. (2015b)	10.1109/ISBI.2015.7164009
TReMAP	Zhou et al. (2016)	10.1007/s12021-015-9278-1
neurolucida	Glaser et al. (1990)	10.1016/0895-6111(90)90105-k
mansegtool	Magliaro et al. (2017)	10.3389/fninf.2017.00036
tree2tree	Basu et al. (2013)	10.1109/TITB.2012.2209670
trees	Cuntz et al. (2010)	10.1371/journal.pcbi.1000877

g-cut	Li et al. (2019)	10.1038/s41467-019-09515-0
-------	------------------	----------------------------

The novelty of the proposed algorithm must be highlighted in comparison with similar algorithms.

We added a sentence with the goal of better clarifying the novelty of the proposed algorithm:

[...] be they branches or spines. We provide this versatile feature in SENPAI.

SENPAI best contribution to the scientific community is the ability to deal with dense scenarios from cleared samples, a context that is particularly challenging for existing algorithms, even though it was proven robust in the segmentation of single-cell scenarios, with different cell morphologies and imaging modalities. Moreover, our algorithm allows for reconstructing large groups of neurons down to the single spine level with a single, unitary approach across different scenarios. This result is achieved thanks to the upgrade of the k-means clustering approach using the topological information at different scales, along with proper class selection criteria for neural structure identification. Another relevant novelty is the development and integration with a parcellation step, based on watershed transform of the intensity image. This allowed to successfully isolate single neurons in different scenarios as well as perform an automatic assignment of spines to the relevant dendrite. Indeed, increasing magnification is often unfeasible, and it comes at the cost of shrinking the field of view. In these cases, spine neck segmentation is impaired by resolution limits. Using the watershed transform on the intensity image, we provide now an efficient parcellation step for the automatic assignment of spines. Spine assignment is a crucial step preliminary to neck reconstruction. The latter can be achieved either using model-based [Erdil et al., 2019; Jain et al., 2021; Rodriguez et al., 2008] or data-driven [Cheng et al., 2007; Su et al., 2014] approaches.

The general workflow of the proposed method must be described in more detail. In present form a reader can not reproduce the algorithms.

Figure 2 was modified to better illustrate each step of the workflow, by using the same source image throughout, by including the final segmentation in panel D and the grayscale image used for the parcellation step in panel E.

The caption was modified accordingly:

Figure 2 – the SENPAI algorithm. **A)** Test dataset: 40x confocal (left) and 93x STED (right) datasets. **B)** The SENPAI rationale, based on the selection of classes displaying negative values of the second derivative along the three main axes. **C)** Rationale for the estimation of the K parameter of the K-means clustering: the selected K is the one for which the histograms of second-order derivatives show three different classes achieving maximal average value: here, from the histogram of D²x (second derivative along the x axis), class 3 (cyan) encodes outer borders along the x direction, as it is the only class with values clearly above 0; similarly, class 4 encodes outer borders along the y direction (D²y histogram); class 2 along the z direction (D²z histogram); class 1 encodes both low- and high-intensity homogeneous image portions, and is labelled as background along with classes 2, 3 and 4. **D)** SENPAI workflow - Step 1: K-means clustering, performed on the unsmoothed image (top) and optionally, in parallel, on the image smoothed with a 3D gaussian filter (below). Class selection is performed independently on K-means classes (color-coded as in panel C) for each clustering level. Resulting binarized images (green for clustering level 1, pink for clustering level 2, white for the overlap) are merged by logic OR. **E)** SENPAI workflow - Step 2: the segmented image is parcellated using morphological reconstruction and 3D watershed transform computed on the morphologically reconstructed grayscale image and applied to the binary segmentation. **Left:** isolation of connected structures belonging to the same neuron: soma markers (yellow, placed by user) define wells for the catchment basins (edges in gray). **Right:** connection of a neural portion (e.g., a neuronal branch) to smaller clusters (i.e., dendritic spines). Groups of neuronal clusters assigned to a single neuronal entity are displayed in different colors in their relative catchment basin (grey).

In addition, we provided further details in the Materials and Method section, clarifying in particular how we marked the somata and listing the functions that we used and that can be found in the toolbox shared folder. We report here the modified section:

SENPAL is an automated tool implemented in Matlab (The MathWorks-Inc., United States) and released as a freely available toolbox at: <https://drive.matlab.com/sharing/b815586c-cd0d-413d-8ce4-14712bca33ba>. It can be used to reconstruct neuronal morphology, from the spines up to the whole neuron arborization, starting from 3D optical images from any type of microscope including confocal, multiphoton, STED, etc. It implements two fundamental steps: a segmentation step, based on topology-Informed K-means, and a parcellation step.

Segmentation step

The first step to obtain faithful three-dimensional neuron reconstructions is to isolate neurons from background. SENPAL tackles this step through its main core segmentation algorithm, a topology-informed K-means clustering. Briefly, the K-means partitions n observations (X_1, \dots, X_n) defined on a d -dimensional space into K clusters, by minimizing the within-cluster variances [Lloyd, 1982]. In SENPAL, such space is made of $d=4$ features of interest, comprising the pixels' intensity and second order spatial derivatives computed along the three main cartesian axes, while the n observations are represented by the total number of pixels entering the K-means. Since K-means convergence may be affected by initial conditions, additional hyperparameters, such as the number of replicates and the maximum number of iterations to reach convergence are included (in SENPAL, replicates are set to 10, and for each replicate, a maximum of 1000 iterations is allowed). Moreover, since the procedure may be computationally expensive, this step can be also run on smaller dataset crops, whose size is user-defined, loaded, and processed sequentially to limit the usage of memory.

Each 3D structure defined by a high-intensity signal on a low-intensity background is characterized by intensity gradients at the boundaries, coded in the second derivative by a positive peak at the external boundary and a negative peak at the internal one. A key feature of SENPAL's K-means clustering is given by the choice of the number of classes K , and by how such classes are interpreted and used for the segmentation. The algorithm iteratively searches for the smallest K that allows to have (i) a class encoding positive values of the second order derivatives for each direction in the 3D space (representing the outer borders of the neuron structures), (ii) each direction (x, y, z) is encoded in a different class. Afterwards, for a given number of classes K , SENPAL identifies neurons by sorting and choosing the obtained clusters. Sorting is based on average pixels' intensity values of each cluster, from the lowest to the highest. Then, the clusters associated with the neuronal structures are given by all the clusters with the highest average pixel intensity values and with all the second order derivatives whose values are negative (these clusters represent signal from the neuron body see Figure 2B-C).

An optional parameter is the size of the pre-filtering kernel, which represents one of the salient features of SENPAL and allows to focus on multiple levels of detail of the image. First, the dataset is filtered using a 3D Gaussian filter and derivatives are computed. After the pre-filter step, the K-means clustering is performed on the pixels' intensity of the unsmoothed dataset, and second order spatial derivatives computed on the smoothed image. This pre-filter step can be also progressively performed twice in a row by using two smoothing levels. This option allows to capture structures at different scales, since it eliminates spurious and unwanted boundaries, e.g., those within thick branches or big somata. When multiple levels of smoothing are used, the resulting segmentations are merged by union to compose the final segmentation (Figure 2D).

However, since inhomogeneities in pixel intensity within big neuron structures (e.g., the soma) cannot be all smoothed, and may be identified as boundaries - leading to false negatives -, we implement a function for filling such holes within the segmentation. Similarly, we eliminate spurious small clusters (smaller than 7 voxels, the smallest 3D symmetrical structure) detected from inhomogeneities in the background.

Parcellation step

The second step of the algorithm aims at isolating single neurons from the foreground identified via the K-means clustering. To this end, SENPAL performs a parcellation procedure based on the watershed transform. This defines

a catchment basin for each structure, that allows to mark what belongs to a particular structure and what does not. The parcellation has a two-fold purpose: when dealing with single neuron reconstruction, it splits neurons that may have been merged in a single structure, while when dealing with spine segmentation, it groups together disconnected structures. Thus, the parcellation is performed through two different functions.

The catchment basins are produced for each marked core. The cores are defined automatically in the super-resolution case, by imposing a threshold on the size of the clusters composing the segmented image. While big clusters are labeled as cores, smaller clusters are labeled as spines to be assigned to such cores. On the other hand, for the standard resolution microscopy, cores correspond to somata. These can be marked by the user thanks to a GUI implemented in SENPAI.

Once the cores are defined, the original grayscale dataset is filtered with a 3-by-3-by-3 median filter, and its complement obtained. A morphological reconstruction is applied on this latter, so that the regional intensity minima outside the cores are eliminated. A watershed transform produces one catchment basin for each remaining regional minimum.

The parcellation can be modified *a posteriori*, thanks to a routine to visually assess the quality of the segmentation neuron-by-neuron, and to set additional markers on spuriously assigned branches. Incorrect assignments are usually caused by the presence of branches of neurons whose soma lies outside the acquired dataset. Operationally, this is done thanks to a function within the SENPAI toolkit, implemented to correct the parcellation by adding to the mask of the somata some additional clusters. These will allow the definition of separate catchment basins for the branches wrongly assigned.

The obtained advantages must be expressed using numerical performance measures. The Conclusions must be completed with these measures of advantages.

Regarding numerical performance measures, for segmentations obtained from 93x STED we added Dice's coefficient to quantify under/oversegmentation with respect to the manual gold standard. We could not produce the same quantification on 40x images as we already discussed on the impossibility to obtain convincing gold standard segmentations. For this reason, Strahler metrics have been used to evaluate the reconstructions on 40x images. On reconstructions on 93x STED images, we clearly show using Dice's coefficient that SENPAI outperforms both Ilastik and HK-Icy for all 5 tested neuronal segments, both considering the whole segment or the only spines. In addition, we updated the table in Figure 6, reporting the number of spines manually detected for each neuron, and the number of False Negatives, False Positives and True Positives for SENPAI, Ilastik and HK-Icy to compare the three algorithms in terms of Sensitivity and Precision in the detection of spines. In general, SENPAI achieves better results.

We report below the modified Figure 6:

The caption was modified accordingly.

Figure 6 – Segmentations of 5 dendritic branches (from A to E) and their localization within a 93x STED dataset. A) On the left, we highlight in red on a 3D multi-stack STED image the dendritic branches A to E contained within a single stack and employed here for the validation of SENPAI. On the right, we compare the segmentations obtained for these branches with the state-of-the-art algorithms. The segmentations were performed with SENPAI (1st column), HK-Icy (2nd column), Ilastik (3rd column) and ManSegTool (4th column). For SENPAI, HK-Icy and Ilastik, the segmentations of the whole image are overlaid in gray, while the parcellation outcomes are highlighted in red. Note that ManSegTool is used to manually segment neurons, so the neighboring neurons are not displayed in grey. **B)** Quantitative comparison for the segmentations of the 5 dendritic branches; area (left) and volume (right) for the segmentations obtained with SENPAI (magenta), Ilastik (green) and HK-Icy (blue). The values are normalized with respect to manual segmentations. HK-Icy and Ilastik were integrated with SENPAI parcellation step to, respectively, assign spines to the dendrite and separate touching dendrites from each other. **C)** Quantitative comparison for the segmentations of the 5 dendritic branches; DICE coefficient was used to compare the segmentations obtained with the three algorithms against the manual segmentation, considering the whole neuron (top) and only spines (bottom). Ideal algorithm would give a DICE coefficient of 1. The analysis on the spines was conducted by masking out the manual segmentation of the dendrite, both as it is and with 4 levels of dilation, to account for effects from oversegmentation of the dendritic branch. **D)** Table summarizing performances in terms of spine detection. Spines were counted in the manual segmentation, then, with respect to the manual segmentation, we defined for SENPAI, HK-Icy and Ilastik the number of False Positives and False Negatives, True Positives, the Sensitivity and the Precision. FP=false positives [count]; FN=false negatives [count]; TP=true positives [count]; S%=sensitivity [%of TP over TP+FN]; P%=precision [% of TP over TP+FP].

In the Results section, we modified the paragraph “Segmentation of super-resolution datasets”, as reported below:

Further analyses were performed to compare the branch areas and volumes obtained from the automatic tools with those obtained with the manual segmentation. We observed consistent differences between SENPAI and HK-Icy and Ilastik. While Ilastik delivers areas and volumes (Figure 6B) that are significantly larger than those

from the manual segmentations (area normalized with respect to the ground-truth, median± median absolute deviation (MAD) =1.45±0.42, volume normalized with respect to the ground-truth median±MAD = 1.91±1.08), SENPAI and HK-Icy's outcomes are very close to ManSegTool's (SENPAI area median±MAD 0.88±0.30, SENPAI volume median±MAD 0.86±0.49, HK-Icy area median±MAD 0.83±1.00, HK-Icy volume median±MAD 0.75±1.13). Particularly relevant is the analysis based on the Dice's coefficient [Dice, 1945] (**Figure 6C**), which quantifies the similarity between shapes. Considering the whole neuron branches, SENPAI achieves the highest values for all the five of them, while Ilastik performs worst for all neural branches but one (SENPAI Dice median±MAD 0.81±0.02, HK-Icy Dice median±MAD 0.78±0.11, Ilastik Dice median±MAD 0.68±0.05). We performed the same analysis with a focus on spines, excluding those voxels that were labelled as belonging to the dendrite by manual segmentation. To avoid bias introduced by eventual oversegmentation of the dendrite, we repeated the analysis with the dendrite mask being recursively dilated, and reported all the points in Figure 6C, bottom. SENPAI performs best again and is matched only for neuron E by HK-Icy (for the first iteration, SENPAI Dice median±MAD 0.64±0.07, HK-Icy Dice median±MAD 0.50±0.14, Ilastik Dice median±MAD 0.43±0.10). Merging information in Figure 6C, Ilastik clearly over-segments while HK-Icy under-segments.

Further comparisons in terms of spine detection are reported in **Figure 6D**. We compared spine-by-spine the reconstructions obtained with SENPAI, HK-Icy and Ilastik with the manual segmentation. We counted for each of the three algorithms the number of true positives, false positives and false negatives and estimated specificity and precision. While producing slightly lower specificity with respect to Ilastik, due to higher false negatives, SENPAI performs best in terms of precision. On average, SENPAI is better than HK-Icy on all topological and morphological metrics and performs better than Ilastik in terms of morphological reconstruction and number of false positives.

Other occurrences of "Figure 6B" and "Figure 6C" have been similarly corrected.

In the discussions, we modified the comments on spine segmentation as reported below:

For the 93x datasets obtained with STED, the results - in terms of assigned spines to the correct dendritic branch - were compared with manual segmentations performed with ManSegTool. All the approaches tested accurately reconstruct the neuron branch of interest (**Figure 6A**), and in particular for SENPAI, in 5 out of 6 cases, it provides faithful information about the morphology of the dendrites and their own corresponding spines. Ilastik provides a clear over-segmentation with respect to the manual ground truth, thus resulting in false positive assignments (**Figure 6A, 6D**) and over-estimated areas and volumes both at the whole branch and at the spines level (**Figure 6B**), while SENPAI correctly segments and assigns the spines in most of the cases investigated (**Figure 6A, C**) and provides estimates for area and volume in line with the manual ground-truth (**Figure 6B**).

The algorithm is used for the particular case of Purkinje cells. How about other type of cells?

We could not produce new images from cleared samples on our own. Nonetheless, we applied the SENPAI algorithm to independent datasets including human pyramidal neurons, human cultured excitatory neurons, mouse cultured excitatory neurons, and mouse retinal neurons. In the results section, we now state that SENPAI was successfully tested on these types of cells.

Generalization to other neuron types and imaging modalities

To support the generalizability of our framework, we tested the algorithm on multiple neuron types (from rat, mice and human), including tissues and cultured cells, in uncleared samples and acquired with different imaging

modalities – see supplementary materials. The neuron types we segmented include human pyramidal neurons, mouse pyramidal neurons, human cultured excitatory neurons, mouse cultured excitatory neurons, and mouse retinal neurons. The imaging modalities, other than confocal and confocal STED, include 2P microscopy. We report both quantitative and qualitative evaluations, that suggest the generalizability of our reconstruction algorithm across different imaging modalities and neuron types.

Among the tested datasets of native (uncleared) tissue tested with SENPAI, 34 belong to the BigNeuron GOLD166 dataset. On the BigNeuron bench-testing platform [Manubens-Gil et al., 2023; Peng et al., 2010; Peng et al., 2014a; Peng et al., 2014b; Peng et al., 2015], among 35 of the most popular state of the art algorithms, SENPAI scored in the top-three for five metrics over eight, and in the top-eleven for the remaining three metrics (see section 10 in the supplementary materials).

Another unclassified tested image is the Neuro-GPS test dataset made available by the authors of Neuro-GPS (see section 11 in the Supplementary Materials). Furthermore, at the INSERM facilities, four more datasets were produced: two datasets including in their field of view a whole labelled neuron, and two higher-resolution datasets from confocal and 3D-STED imaging the same dendritic branch with its spines. The results obtained from these four datasets are presented in sections 12 and 13, and Figures 10 and 11 in the Supplementary Materials.

In the supplementary materials, we report the new results in detail. We redirect the reviewer to sections 10, 11, 12 and 13 of the Supplementary Material. We report below Figure S10 as an example for the new achievements.

Figure S10: SENPAI's outcome on two 3D stacks of cultured hippocampal pyramidal cells (pixel size 91,41 nm x 91,41 nm x 0.28nm). A-B) Original Image Maximum projection for the GFP channel. **C-D)** segmentation obtained with SENPAI with depth color-coded (cold colors indicate deeper planes). **E-F)** Skeletonizations of SENPAI segmentations produced with NeuTube.

Which are the limitations or disadvantage of the method in other cases?

We modified a paragraph in the Discussion section to elaborate on possible limitations for our segmentation algorithm.

The performance of SENPAI was assessed on cleared datasets of Purkinje cells imaged with confocal and 3D STED modalities. This scenario was chosen as a proof of concept since Purkinje cells are known to be highly densely arranged within the murine cerebellum. Moreover, SENPAI has been successfully tested on non-cleared samples acquired with multiple widespread acquisition modalities, such as confocal, 3D STED and 2P, and including different neuron types. Nonetheless, several other imaging modalities are in use within the scientific community, as well as labelling modalities that are specific to some parts of the neuron, e.g., membright for the membranes [Collot et al., 2019] or phalloidin for spine heads [Hotulainen and Lappalainen, 2006]. In all the cases where local inhomogeneities characterize neuronal structures, e.g., for non-uniform distribution of staining across the cell, SENPAI might show a suboptimal behaviour. Specifically, the use of spatial derivatives, that are sensitive to image intensity changes, might lead to hollow 3D segmentations or poorly connected reconstruction of the dendrites. In some cases, such as those presented in the supplementary material, the 3D spatial smoothing preprocessing step allowed to increase neural structure homogeneity and consequently the behaviour of second order derivatives. Further work, as for instance the use of other preprocessing steps, might thus be needed to improve the generalizability of SENPAI to specific staining and imaging modalities.

Other future developments will include the test of SENPAI to cleared samples from other brain regions, to fully evaluate the potential of the proposed framework. We will also explore the possibility of estimating the path of spines necks even when SENPAI is not able to connect the spine with the main neural body and to provide the user with the possibility of characterizing spine population in terms of density and morphology.

OTHER EDITS

We added an “Author Contributions” section.

Added a sentence in the Introduction:

In the light of these challenges, we have developed a novel framework for extracting faithful morphological information from brain tissues at the neuronal and sub-neuronal level via imaging and image processing tools. Specifically, we developed a framework for processing and acquiring brain tissue, and to return large volume datasets representing neurons with an improved image quality. Then, we present a data-driven approach – SENPAI (SEgmentation of Neurons using PArtil derivative Information) [...]

We updated the abstract as follows:

The development of robust tools for segmenting cellular and sub-cellular neuronal structures lags behind the massive production of high-resolution 3D images of neurons in brain tissue. The main challenges are related to the high neuronal density and low signal-to-noise characteristics in thick samples, as well as the heterogeneity of data acquired with different imaging methods. To address this issue, we designed a framework which includes sample preparation for high resolution imaging and image analysis. Specifically, we set up a method for labeling thick samples and developed SENPAI, a new scalable algorithm for segmenting neurons at cellular and sub-cellular scales in conventional and super resolution STED microscopy images of brain tissues. Furthermore, we propose a new validation paradigm for testing the segmentation performance when a manual ground-truth may not exhaustively describe neuronal arborization. SENPAI provides accurate multi-scale segmentation, from entire neurons down to spines, outperforming state-of-art tools. The framework will empower image processing of complex neuronal circuitries.

We updated the discussion with the following paragraph:

In conclusion, we report a new two-step segmentation and parcellation method –SENPAI- which can be combined with brain tissue preparation and labelling to isolate neurons from densely packed neighbourhoods and return morphologically faithful segmentations of the cells and their branches at micron and sub-micron scales. The pipeline preserves surface details as well as multi-scale structural complexity, providing enhanced robustness to signal inhomogeneity even across thick samples. We show that SENPAI is superior in performance and robustness with respect to other segmentation algorithms and can be used across scales for different imaging modalities and different types of neuron or brain tissue samples. SENPAI can be exploited as a standalone, open-source framework to support the development of connectome maps and improve our understanding of the brain’s structure and consequently its functional behaviour.

BIBLIOGRAPHY

- Ascoli GA, Donohue DE, Halavi M (2007): NeuroMorpho.Org: A Central Resource for Neuronal Morphologies. *J Neurosci* 27:9247–9251. doi: 10.1523/JNEUROSCI.2055-07.2007.
- Berg S, Kutra D, Kroeger T, Straehle CN, Kausler BX, Haubold C, Schiegg M, Ales J, Beier T, Rudy M, Eren K, Cervantes JI, Xu B, Beuttenmueller F, Wolny A, Zhang C, Koethe U, Hamprecht FA, Kreshuk A (2019): ilastik: interactive machine learning for (bio)image analysis. *Nat Methods* 16:1226–1232. doi: 10.1038/s41592-019-0582-9.

- Bourne J, Harris KM (2007): Do thin spines learn to be mushroom spines that remember? *Curr Opin Neurobiol* 17:381–386. doi: 10.1016/j.conb.2007.04.009.
- Callara AL, Magliaro C, Ahluwalia A, Vanello N (2020): A Smart Region-Growing Algorithm for Single-Neuron Segmentation From Confocal and 2-Photon Datasets. *Front Neuroinform* 14. doi: 10.3389/fninf.2020.00009.
- von Chamier L, Laine RF, Jukkala J, Spahn C, Krentzel D, Nehme E, Lerche M, Hernández-Pérez S, Mattila PK, Karinou E, Holden S, Solak AC, Krull A, Buchholz T-O, Jones ML, Royer LA, Leterrier C, Shechtman Y, Jug F, Heilemann M, Jacquemet G, Henriques R (2021): Democratising deep learning for microscopy with ZeroCostDL4Mic. *Nat Commun* 12:2276. doi: 10.1038/s41467-021-22518-0.
- de Chaumont F, Dallongeville S, Chenouard N, Hervé N, Pop S, Provoost T, Meas-Yedid V, Pankajakshan P, Lecomte T, Le Montagner Y, Lagache T, Dufour A, Olivo-Marin J-C (2012): Icy: an open bioimage informatics platform for extended reproducible research. *Nat Methods* 9:690–696. doi: 10.1038/nmeth.2075.
- Chen R, Liu M, Chen W, Wang Y, Meijering E (2023): Deep learning in mesoscale brain image analysis: A review. *Comput Biol Med* 167:107617. doi: 10.1016/j.combiomed.2023.107617.
- Cheng J, Zhou X, Miller E, Witt RM, Zhu J, Sabatini BL, Wong STC (2007): A novel computational approach for automatic dendrite spines detection in two-photon laser scan microscopy. *J Neurosci Methods* 165:122–134. doi: 10.1016/j.jneumeth.2007.05.020.
- Citri A, Malenka RC (2008): Synaptic Plasticity: Multiple Forms, Functions, and Mechanisms. *Neuropsychopharmacology* 33:18–41. doi: 10.1038/sj.npp.1301559.
- Collot M, Ashokkumar P, Anton H, Boutant E, Faklaris O, Galli T, Mély Y, Danglot L, Klymchenko AS (2019): MemBright: A Family of Fluorescent Membrane Probes for Advanced Cellular Imaging and Neuroscience. *Cell Chem Biol* 26:600-614.e7. doi: 10.1016/j.chembiol.2019.01.009.
- Dice LR (1945): Measures of the Amount of Ecologic Association Between Species. *Ecology* 26:297–302. doi: 10.2307/1932409.
- Dufour A, Meas-Yedid V, Grassart A, Olivo-Marin J-C (2008): Automated quantification of cell endocytosis using active contours and wavelets. In: . 2008 19th International Conference on Pattern Recognition. IEEE. pp 1–4. doi: 10.1109/ICPR.2008.4761748.
- Erdil E, Ozgur Argunsah A, Tasdizen T, Unay D, Cetin M (2019): Combining Nonparametric Spatial Context Priors With Nonparametric Shape Priors for Dendritic Spine Segmentation in 2-Photon Microscopy Images. In: . 2019 IEEE 16th International Symposium on Biomedical Imaging (ISBI 2019). IEEE. pp 204–207. doi: 10.1109/ISBI.2019.8759273.
- Feher J (2012): Balance and Control of Movement. In: . *Quantitative Human Physiology*. Elsevier. pp 341–353. doi: 10.1016/B978-0-12-382163-8.00037-2.
- Feng L, Zhao T, Kim J (2015): neuTube 1.0: A New Design for Efficient Neuron Reconstruction Software Based on the SWC Format. *eneuro* 2:ENEURO.0049-14.2014. doi: 10.1523/ENEURO.0049-14.2014.
- García-López P, García-Marín V, Freire M (2007): The discovery of dendritic spines by Cajal in 1888 and its relevance in the present neuroscience. *Prog Neurobiol* 83:110–130. doi: 10.1016/j.pneurobio.2007.06.002.
- Ghahremani P, Boorboor S, Mirhosseini P, Gudisagar C, Ananth M, Talmage D, Role LW, Kaufman AE (2021): NeuroConstruct: 3D Reconstruction and Visualization of Neurites in Optical Microscopy

- Brain Images. *IEEE Trans Vis Comput Graph*:1–1. doi: 10.1109/TVCG.2021.3109460.
- Glausier JR, Lewis DA (2013): Dendritic spine pathology in schizophrenia. *Neuroscience* 251:90–107. doi: 10.1016/j.neuroscience.2012.04.044.
- Harris K, Jensen F, Tsao B (1992): Three-dimensional structure of dendritic spines and synapses in rat hippocampus (CA1) at postnatal day 15 and adult ages: implications for the maturation of synaptic physiology and long-term potentiation [published erratum appears in *J Neurosci* 1992 Aug;1. *J Neurosci* 12:2685–2705. doi: 10.1523/JNEUROSCI.12-07-02685.1992.
- Harris KM (2020): Synaptic Odyssey. *J Neurosci* 40:61–80. doi: 10.1523/JNEUROSCI.0735-19.2019.
- Hirano T (2018): Purkinje Neurons: Development, Morphology, and Function. *The Cerebellum* 17:699–700. doi: 10.1007/s12311-018-0985-7.
- Hotulainen P, Lappalainen P (2006): Stress fibers are generated by two distinct actin assembly mechanisms in motile cells. *J Cell Biol* 173:383–394. doi: 10.1083/jcb.200511093.
- Jain S, Mukherjee S, Danglot L, Olivo-Marin J-C (2021): Morphological Reconstruction of Detached Dendritic Spines via Geodesic Path Prediction. In: . 2021 IEEE 18th International Symposium on Biomedical Imaging (ISBI). IEEE. pp 944–947. doi: 10.1109/ISBI48211.2021.9433981.
- Kasai H, Hayama T, Ishikawa M, Watanabe S, Yagishita S, Noguchi J (2010): Learning rules and persistence of dendritic spines. *Eur J Neurosci* 32:241–249. doi: 10.1111/j.1460-9568.2010.07344.x.
- Knott G, Holtmaat A (2008): Dendritic spine plasticity—Current understanding from in vivo studies. *Brain Res Rev* 58:282–289. doi: 10.1016/j.brainresrev.2008.01.002.
- Ledderose J, Senci3n L, Salgado H, Arias-Carri3n O, Trevi3o M (2014): A software tool for the analysis of neuronal morphology data. *Int Arch Med* 7:6. doi: 10.1186/1755-7682-7-6.
- Li R, Zhu M, Li J, Bienkowski MS, Foster NN, Xu H, Ard T, Bowman I, Zhou C, Veldman MB, Yang XW, Hintiryan H, Zhang J, Dong H-W (2019): Precise segmentation of densely interweaving neuron clusters using G-Cut. *Nat Commun* 10:1549. doi: 10.1038/s41467-019-09515-0.
- Liu Y, Wang G, Ascoli GA, Zhou J, Liu L (2022): Neuron tracing from light microscopy images: automation, deep learning and bench testing. Ed. Hanchuan Peng. *Bioinformatics* 38:5329–5339. doi: 10.1093/bioinformatics/btac712.
- Liu Z, Xu F (2023): Interpretable neural networks: principles and applications. *Front Artif Intell* 6. doi: 10.3389/frai.2023.974295.
- Lloyd S (1982): Least squares quantization in PCM. *IEEE Trans Inf Theory* 28:129–137. doi: 10.1109/TIT.1982.1056489.
- Magliaro C, Callara AL, Vanello N, Ahluwalia A (2019): Gotta Trace ‘em All: A Mini-Review on Tools and Procedures for Segmenting Single Neurons Toward Deciphering the Structural Connectome. *Front Bioeng Biotechnol* 7. doi: 10.3389/fbioe.2019.00202.
- Manubens-Gil L, Zhou Z, Chen H, Ramanathan A, Liu X, Liu Y, Bria A, Gillette T, Ruan Z, Yang J, Radojević M, Zhao T, Cheng L, Qu L, Liu S, Bouchard KE, Gu L, Cai W, Ji S, Roysam B, Wang C-W, Yu H, Sironi A, Iascone DM, Zhou J, Bas E, Conde-Sousa E, Aguiar P, Li X, Li Y, Nanda S, Wang Y, Muresan L, Fua P, Ye B, He H, Staiger JF, Peter M, Cox DN, Simonneau M, Oberlaender M, Jefferis G, Ito K, Gonzalez-Bellido P, Kim J, Rubel E, Cline HT, Zeng H, Nern A, Chiang A-S, Yao J, Roskams J, Livesey R, Stevens J, Liu T, Dang C, Guo Y, Zhong N, Tourassi G, Hill S, Hawrylycz M, Koch C, Meijering E, Ascoli GA, Peng H (2023): BigNeuron: a resource to benchmark and predict performance of algorithms for automated tracing of neurons in light microscopy datasets. *Nat Methods* 20:824–

835. doi: 10.1038/s41592-023-01848-5.

- Mazzamuto G, Costantini I, Neri M, Roffilli M, Silvestri L, Pavone FS (2018): Automatic Segmentation of Neurons in 3D Samples of Human Brain Cortex. In: pp 78–85. doi: 10.1007/978-3-319-77538-8_6.
- Milligan K, Balwani A, Dyer E (2019): Brain mapping at high resolutions: Challenges and opportunities. *Curr Opin Biomed Eng* 12:126–131. doi: 10.1016/j.cobme.2019.10.009.
- Nanda S, Bhattacharjee S, Cox DN, Ascoli GA (2020): Distinct Relations of Microtubules and Actin Filaments with Dendritic Architecture. *iScience* 23:101865. doi: 10.1016/j.isci.2020.101865.
- Noguchi J, Matsuzaki M, Ellis-Davies GCR, Kasai H (2005): Spine-Neck Geometry Determines NMDA Receptor-Dependent Ca²⁺ Signaling in Dendrites. *Neuron* 46:609–622. doi: 10.1016/j.neuron.2005.03.015.
- Nourbakhsh F, Abdeladim L, Clavreul S, Loulier K, Beaurepaire E, Livet J, Chessel A (2018): Neural Cell Segmentation in Large-Scale 3D Color Fluorescence Microscopy Images for Developmental Neuroscience. In: . 2018 25th IEEE International Conference on Image Processing (ICIP). IEEE. pp 3828–3832. doi: 10.1109/ICIP.2018.8451702.
- Ofer N, Berger DR, Kasthuri N, Lichtman JW, Yuste R (2021): Ultrastructural analysis of dendritic spine necks reveals a continuum of spine morphologies. *Dev Neurobiol* 81:746–757. doi: 10.1002/dneu.22829.
- Peng H, Bria A, Zhou Z, Iannello G, Long F (2014a): Extensible visualization and analysis for multidimensional images using Vaa3D. *Nat Protoc* 9:193–208. doi: 10.1038/nprot.2014.011.
- Peng H, Hawrylycz M, Roskams J, Hill S, Spruston N, Meijering E, Ascoli GA (2015): BigNeuron: Large-Scale 3D Neuron Reconstruction from Optical Microscopy Images. *Neuron* 87:252–256. doi: 10.1016/j.neuron.2015.06.036.
- Peng H, Long F, Zhao T, Myers E (2011): Proof-editing is the Bottleneck Of 3D Neuron Reconstruction: The Problem and Solutions. *Neuroinformatics* 9:103–105. doi: 10.1007/s12021-010-9090-x.
- Peng H, Ruan Z, Long F, Simpson JH, Myers EW (2010): V3D enables real-time 3D visualization and quantitative analysis of large-scale biological image data sets. *Nat Biotechnol* 28:348–353. doi: 10.1038/nbt.1612.
- Peng H, Tang J, Xiao H, Bria A, Zhou J, Butler V, Zhou Z, Gonzalez-Bellido PT, Oh SW, Chen J, Mitra A, Tsien RW, Zeng H, Ascoli GA, Iannello G, Hawrylycz M, Myers E, Long F (2014b): Virtual finger boosts three-dimensional imaging and microsurgery as well as terabyte volume image visualization and analysis. *Nat Commun* 5:4342. doi: 10.1038/ncomms5342.
- Peng H, Zhou Z, Meijering E, Zhao T, Ascoli GA, Hawrylycz M (2017): Automatic tracing of ultra-volumes of neuronal images. *Nat Methods* 14:332–333. doi: 10.1038/nmeth.4233.
- Penzes P, Cahill ME, Jones KA, VanLeeuwen J-E, Woolfrey KM (2011): Dendritic spine pathology in neuropsychiatric disorders. *Nat Neurosci* 14:285–293. doi: 10.1038/nn.2741.
- Pesce L, Laurino A, Scardigli M, Yang J, Boas DA, Hof PR, Destrieux C, Costantini I, Pavone FS (2022): Exploring the human cerebral cortex using confocal microscopy. *Prog Biophys Mol Biol* 168:3–9. doi: 10.1016/j.pbiomolbio.2021.09.001.
- Quan T, Zhou H, Li J, Li S, Li A, Li Y, Lv X, Luo Q, Gong H, Zeng S (2016): NeuroGPS-Tree: automatic reconstruction of large-scale neuronal populations with dense neurites. *Nat Methods* 13:51–54. doi: 10.1038/nmeth.3662.

- Rasia-Filho AA, Calcagnotto ME, von Bohlen und Halbach O (2023): Introduction: What Are Dendritic Spines? In: pp 1–68. doi: 10.1007/978-3-031-36159-3_1.
- Rodriguez A, Ehlenberger DB, Dickstein DL, Hof PR, Wearne SL (2008): Automated Three-Dimensional Detection and Shape Classification of Dendritic Spines from Fluorescence Microscopy Images. Ed. Bernado Sabatini. PLoS One 3:e1997. doi: 10.1371/journal.pone.0001997.
- Sala C, Segal M (2014): Dendritic Spines: The Locus of Structural and Functional Plasticity. *Physiol Rev* 94:141–188. doi: 10.1152/physrev.00012.2013.
- Scheff SW, Price DA, Schmitt FA, Mufson EJ (2006): Hippocampal synaptic loss in early Alzheimer's disease and mild cognitive impairment. *Neurobiol Aging* 27:1372–1384. doi: 10.1016/j.neurobiolaging.2005.09.012.
- Serrano-Pozo A, Frosch MP, Masliah E, Hyman BT (2011): Neuropathological Alterations in Alzheimer Disease. *Cold Spring Harb Perspect Med* 1:a006189–a006189. doi: 10.1101/cshperspect.a006189.
- Sholl DA (1955): The organization of the visual cortex in the cat. *J Anat* 89:33–46.
- Spires TL (2005): Dendritic Spine Abnormalities in Amyloid Precursor Protein Transgenic Mice Demonstrated by Gene Transfer and Intravital Multiphoton Microscopy. *J Neurosci* 25:7278–7287. doi: 10.1523/JNEUROSCI.1879-05.2005.
- Sternberg (1983): Biomedical Image Processing. *Computer (Long Beach Calif)* 16:22–34. doi: 10.1109/MC.1983.1654163.
- Stuckner J, Harder B, Smith TM (2022): Microstructure segmentation with deep learning encoders pre-trained on a large microscopy dataset. *npj Comput Mater* 8:200. doi: 10.1038/s41524-022-00878-5.
- Su R, Sun C, Zhang C, Pham TD (2014): A novel method for dendritic spines detection based on directional morphological filter and shortest path. *Comput Med Imaging Graph* 38:793–802. doi: 10.1016/j.compmedimag.2014.07.006.
- Tomer R, Ye L, Hsueh B, Deisseroth K (2014): Advanced CLARITY for rapid and high-resolution imaging of intact tissues. *Nat Protoc* 9:1682–1697. doi: 10.1038/nprot.2014.123.
- Ueda HR, Ertürk A, Chung K, Gradinaru V, Chédotal A, Tomancak P, Keller PJ (2020): Tissue clearing and its applications in neuroscience. *Nat Rev Neurosci* 21:61–79. doi: 10.1038/s41583-019-0250-1.
- Vormberg A, Effenberger F, Muellerleile J, Cuntz H (2017): Universal features of dendrites through centripetal branch ordering. Ed. Boris S. Gutkin. *PLOS Comput Biol* 13:e1005615. doi: 10.1371/journal.pcbi.1005615.

Reviewers' Comments:

Reviewer #1:

Remarks to the Author:

Cauzzo et al. provide a new segmentation tool for densely labeled neuronal tissue. They test their algorithm primarily on cerebellar purkinje cells and compare it with existing segmentation algorithms.

The authors successfully revised their manuscript based on the reviewers comments. The two step segmentation algorithm SENPAI, which uses k-means clustering of 1st and 2nd derivative of pixel intensity profiles, is novel and seems to work excellently. In combination with the "parcellation" step the SENPAI is even able to assign dendritic spines to dendrites. SENPAI represents a novel segmentation tool that will be useful and interesting for the research community.

Minor points:

Figure 2E, F: "Parcellation" presumably should mean "dividing into small parcels". However, please check whether the word is used like that and fits to what the authors would like to say. Indeed it is rather an "assignment" of structures (like spines) to another structure (e.g. dendrite). Please consider revision.

L76-81: Very long sentence. Consider revision

L126: "close packed", should be "closely packed"?

L130: "mice line", should be "mouse line"?

L180: "spine head", should be "spine heads"

L181: same for spine neck

L182: "take up"; consider "improve"

Consider using "neuronal" instead of "neural" throughout the text

Figure2: The figure has a panel F that is not referred to, neither in the legend, nor in the text.

L330: "Neuron branches", should be "dendritic branches"?; consider revision throughout the text

L354-358: The authors claim "significantly larger"; they need to provide the statistical test either in the legend of the figure or in the text.

L361-362: Please add a statistic comparison/test for the numbers presented. (See comment above)

L363-369: Statistic test for spine comparisons need also to be added.

The reviewer is still not fully convinced that SENPAI outperforms HK-Icy. From my point of view both algorithms perform similar and might have certain advantages and disadvantages.

Rebuttal letter, reply to reviewer L446: White laser..

Reply: I am disappointed. The authors reply to the reviewer that they used 594nm excitation wavelength of a white pulsed laser and emission was filtered at 600-640 nm. However, they do not provide these details in their manuscript text. Indeed the authors need to provide these details for all imaging modalities they carried out. Their findings will not be reproducible without these specifications.

Chapter "Generalization of other neuron types and imaging modalities"

It would have been informative to include a figure illustrating the results of the chapter. In addition, the paragraph lacks a conclusive sentence at the end.

L488: "reporter mice lines", should be "reporter mouse lines"?

Reviewer #2:

Remarks to the Author:

I appreciate the efforts made by the authors to address my previous concerns. The manuscript has significantly improved, and I support its publication. I only found two minor mistakes that should be fixed:

- 1) In the legend for Figure 2, the description for panel F is mistakenly included in that for panel E.
- 2) In Table S2, row F, the total normalized dendritic length per SO obtained from the Ilastik algorithm aligns with the published range and should be underlined. Conversely, in row G, none of the computed values align with the published values and should therefore not be underlined.

Additionally, I recommend a thorough proofreading of the final manuscript to catch minor language and grammatical errors, such as the repeated use of "state-of-art" instead of the correct "state-of-the-art."

Reviewer #3:

Remarks to the Author:

The authors revised the manuscript in accordance with the reviewer's comments. In my opinion the paper is publishable in present form.

REVIEWERS' COMMENTS

Reviewer #1 (Remarks to the Author):

Cauzzo et al. provide a new segmentation tool for densely labeled neuronal tissue. They test their algorithm primarily on cerebellar purkinje cells and compare it with existing segmentation algorithms. The authors successfully revised their manuscript based on the reviewers comments. The two step segmentation algorithm SENPAI, which uses k-means clustering of 1st and 2nd derivative of pixel intensity profiles, is novel and seems to work excellently. In combination with the "parcellation" step the SENPAI is even able to assign dendritic spines to dendrites. SENPAI represents a novel segmentation tool that will be useful and interesting for the research community.

Minor points:

Figure 2E, F: "Parcellation" presumably should mean "dividing into small parcels". However, please check whether the word is used like that and fits to what the authors would like to say. Indeed it is rather an "assignment" of structures (like spines) to another structure (e.g. dendrite). Please consider revision.

We think that the term "parcellation" fits our scope, and for three reasons. First, it is commonly used in the field, as other software allows for parcellation of the image space, although with different results. Second, the use of the term "parcellation" allows to generalize the discussion to the two different tasks that can be performed by SENPAI step2, i.e., neuronal separation and spines assignment. Third, what SENPAI performs is always a parcellation of the image space (or a division into small parcels of the image space): what comes next, i.e., neuronal separation or spines assignment, is a consequence.

L76-81: Very long sentence. Consider revision

The sentence

"As such, their success is strongly influenced by both the acquisition modality and cell type and by the kind of features they rely on (e.g., local methods based on a progressive propagation of the tracing or segmentation from a reference point are particularly sensitive to noise or local inhomogeneity, while global methods based on SNR will fail at handling smooth changes in SNR across the image stack) to the point that developing general-purpose solutions is often impracticable."

Was broken into:

"As such, their success is strongly influenced by both the acquisition modality and cell type and by the kind of features they rely on. For example, local methods based on a progressive propagation of the tracing or segmentation from a reference point are particularly sensitive to noise or local inhomogeneity, while global methods based on SNR will fail at handling smooth changes in SNR across the image stack). For this reason, developing general-purpose solutions is often impracticable."

L126: "close packed", should be "closely packed"?

The sentence

"We employ morphological reconstruction and watershed transform to ensure the correct separation of neurons when they are close packed."

Was changed as follows:

"We employ morphological reconstruction and the watershed transform to ensure the correct separation of neurons when they are closely packed."

L130: "mice line", should be "mouse line"?

We replaced "mice line" with "mouse line".

L180: "spine head", should be "spine heads"

We replaced "spine head" with "spine heads".

L181: same for spine neck

We replaced "spine neck" with "spine necks".

L182: "take up"; consider "improve"

We replaced "take up" with "improve".

Consider using "neuronal" instead of "neural" throughout the text

We replaced every instance of "neural" in the text with "neuronal".

Figure2: The figure has a panel F that is not referred to, neither in the legend, nor in the text.

Legend of Figure 2 now separates the comments for panels E and F, as follows:

E) SENPAI workflow - Step 2 for neuron separation: the segmented image is parcellated using morphological reconstruction and 3D watershed transform computed on the morphologically reconstructed grayscale image and applied to the binary segmentation. Left: raw image; Right: isolation of connected structures belonging to the same neuron: soma markers (yellow, placed by user) define wells for the catchment basins (edges in gray). **F)** SENPAI workflow - Step 2 applied to spine assignation. Left: raw image; Middle: 3D rendering; Right: 2D rendering of the parcellation with the connection of a neuronal portion (e.g., a dendrite branch) to smaller clusters (i.e., dendritic spines). Groups of neuronal clusters assigned to a single neuronal entity are displayed in different colors in their relative catchment basin (grey)."

L330: "Neuron branches", should be "dendritic branches"?; consider revision throughout the text

All instances of "neuron branches" in the text were changed to "dendritic branches".

L354-358: The authors claim "significantly larger"; they need to provide the statistical test either in the legend of the figure or in the text.

We added a statistical test for the results presented in Figure 6B.

L361-362: Please add a statistic comparison/test for the numbers presented. (See comment above)

We added a statistical test for the results presented in Figure 6B. In the legend, we added

"volume result showed significant differences; Friedman $p=0.015$; post-hoc with Conover's test highlighted Ilastik difference to both SENPAI and HK-Icy, $p<0.01$, highlighted with grey bars"

L363-369: Statistic test for spine comparisons need also to be added.

We added a statistical test for spine comparisons, using Friedman and post-hoc Conover's tests. In the legend, we added

"The analysis on the spines was conducted by masking out the manual segmentation of the dendrite (both whole neuron and average spine Dice coefficients from SENPAI were different to both Ilastik and HK-Icy; Friedman tests $p=0.015$; post-hoc with Conover's tests, $p<0.01$)."

The reviewer is still not fully convinced that SENPAI outperforms HK-Icy. From my point of view both algorithms perform similar and might have certain advantages and disadvantages.

We thank the reviewer for the comment. As we discussed in the paper, we believe that there is no single test or metric that can define a general-purpose best performing algorithm. When tested on different cell types or acquisition modalities, results might be different. For this reason, we presented SENPAI results in the specific task of segmenting dense scenarios from clarified samples. Nonetheless, statistics as those reported in Figure 4 and Figure 6 show statistically better performance for SENPAI. We also provide results from testing conducted on public datasets, providing further indications on the performance of our algorithm.

Rebuttal letter, reply to reviewer L446: White laser...

Reply: I am disappointed. The authors reply to the reviewer that they used 594nm excitation wavelength of a white pulsed laser and emission was filtered at 600-640 nm. However, they do not provide these details in their manuscript text. Indeed the authors need to provide these details for all imaging modalities they carried out. Their findings will not be reproducible without these specifications.

We now correct and report these details in the manuscript, namely in the "Testing datasets" paragraph of the Methods section:

"The algorithm was tested on datasets acquired with a Nikon A1 confocal microscope (40x objective, excitation length 457 nm, bandwidth 500-550nm), and a LEICA SP8 STED 3DX (93x objective, pulsed white-light laser 598 nm, pulsed 775 nm depletion laser, bandwidth 605-777 nm)."

All the images acquired by our team will be available on ZENODO along with complete metadata info.

Chapter "Generalization of other neuron types and imaging modalities"

It would have been informative to include a figure illustrating the results of the chapter. In addition, the paragraph lacks a conclusive sentence at the end.

We added a Figure to the Manuscript (Figure 7) with two examples of results obtained on non-clarified samples. We report below the figure and its caption:

Figure 7 – Exemplary results obtained on non-clarified samples. For both images we report the maximum projection on the left, the segmentation obtained with SENPAI on the middle (depth color-coded, cold colors indicate deeper planes), and the skeletonization of the segmentation on the right, as obtained using the NeuTube software. **Above**) Exemplary dataset (m16_cing_1_9_cropped_neurona.v3dpbd, human pyramidal cell labeled with Lucifer Yellow and acquired through confocal microscopy, resolution $0.24 \mu\text{m} \times 0.24 \mu\text{m} \times 0.42 \mu\text{m}^7$) from the human Allen confocal dataset. **Below**) 3D stack of cultured rat hippocampal pyramidal cells (pixel size $91,41 \text{ nm} \times 91,41 \text{ nm} \times 280 \text{ nm}$). Further tests on non-clarified samples are reported in the Supplementary Information.

L488: “reporter mice lines”, should be “reporter mouse lines”?

We replaced “reporter mice lines” with “reporter mouse lines”.

Reviewer #1 (Remarks on code availability):

Code was accessible. However, the reviewer has no expertise in reviewing MatLab code.

Reviewer #2 (Remarks to the Author):

I appreciate the efforts made by the authors to address my previous concerns. The manuscript has significantly improved, and I support its publication. I only found two minor mistakes that should be fixed:

1) In the legend for Figure 2, the description for panel F is mistakenly included in that for panel E.

Legend of Figure 2 now separates the comments for panels E and F, as follows:

E) SENPAI workflow - Step 2 for neuron separation: the segmented image is parcellated using morphological reconstruction and 3D watershed transform computed on the morphologically reconstructed grayscale image and applied to the binary segmentation. Left: raw image; Right: isolation of connected structures belonging to the same neuron: soma markers (yellow, placed by user) define wells for the catchment basins (edges in gray). **F)** SENPAI workflow - Step 2 applied to spine assignation. Left: raw image; Middle: 3D rendering; Right: 2D rendering of the parcellation with the connection of a neuronal portion (e.g., a dendrite branch) to smaller clusters (i.e., dendritic spines). Groups of neuronal clusters assigned to a single neuronal entity are displayed in different colors in their relative catchment basin (grey)."

2) In Table S2, row F, the total normalized dendritic length per SO obtained from the Ilastik algorithm aligns with the published range and should be underlined. Conversely, in row G, none of the computed values align with the published values and should therefore not be underlined.

The reviewer is right for what concerns row F, therefore we underlined the value. On the opposite, for row G, the range [-0.45 -0.17] refers to 3D-distributed morphologies, while for planar morphologies such as Purkinje the indication is "close to zero", therefore all values are aligned.

Additionally, I recommend a thorough proofreading of the final manuscript to catch minor language and grammatical errors, such as the repeated use of "state-of-art" instead of the correct "state-of-the-art."

The manuscript was revised by a native english speaker.

Reviewer #2 (Remarks on code availability):

The code faithfully reproduces the segmentation shown in Figure 3 of the manuscript and includes a README file with enough detail to run the code.

Reviewer #3 (Remarks to the Author):

The authors revised the manuscript in accordance with the reviewer's comments. In my opinion the paper is publishable in present form.

Reviewer #3 (Remarks on code availability):

The results are reproducible with the code.